

**Iodine chemistry after dark**
Alfonso Saiz-Lopez[1], John M.C. Plane[2], Carlos A. Cuevas[1], Anoop S. Mahajan[3], Jean-François
Lamarque[4] and Douglas E. Kinnison[4]
[1]Department of Atmospheric Chemistry and Climate, Institute of Physical Chemistry
Rocasolano, CSIC, Madrid, Spain
[2]School of Chemistry, University of Leeds, Leeds, UK
[3]Indian Institute of Tropical Meteorology, Pune, India
[4]Atmospheric Chemistry Observations and Modelling, NCAR, Colorado, USA
Correspondence to: A. Saiz-Lopez (a.saiz@csic.es)



**Abstract**
Little attention has so far been paid to the nighttime atmospheric chemistry of iodine species.
Current atmospheric models predict a buildup of HOI and $I_2$ during the night that leads to a spike
of IO at sunrise, which is not observed by measurements. In this work, electronic structure
calculations are used to survey possible reactions that HOI and $I_2$ could undergo at night in the
lower troposphere, and hence reduce their nighttime accumulation. The new reaction $NO_3$ + HOI
$\rightarrow$ IO + $HNO_3$ is proposed, with a rate coefficient calculated from statistical rate theory over the
temperature range 260 - 300 K and at a pressure of 1000 hPa to be $k(T) = 2.7 \times 10^{-12} (300 \text{ K} / T$
$)^{2.66}$ $cm^3$ $molecule^{-1}$ $s^{-1}$. This reaction is included in two atmospheric models, along with the
known reaction between $I_2$ and $NO_3$, to explore a new nocturnal iodine radical activation
mechanism. The results show that this iodine scheme leads to a considerable reduction of
nighttime HOI and $I_2$, which results in the enhancement of more than 25% of nighttime ocean
emissions of HOI + $I_2$ and the removal of the anomalous spike of IO at sunrise. We suggest that
active nighttime iodine can also have a considerable, so far unrecognized, impact on the
reduction of the $NO_3$ radical levels in the MBL and hence upon the nocturnal oxidizing capacity
of the marine atmosphere. The effect of this is exemplified by the indirect effect on dimethyl
sulfide (DMS) oxidation.



## 1. Introduction

Active nighttime iodine chemistry was first evidenced a decade ago when it was shown that nocturnal $I_2$ emitted by macroalgae could react with $NO_3$ leading to the formation of IO and OIO, which were measured in the coastal marine boundary layer (MBL) at Mace Head, Ireland (Saiz-Lopez and Plane, 2004). The nitrate radical has also been recently suggested as a nocturnal loss of $CH_2I_2$, which helps to reconcile observed and modelled concentrations of this iodocarbon over the remote MBL (Carpenter et al., 2015). However, most of the work on reactive atmospheric iodine has focused on the use of daytime observations and models to assess its role in the catalytic destruction of ozone and the oxidizing capacity of the troposphere (e.g. Saiz-Lopez et al. (2012b) and references therein). In the MBL, iodine, along with bromine, catalysed ozone destruction contributes up to 45% of the observed daytime depletion (Read et al., 2008; Mahajan et al., 2010a), although this contribution shows large geographical variability (Mahajan et al., 2012; Gómez Martín et al., 2013; Prados-Roman et al., 2015b; Volkamer et al., 2015). Iodine compounds have also been implicated in the formation of aerosols, although the mechanisms and magnitudes of these processes are not fully understood (O'Dowd et al., 2002; McFiggans et al., 2004; Saunders and Plane, 2005; Pechtl et al., 2006; Saiz-Lopez et al., 2006; Mahajan et al., 2009a; Hoffmann et al., 2001; Gomez Martin et al., 2013; Sommariva et al., 2012; Allan et al., 2015; Roscoe et al., 2015). Reactive forms of inorganic iodine may also contribute to the oxidation of elemental mercury over the tropical oceans (Wang et al., 2014). In recent years, iodine sources and chemistry have also been implemented in global models demonstrating the effect of iodine chemistry in the oxidation capacity of the global marine troposphere (Ordóñez et al., 2012; Saiz-Lopez et al., 2012a; Saiz-Lopez et al., 2014; Sherwen et al., 2016).





Iodine is emitted into the atmosphere from the ocean surface in both organic and inorganic
forms. The main organic compounds emitted are methyl iodide ($CH_3I$), ethyl iodide ($C_2H_5I$), and
propyl iodide (1- and 2-$C_3H_7I$), chloroiodomethane ($CH_2ICl$), bromoiodomethane ($CH_2IBr$), and
diiodomethane ($CH_2I_2$) (Carpenter, 2003; Butler et al., 2007; Jones et al., 2010; Mahajan et al.,
2012). However, these organic compounds contribute only up to a third of the MBL iodine
loading (Großmann et al., 2013; Mahajan et al., 2010a; Jones et al., 2010; Prados-Roman et al.,
2015b). Inorganic emissions of HOI and $I_2$, which result from the deposition of $O_3$ at the ocean
surface and subsequent reaction with $I^-$ ions in the surface microlayer, account for the main
source of iodine in the MBL (Carpenter et al., 2013). Recent laboratory experiments have shown
that HOI is the major compound emitted, and provided parameterizations of the fluxes of both
species depending on wind speed, temperature, and the concentrations of $O_3$ and $I^-$ (Carpenter et
al., 2013; MacDonald et al., 2014). These parameterized fluxes of HOI and $I_2$ have then been
used in a one-dimensional model to study the diurnal evolution of the IO and $I_2$ mixing ratios at
the Cape Verde Atmospheric Observatory (CVAO) (Carpenter et al., 2013; Lawler et al., 2014).
The model simulations replicate well the levels and general diurnal profiles of IO and $I_2$,
although an early morning 'dawn spike' in IO is predicted by the models, but has not been
observed  (Read et al., 2008; Mahajan et al., 2010a). The morning peak predicted by current
iodine chemistry models is due to a buildup of the emitted $I_2$ and HOI (which is converted into
$I_2$/IBr/ICl through heterogeneous sea-salt recycling) over the course of the night, followed by
rapid photolysis at sunrise.
Traditionally it has been thought that iodine chemistry has a negligible effect on oxidizing
capacity of the nocturnal marine atmosphere. As a consequence, unlike the demonstrated effect
of iodine on the levels of daytime oxidants, the impact of active iodine upon the main nighttime





oxidant, $NO_3$, remains an open question. This is important given that in many parts of the ocean
the $NO_3$ + DMS reaction is at least as important as OH + DMS in oxidizing DMS (Allan et al.,
2000), and hence a reduction of $NO_3$ may have an effect in the production of $SO_2$ and methane
sulfonic acid (MSA). Here, we discuss possible mechanisms of nighttime iodine radical
activation and their potential effect on nighttime iodine ocean fluxes and the currently modeled
dawn spike in IO. A new reaction of HOI with $NO_3$ is proposed, supported by theoretical
calculations. We explore the implications of this new reaction both for iodine and $NO_3$
chemistries.
**2. Nocturnal iodine radical activation mechanism**
We use the reaction mechanism that has recently been described in a global modelling study by
Saiz-Lopez et al. (2014). In addition to the reactions included in that scheme, we also include
nighttime gas-phase reactions based on the theoretical calculations described below. The
additional reactions are listed in Table 1 and a scheme with this new nocturnal chemistry is
included in Figure 1.
To the best of our knowledge, reactions of HOI specific to night time have not been studied,
either theoretically or through laboratory experiments. Currently, HOI is thought to build up
overnight until sunrise, with only heterogeneous uptake on seasalt aerosol as a nighttime loss
process (Saiz-Lopez et al., 2012b; Simpson et al., 2015). In addition to the well known $I_2$ + $NO_3$
reaction (R1) (Chambers et al., 1992), here we consider several possible HOI reactions that could
occur at night, in the absence of photolysis and OH:

22        $HOI + NO_2 \rightarrow I + HNO_3$                          (R2)





$HOI + HNO_3 \rightarrow IONO_2 + H_2O$ (R3)
$HOI + NO_3 \rightarrow IO + HNO_3$ (R4)
**3. Theoretical calculations**
In order to explore the feasibility of reactions 2–4 taking place under the conditions of the lower
troposphere, we carried out electronic structure calculations using the hybrid density
functional/Hartree-Fock B3LYP method from within the Gaussian 09 suite of programs (Frisch
et al., 2009), combined with a G2 level basis set for I (Glukhovtsev et al., 1995) and the standard
6-311+g(2d,p) triple zeta basis set for O, N and H. Following geometry optimizations of the
relevant points on the potential energy surfaces, and the determination of their corresponding
vibrational frequencies and (harmonic) zero-point energies, energies relative to the reactants
were obtained. Spin-orbit splittings of -17 and -5 kJ mol$^{-1}$ were applied to the energies I and IO,
respectively; these were estimated by comparing the theoretical and experimental bond energies
of I$_2$ and IO (Plane et al., 2006; Kaltsoyannis and Plane, 2008).
Reaction 2 is endothermic by 9 kJ mol$^{-1}$ and so, within the expected error of ±10 kJ mol$^{-1}$ at this
level of theory, might be reasonably fast. However, the transition state of the reaction, which is
illustrated in Figure 2(a), is 73 kJ mol$^{-1}$ above the reactants and so this reaction will not occur at
tropospheric temperatures. Reaction 3 is exothermic by 11 kJ mol$^{-1}$. An HOI--HNO$_3$ complex
first forms (Figure 2(b)), which is 21 kJ mol$^{-1}$ below the reactants. However, this complex
rearranges to the IONO2 + H$_2$O products via the cyclic transition state shown in Figure 2(c),
which is 110 kJ mol$^{-1}$ above the reactants.



The stationary points on the potential energy surface (PES) for reaction 4 are illustrated in Figure
3. HOI and $NO_3$ associate to form a complex which is 24 kJ mol$^{-1}$ below the reactant entrance
channel. H-atom transfer involves a submerged transition state to form a IO--$HNO_3$ complex,
which can then dissociate to the products IO + $HNO_3$. Overall, the reaction is exothermic by 11
kJ mol$^{-1}$. The vibrational frequencies, rotational energies and geometries (in Cartesian
co-ordinates) of these intermediates are listed in Table 2.
The rate coefficient for reaction 4 was then estimated using Rice-Ramsperger-Kassel-Markus
(RRKM) theory, employing a multi-well energy-grained master equation solver based on the
inverse Laplace transform method - MESMER (Master Equation Solver for Multi-well Energy
Reactions) (Roberston et al., 2014). The reaction proceeds via the formation of the excited
HOI--$NO_3$ complex from HOI + $NO_3$. This complex can then dissociate back to the reactants or
rearrange to the IO--$HNO_3$ intermediate complex over the transition state, which can in turn
dissociate to the products IO + $HNO_3$. Either of intermediates can also be stabilized by collision
with the third body ($N_2$). The time evolution of all these possible outcomes is modelled using the
master equation.
The internal energies of the intermediates on the PES were divided into a contiguous set of
grains (width 10 cm$^{-1}$), each containing a bundle of rovibrational states calculated with the
molecular parameters in Table 2. Each grain was then assigned a set of microcanonical rate
coefficients linking it to other intermediates, calculated by RRKM theory. For dissociation to
products or reactants, microcanonical rate coefficients were determined using inverse Laplace
transformation to link them directly to the capture rate coefficient, $k_{capture}$. For reaction 4 and the
reverse reaction IO + $HNO_3$ involving neutral species, $k_{capture}$ was set to a typical capture rate
coefficient of $2.5 \times 10^{-10}$ ($T$/300 K)$^{1/6}$ cm$^3$ molecule$^{-1}$ s$^{-1}$, where the small positive temperature





dependence is characteristic of a long-range potential governed by dispersion and dipole-dipole
forces (Georgievskii and Klippenstein, 2005).
The probability of collisional transfer between grains was estimated using the exponential down
model, where the average energy for downward transitions was set to $<\Delta E>_{down}$ = 300 cm$^{-1}$ for
N$_2$ as the third body (Gilbert and Smith, 1990). MESMER determines the temperature- and
pressure-dependent rate coefficient from the full microcanonical description of the system time
evolution by performing an eigenvector/eigenvalue analysis (Bartis and Widom, 1974). The
resulting rate coefficient over the temperature range 260 - 300 K at a pressure of 1000 hPa is
$k_4(T)$ = 2.7 × 10$^{-12}$ (300 K / $T$ )$^{2.66}$ cm$^3$ molecule$^{-1}$ s$^{-1}$. Because the intermediate complexes are
not strongly bound, and the transition state and products are below the entrance channel, the only
products formed in reaction 4 under atmospheric conditions are IO + HNO$_3$. The absence of a
barrier above the entrance channel means that the uncertainty in $k_4$ principally arises from the
estimated capture rate coefficient and so is likely to be no more than a factor of 2.
Note that NO$_3$ also reacts with CH$_2$I$_2$ with a rate constant ~2-4×10$^{-13}$ cm$^3$ molecule$^{-1}$ s$^{-1}$, which
can have a significant effect on nighttime CH$_2$I$_2$ concentration (Carpenter et al., 2015). However
the products of this reaction are still uncertain (Nakano et al., 2006; Carpenter et al., 2015) and
its rate is considerably slower than that of R4.
In summary, the only likely gas-phase reactions that I$_2$ and HOI undergo in the nighttime
troposphere are R1 and R4, respectively. These are included in the model reaction scheme to
examine their impacts on the evolution of iodine species in the atmosphere.



## 4. Atmospheric modelling

We use two atmospheric chemical transport models to study *i*) the impact of this new chemistry on the nighttime chemistry and partitioning of iodine species, and *ii*) the resulting geographical distribution of nocturnal iodine and impact on $NO_3$ within the global marine boundary layer.

The first model, Tropospheric HAlogen chemistry MOdel (THAMO), is used for a detailed kinetics study of the impact of the different reactions shown in Table 1 as well as to assess which uptake rates best reproduce observations from a field study at the CVAO (Carpenter et al., 2011). THAMO has been used in the past to study iodine chemistry at the CVAO and further details including the full chemical scheme can be found elsewhere (Saiz-Lopez et al., 2008; Mahajan et al., 2009b; Mahajan et al., 2010b; Mahajan et al., 2010a; Lawler et al., 2014; Read et al., 2008). Briefly, THAMO is a 1-D chemistry transport model with 200 stacked boxes at a vertical resolution of 5m (total height 1 km). The model treats iodine, bromine, $O_3$, $NO_x$ and $HO_x$ chemistry, and is constrained with typical measured values of other chemical species in the MBL: [CO]=110 nmol mol$^{-1}$; [DMS]=30 pmol/mol; [CH4]=1820 nmol mol$^{-1}$;[ethane]=925 pmol/mol; [CH3CHO]=970 pmol/mol; [HCHO]=500 pmol/mol; [isoprene]=10 pmol/mol; [propane]=60 pmol/mol; [propene]=20 pmol/mol. The average background aerosol surface area (ASA) used is $1\times10^{-6}$ cm$^2$ cm$^{-3}$ (Read et al., 2008; Read et al., 2009; Lee et al., 2009; Lee et al., 2010). The model is initialized at midnight and the evolution of iodine species, $O_3$, $NO_x$ and $HO_x$ is followed until the model reaches steady state.

The second model is the global 3D chemistry-climate model CAM-Chem (Community Atmospheric Model with chemistry, version 4.0), which is used to study the impact of reactions 1 and 4 on a global scale. The model includes a comprehensive chemistry scheme to simulate the





evolution of trace gases and aerosols in the troposphere and the stratosphere (Lamarque et al.,
2012). The model runs with the iodine and bromine chemistry schemes from previous studies
(Fernandez et al., 2014; Saiz-Lopez et al., 2014; Saiz-Lopez et al., 2015), including the
photochemical breakdown of bromo- and iodo-carbons emitted from the oceans (Ordóñez et al.,
2012) and abiotic oceanic sources of HOI and $I_2$ (Prados-Roman et al., 2015a). CAM-Chem has
been configured in this work with a horizontal resolution of 1.9º latitude by 2.5º longitude and 26
vertical levels, from the surface to ~40km altitude. All model runs in this study were performed
in the specified dynamics mode (Lamarque et al., 2012) using offline  meteorological fields
instead of an online calculation, to allow direct comparisons between different simulations. This
offline meteorology consists of a high frequency meteorological input from a previous free
running climatic simulation.
**5. Results and discussion**
Of the possible nocturnal iodine activation reactions involving the inorganic iodine source gases
$I_2$ and HOI, only reactions R1 and R4 appear to be likely candidates (see Section 3). We
therefore designed two modelling scenarios: Scenario 1 (S1), without nighttime reactions of $I_2$ or
HOI with $NO_3$; and Scenario 2 (S2), including reactions R1 and R4 for the degradation of HOI
and $I_2$ by $NO_3$. In the one-dimensional model THAMO, the $I_2$ and HOI are injected into the
atmosphere from the ocean surface using the flux parameterizations derived from laboratory
experiments (MacDonald et al., 2014; Carpenter et al., 2013). Figure 4 shows the resulting
diurnal evolution of the HOI and $I_2$ mixing ratios in the two scenarios. The $I_2$ mixing ratio peaks
during the night in both the scenarios due to quick loss by photolysis during the daytime. By





contrast, HOI peaks during the daytime due to its production through the reaction of IO with
$HO_2$. In the first scenario, without the inclusion of reactions R1 and R4, Figure 4 (right-hand side
panels) shows that HOI and $I_2$ both build up during the night, reaching a concentration peak just
before dawn. This is especially noticeable for $I_2$ as the daytime concentrations are much lower
than during the night. For both species, inclusion of reactions with $NO_3$ causes a decrease in their
respective nocturnal concentrations (Fig. 4, left-hand side panels). The inclusion of reactions R1
and R4 also leads to a modelled $I_2$ concentration which is in better agreement with the
observations of the molecule made at CVAO (Lawler et al., 2014), reaching peak values of
about 1 pmol/mol, as compared to about 3 pmol/mol for the scenario without nighttime reactions.
An additional consequence of including reactions R1 and R4 is the significant increase of the
sea-air fluxes of HOI and $I_2$ at night due to their atmospheric removal by $NO_3$ (Fig. 4, bottom
panel).
It should be noted that during nighttime the uptake of emitted species such as $I_2$ and HOI, and the
uptake of reservoir species such as $IONO_2$, can play a major role in the cycling of iodine.
Observations at CVAO show that $I_2$ peaked at about 1 pmol/mol during the night and that ICl
was not detected above the 1 pmol/mol detection limit of the instrument (Lawler et al., 2014). In
order to match these observations, we need to reduce the uptake and heterogeneous recycling of
iodine species. The uptake rates of chemical species on the background seasalt aerosols are
determined by their uptake coefficients ($\gamma$). The database of mass accommodation and/or uptake
coefficients is rather sparse and essentially limited to $I_2$, HI, HOI, ICl, IBr on pure water/ice and
on sulphuric acid particles (Sander et al., 2006). Other iodine species which are likely to undergo
uptake onto aerosol are OIO, $HIO_3$, $INO_2$, $IONO_2$, $I_2O_2$ (Saiz-Lopez et al., 2012a; Sommariva et
al., 2012). Uptake of HOI is very uncertain, with $\gamma$(HOI) ranging from $2 \times 10^{-3}$ to 0.3 depending





on the surface composition and state  (Holmes et al., 2001). Sommariva et al. (2012) assumed
$\gamma$(HOI) to be 0.6, similar to the value for HOBr measured by Wachsmuth et al. (2002). In the
case of IONO$_2$, the uptake coefficient has not been measured, with most models using values of
0.1 (von Glasow et al., 2002; Saiz-Lopez et al., 2008; Mahajan et al., 2009b; Mahajan et al.,
2010b; Mahajan et al., 2010a; Leigh et al., 2010; Sommariva et al., 2012; Lawler et al., 2014).
The modelled levels of I$_2$ and ICl change with different values of uptake coefficients. To match
the CVAO I$_2$ and ICl observations (Lawler et al., 2014), we have used $\gamma = 0.01$ for HOI and
IONO$_2$, which is within the uncertainty in the literature, and assumed that 80% is recycled as I$_2$.
Further measurements of these dihalogen species are needed to better constrain their
heterogeneous recycling on seasalt aerosols.
Figure 5 shows the diurnal evolution of IO, NO$_3$ and IONO$_2$ in both model scenarios. Although
the daytime peak values of IO  are well reproduced in both scenarios, reaching about 1.5
pmol/mol around noon similar to the ground-based observations (Read et al., 2008), the inclusion
of reactions R1 and R4 leads to the removal of the dawn spike in IO, which is predicted by
current iodine models but was not observed at CVAO (Read et al., 2008; Mahajan et al., 2010a) .
The IO dawn spike predicted by models is due to a buildup of the emitted I$_2$ and HOI (which is
converted into I$_2$/IBr/ICl through heterogeneous recycling) over the night, followed by rapid
photolysis after first sunlight. However, due to the considerable removal of HOI and I$_2$ through
the night due to reaction with ambient NO$_3$, this spike does not appear in the second scenario,
leading to a modification of the diurnal profile of IO that better matches with observations.
Reactions R1 and R4 also reduce the NO$_3$ mixing ratio (Fig. 4, middle panels). In scenario 1, the
NO$_3$ is modelled to peak at about 14 pmol/mol just before dawn. However, the inclusion of
reactions R1 and R4 leads to near complete depletion of NO$_3$ close to the surface, with the peak




level at the surface reaching only 2 pmol/mol, since reactions R1 and R4 become the main
atmospheric loss processes for $NO_3$ in the lower MBL. These reactions lead however to the
buildup of $IONO_2$ during the night (Fig. 5, bottom panels). In the absence of reactions R1 and
R4, significant levels of $IONO_2$ are seen only at dawn and dusk since no other reactions produce
$IONO_2$ at night, and during the day $IONO_2$ is removed by photolysis. However, with continuous
conversion of $I_2$ and HOI to $IONO_2$ by reactions R1 and R4 in scenario 2, $IONO_2$ is modelled to
reach up to 3 pmol/mol in the nocturnal MBL.
Given the associated uncertainty in the theoretical estimate of the $k_4$, we used THAMO to assess
the sensitivity of surface $NO_3$ to $k_4$. Figure 6 shows that $NO_3$ is in fact highly coupled to $k_4$, with
the expected uncertainty in $k_4$ of a factor of 2 (see above) giving rise to a similar uncertainty in
$NO_3$. A laboratory measurement of $k4$ should therefore be undertaken in the future.
We now implement the nighttime reactions in the 3D global model (CAM-Chem) to assess the
resulting geographical distributions and impacts of these reactions. We have also run two
different scenarios in CAM-Chem, the first without R1 and R4 in the chemical scheme, and the
second including the new nighttime iodine chemistry. Figure 7 shows how the inclusion of R1
and R4 reduces globally the nighttime concentrations of $I_2$ and HOI. The plots correspond to the
midnight averaged (from 00LT to 01LT) differences between the model scenarios. Considerable
reductions of up to 0.5 and 10 pmol/mol (i.e. up to 100% removal) are observed for $I_2$ and HOI,
respectively, particularly over coastal polluted regions where continental pollution outflow leads
to higher levels of $NO_3$ in the nighttime MBL. Major shipping routes also show strong nocturnal
iodine activity due to the characteristically high $NO_x$, and resulting $NO_3$, associated with
shipping emissions.





Figure 8 shows the effect of this nocturnal chemistry on the concentrations of $IONO_2$ and $NO_3$.
As in the previous figure, the plots correspond to the nighttime averaged difference between the
second and the first scenarios. The maps show an increase of $IONO_2$ of up to 15 pmol/mol
(~600%) over polluted coastal areas, due to efficient conversion of $NO_3$ into $IONO_2$. The bottom
panel of Figure 7 shows the expected decrease of $NO_3$ levels associated with the inclusion of
reactions R1 and R4, with decreases of up to ~4 pmol/mol (up to 60%) over marine polluted
regions. We model global percentage reductions in the $NO_3$ concentrations of 7.1% (60S-60N),
with nitrate removal of up to 80% in non-polluted remote oceanic regions with low $NO_3$ levels.
This in turn can affect the modelled oxidation of DMS by $NO_3$. We estimate that the reduction in
$NO_3$, due to the inclusion of R1 and R4, results in a model increase in DMS levels of up to 7
pmol/mol (about 20%) in marine regions affected by continental pollution outflow (Fig. 9). We
therefore suggest that the inclusion of the new nighttime iodine chemistry can have a large, so far
unrecognized, impact on the nocturnal oxidizing capacity of the marine atmosphere.
The hourly evolution of the main species involved in this study is shown in Figures 10 and 11,
which include the levels of HOI, $I_2$, $IONO_2$ and $NO_3$ in the MBL over regions where nocturnal
iodine is modelled to be particularly active. The first region is located within the Mediterranean
Sea, an area that shows large differences during the summer months when high ozone levels
drive large emissions of HOI and $I_2$ from the sea, and the high levels of $NO_3$ at nighttime make
this chemistry especially important. The hourly average in August is shown in Figure 10 for
HOI, $IONO_2$ and $I_2$. HOI and $IONO_2$ (Fig 10 ) are the species whose concentration differ most
between scenarios as HOI is removed and $IONO_2$ produced by R4 (and, to a lesser extent, R1).
Over the coastal region of the Baja California Peninsula, the modelled differences between the





two scenarios are even higher than over the Mediterranean Sea (Figure 11). Large differences in
MBL $NO_3$, up to 28%, are modelled during the night.

## 6. Summary and conclusions

The viability of the reaction of HOI with $NO_2$, $HNO_3$ and $NO_3$ has been studied by theoretical
calculations. The results indicate that only the reaction of HOI with $NO_3$, to yield IO + $HNO_3$, is
possible under tropospheric conditions. The inclusion of this reaction, along with that of $I_2$ +
$NO_3$, has a number of significant implications: *i*) nocturnal iodine radical chemistry is activated;
*ii*) this causes enhanced nighttime oceanic emissions of HOI and $I_2$; *iii*) nighttime iodine species
are partitioned into high levels of $IONO_2$; *iv*) the IO spike, modelled by current iodine models
but not shown by observations, is removed; and, *v*) a reduction of the levels of nitrate radical in
the MBL, with the associated less efficient oxidation of DMS, which has important implications
for our understanding of the nocturnal oxidizing capacity of the marine atmosphere.

## Acknowledgments

This work was supported by the Spanish National Research Council (CSIC). The National
Center for Atmospheric Research (NCAR) is funded by the National Science Foundation NSF.
The Climate Simulation Laboratory at NCAR's Computational and Information Systems
Laboratory (CISL) provided the computing resources (ark:/85065/d7wd3xhc). As part of the
CESM project, CAM-Chem is supported by the NSF and the Office of Science (BER) of the US
Department of Energy. This work was also sponsored by the NASA Atmospheric Composition





Modeling and Analysis Program Activities (ACMAP, number NNX11AH90G).

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

| No. | Reaction | Notes |
|---|---|---|
| R1. | $I_2 + NO_3 \rightarrow I + IONO_2$ | $1.5 \times 10^{-12}$ cm$^3$ molecule$^{-1}$ s$^{-1}$ [*Chambers et al., 1992*] |
| R2. | $HOI + NO_2 \rightarrow I + HNO_3$ | Endothermic by 9 kJ mol$^{-1}$ and the transition state is 73 kJ mol$^{-1}$ above the reactants |
| R3. | $HOI + HNO_3 \rightarrow IONO_2 + H_2O$ | Exothermic by 11 kJ mol$^{-1}$. The reaction first forms a complex 21 kJ mol$^{-1}$ below the reactants but this rearranges to the products via a transition state that is 110 kJ mol$^{-1}$ above the reactants. |
| R4. | $HOI + NO_3 \rightarrow IO + HNO_3$ | Exothermic by 11 kJ mol$^{-1}$ with all transition states below the reactants. $k(T) = 2.7 \times 10^{-12} (300 \text{ K} / T)^{2.66}$ cm$^3$ molecule$^{-1}$ s$^{-1}$ |





**Table 2.** Calculated vibrational frequencies, rotational constants and energies of the stationary
points and asymptotes on the HOI + NO$_3$ doublet potential energy surface

| Species | Geometry[a] | Vibrational frequencies[b] | Rotational constants [c] | Potential energy [d] |
|---|---|---|---|---|
| HOI + NO$_3$ | | 603, 1084, 3803 & 261, 261, 805,1108, 1108, 1126 | 623.9, 8.182, 8.076 & 13.84, 13.84, 6.919 | 0.0 |
| IOH-NO$_3$ complex | O 1.623,0.284,-0.331<br>H 1.484,-0.657,-0.043<br>I 0.009,1.205,0.286<br>N -0.456,-2.265,0.030<br>O -1.052, -3.321,-0.0473<br>O -1.147,-1.195,-0.228<br>O 0.742,-2.161,0.333 | 55, 84, 118, 161, 196, 615, 629, 667, 705, 803, 968, 1228, 1273,1491, 3268 | 5.610, 0.916, 0.806 | -24.0 |
| IO-H-NO$_2$ TS | O 0.309,1.515,0.247<br>H -0.834,1.314,-0.017<br>I 1.280,-0.089,-0.093<br>N -2.349,-0.133,0.019<br>O -3.518, ,-0.429,-0.035<br>O -1.444,-0.962,0.257<br>O -2.019,1.117,-0.187 | 1249*i*, 70, 97, 103, 225, 472, 676, 698, 797, 806, 1041, 1147, 1308, 1513, 1626 | 6.300, 0.864, 0.767 | -16.4 |
| IO-HNO$_3$ complex | O 0.571,1.350,0.348<br>H -1.111,1.098,-0.020<br>I 1.870,0.0645,-0.152<br>N -2.503,-0.202,0.0186<br>O -3.673,-0.396,-0.170<br>O -1.654,-0.986,0.401<br>O -2.081,1.090,-0.242 | 35, 43, 76, 126, 198, 623, 677, 703, 772, 798, 939, 1331, 1416, 1713, 3281 | 7.058, 0.605, 0.566 | -34.8 |
| IO + HNO$_3$ | | 648 & 477, 585, 649, 782, 901, 1320, 1345, 1738, 3724 | 9.844 & 13.01, 12.05, 6.258 | -10.6 |

[a] Cartesian co-ordinates in Å. [b] In cm$^{-1}$. [b] In GHz. [c] In kJ mol$^{-1}$, including zero-point energy and spin-
orbit coupling of I and IO (see text).


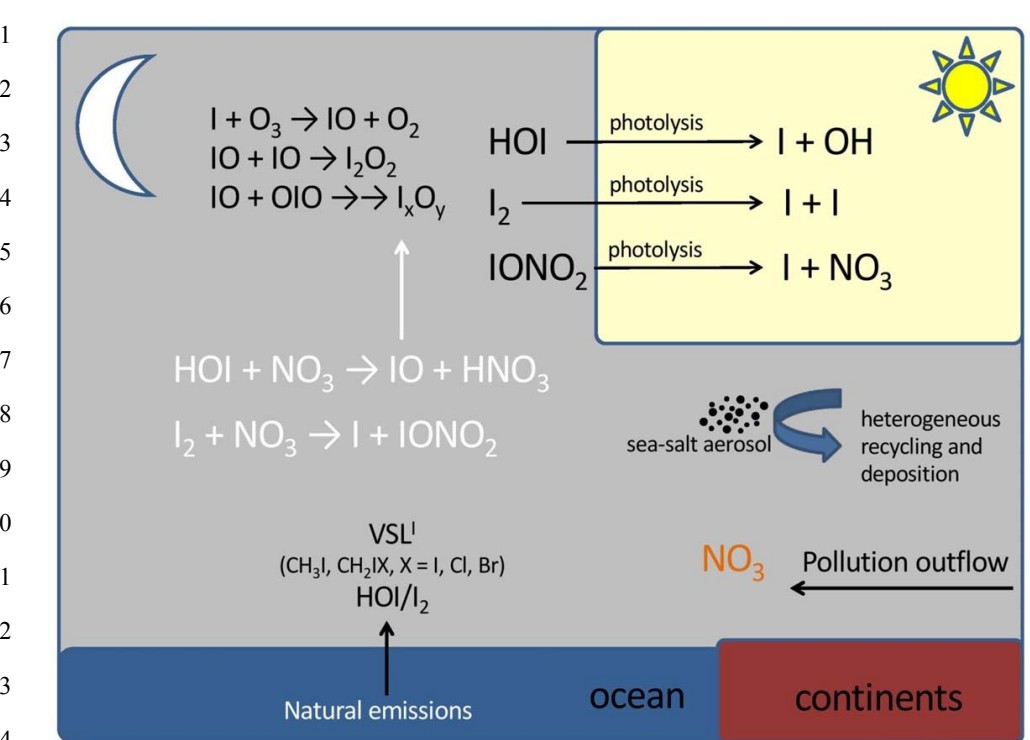

**Figure 1**. New nocturnal iodine chemistry (in white) implemented in the THAMO and CAM-Chem models.

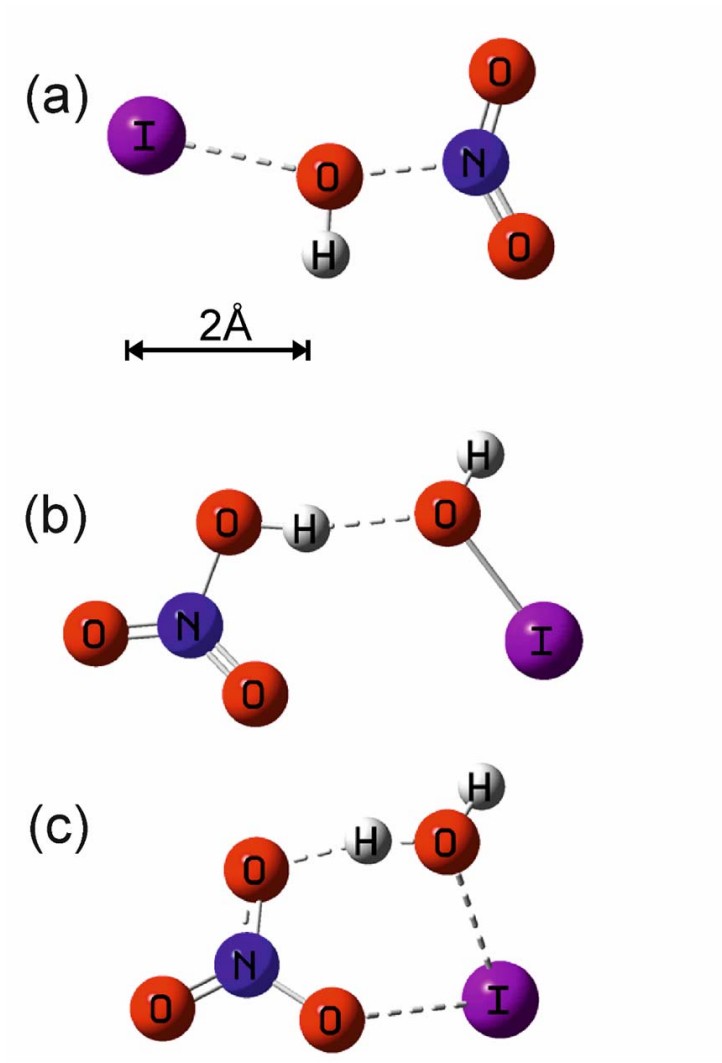

**Figure 2:** (a)  Transition state for the reaction between HOI and $NO_2$ to form $HNO_3$ + I; (b)
complex formed between HOI and $HNO_3$, which then reacts via transition state (c) to form
IONO2 + $H_2O$.



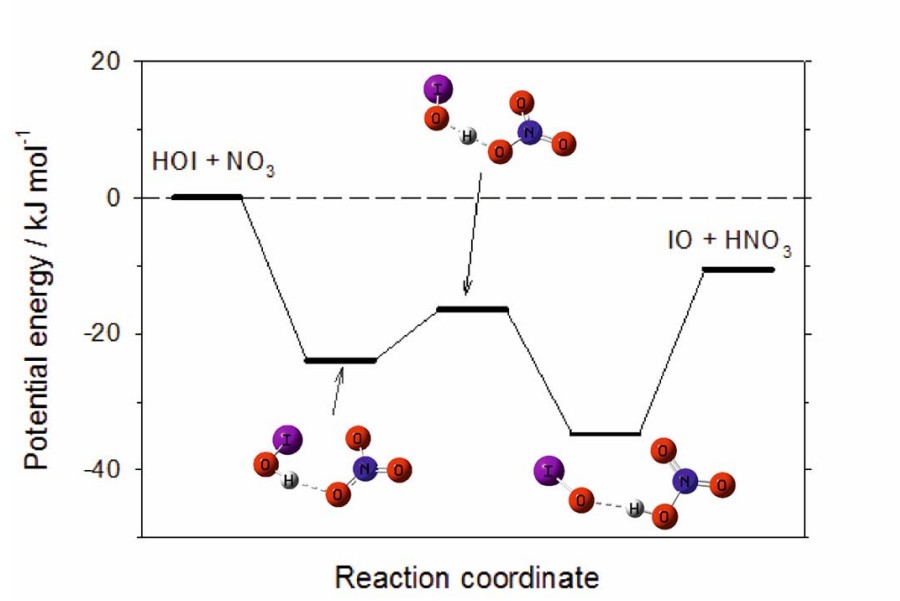

4    **Figure 3**. Potential energy surface for the reaction between HOI and $NO_3$, which contains two
5    intermediate complexes separated by a submerged barrier.



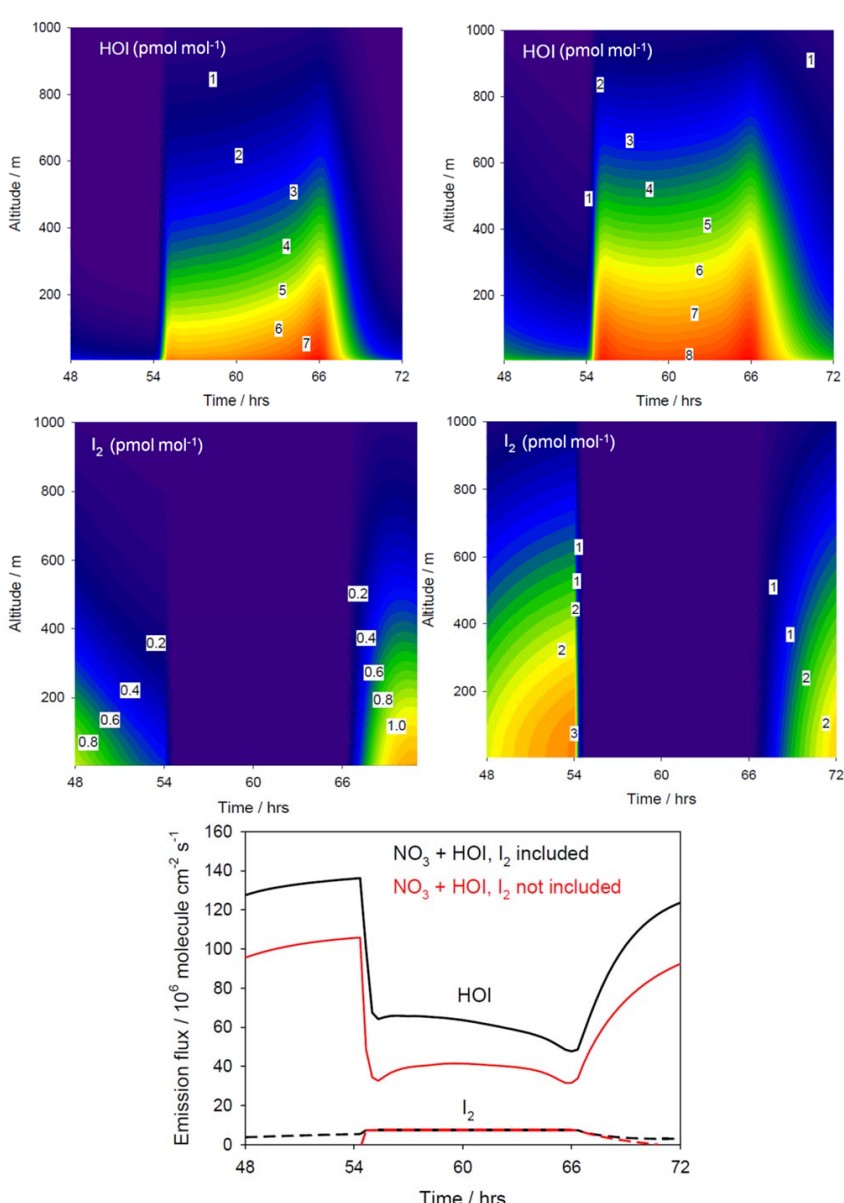

**Figure 4.** THAMO modeled diurnal variation of HOI, $I_2$ and the HOI/$I_2$ flux from the ocean
surface. The right hand panels are from scenario 1, which do not include night time reactions of
HOI and $I_2$ with $NO_3$, while the left hand panels include the reactions in scenario 2.




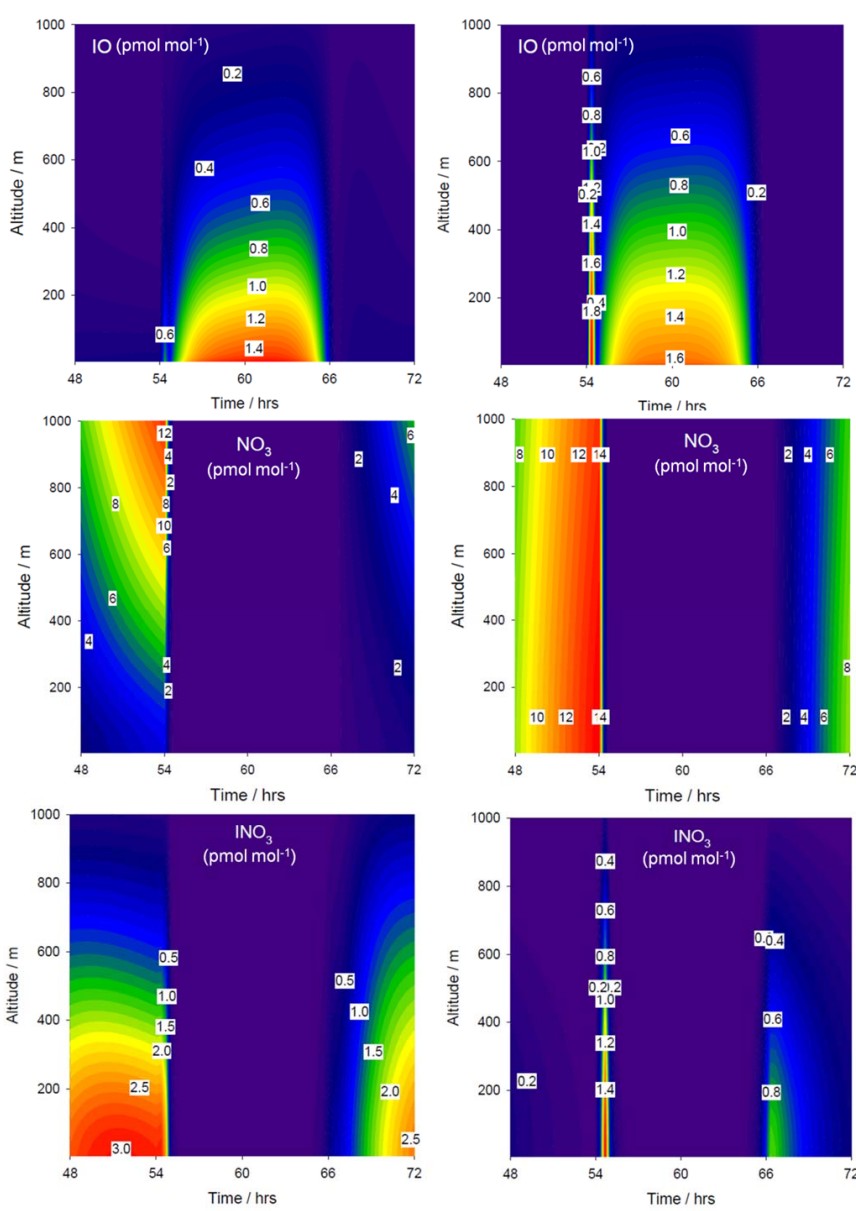

**Figure 5.** THAMO modeled diurnal variation of IO, $NO_3$ and the $IONO_2$. The right hand panels
are from scenario 1, which do not include night time reactions of HOI and $I_2$ with $NO_3$, while the
left hand panels include the reactions in scenario 2.





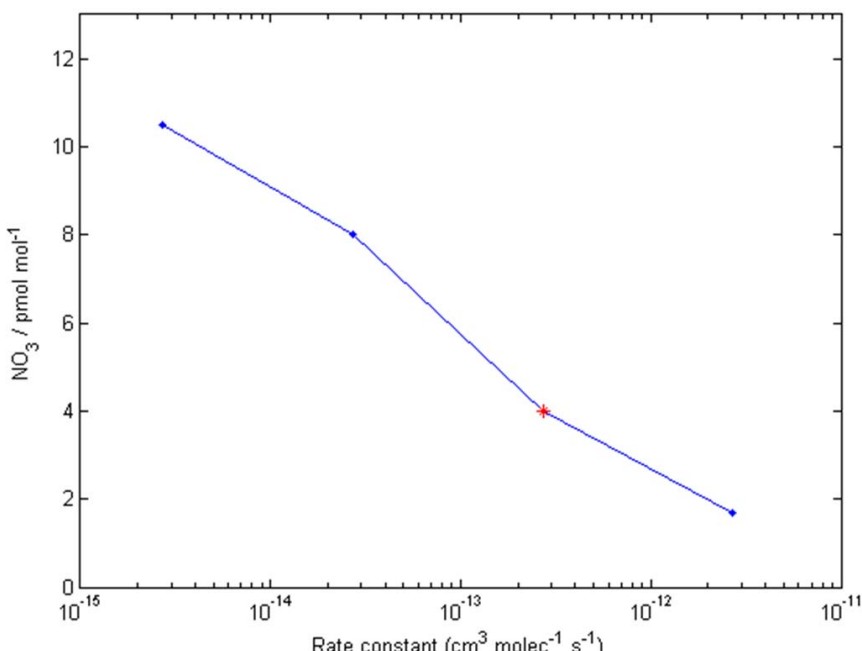

2 **Figure 6**. Sensitivity run showing the effect of the uncertainty in the rate constant estimation on

3 the reduction of $NO_3$ at the surface - the red point is the theoretical estimate.



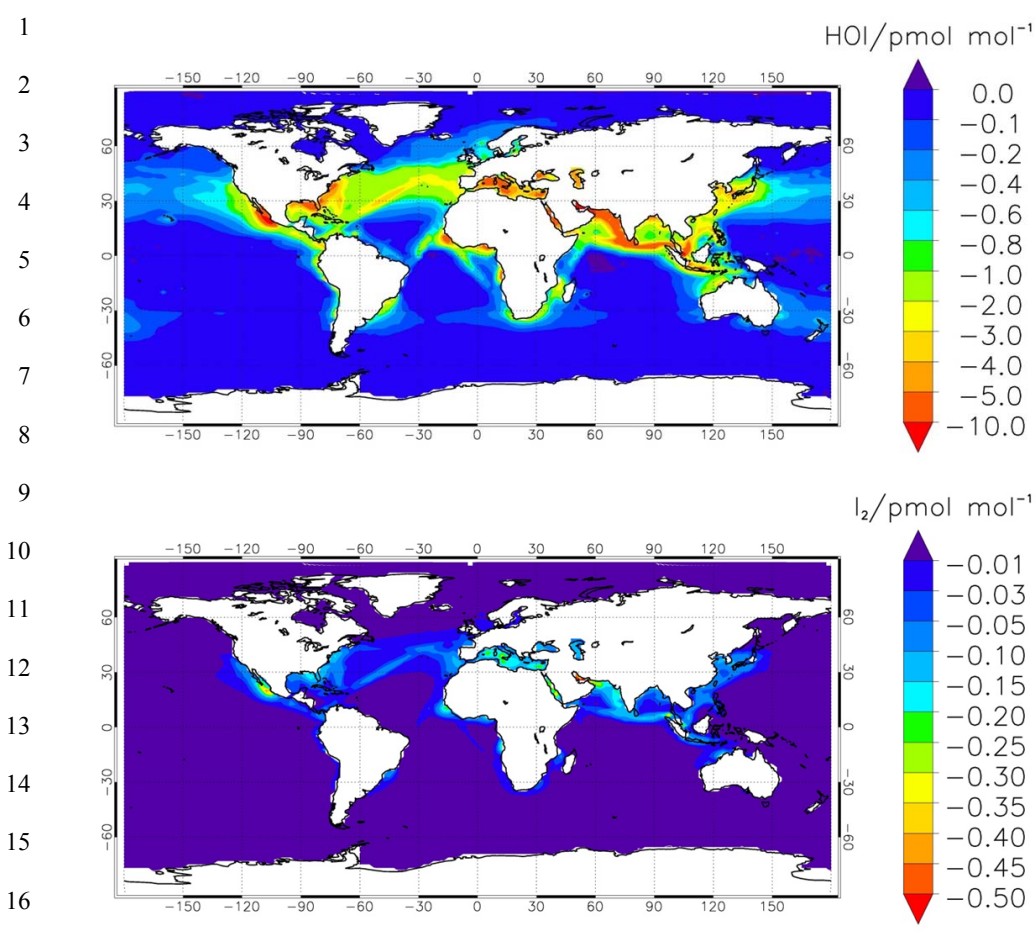

**Figure 7.** Modelled annual average of HOI (a) and $I_2$ (b) during night. The panels show the difference in vertical mixing ratio between the simulations with and without reactions (1) and (2).



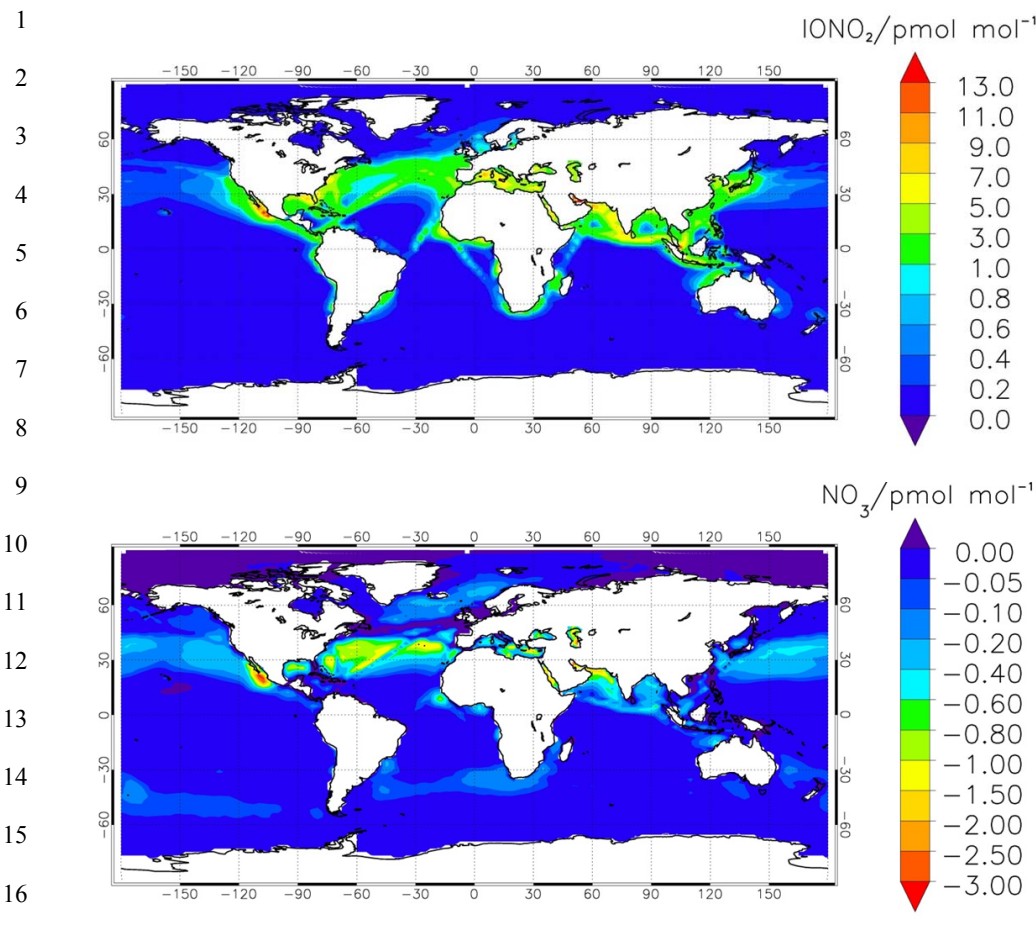

**Figure 8.** Modelled annual average of IONO$_2$ (a) and NO$_3$ (b) during night time over the ocean surface, as the difference in volume mixing ratio between the simulations with and without reactions (1) and (2).



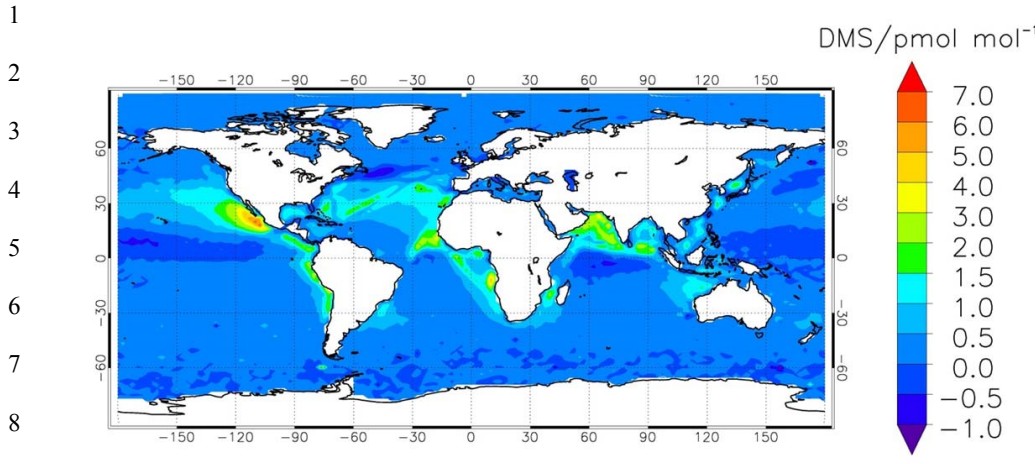

**Figure 9**. Increase in the DMS levels at night time over the ocean surface due to the inclusion of
the reactions R1 and R2 in CAM-Chem.





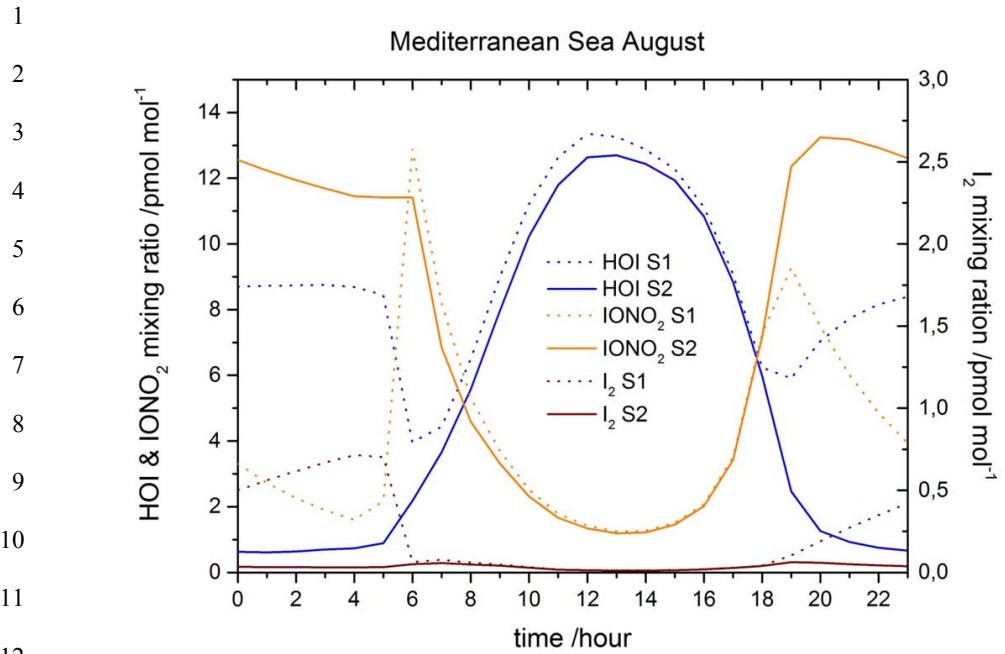

**Figure 10**. Hourly averaged concentration of HOI, IONO$_2$ and I$_2$ in the Mediterranean Sea at
surface level (lon:10º→20ºE, lat:33º→40ºN)



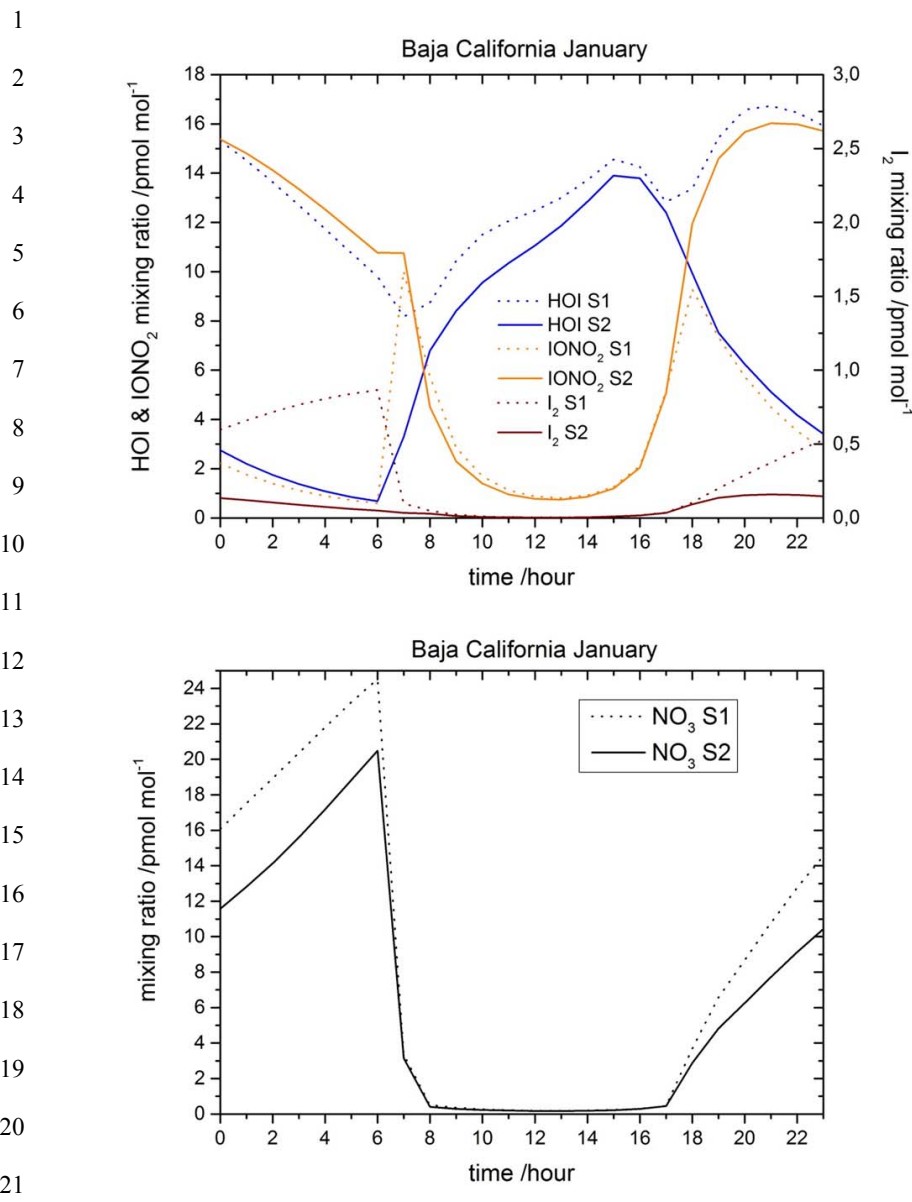

**Figure 11**. Hourly averaged concentration of HOI, IONO$_2$ and I$_2$ (upper panel) and NO$_3$ (bottom panel) in the Pacific Ocean close to Baja California (lon:-110°→-106°E, lat:16°→23°N)