# Peer review of "Nighttime atmospheric chemistry of iodine"

_Atmospheric Chemistry and Physics, 2016_

## Short Comment (SC1) · 4 Jul 2016

This paper presents a very interesting study on the iodine atmospheric chemistry using two modelling approaches: one at the molecular level, one at the global level. The potential energy surfaces for three different reactions were explored for the first time by theoretical calculations: $HOI + NO_2$, $HOI + HNO_3$, and $HOI + NO_3$. I have several comments concerning this work. 1) Spin-orbit correction (SOC) is very important for iodine-containing species. The authors stated on page 6 line 12 that "spin-orbit splittings of -17 and -5 kJ mol-1 were applied to energies of I and IO". These values do not correspond to the well known value for I atom (-30.3 kJ mol-1 from C.E. Moore, Atomic Energy Levels, USGPO, Vols. II and III. NSRDS-NBS 35, Washington, DC, 1971). Over the last years, my group performed theoretical calculations to get the SOC values for numerous iodine-containing species using the CASPT2/RASSI methodology. The corresponding values for I, IO, and HOI are -30.0, -14.4, and -5.9 kJ/mol (Meciarova

et al., CPL, 2011, 517, 149; Khanniche et al., JPCA, 2016, 120, 1737; Sulkova et al., JPCA, 2013, 117, 771). These calculations were also validated by comparison to few available data. I recommend the authors to update their energetics according to the correct SOC values. 2) In Table 2, there are several low frequency modes for molecular complexes and transition state. Are they among them one or several which should be treated as hindered rotors? It is also interesting to compare calculated vibrational frequencies for HOI, NO3, IO, HNO3 with their available experimental counterparts. 3) Nitrogen oxides also exhibits often an instability of the wavefunction (internal or RHF -> UHF). Is-it the case here for all stationary points? This instability will affect the energetics. The Gaussian09 software includes an option to check it. 4) What is the level of theory used for the energetics? It is not clear.

Concerning the atmospheric modelling, the reaction of NO3 with iodocarbons is important as noticed on page 8 line 15, it could be also important to see if adding the reactivity of atmospheric VOC with iodine-containing species will affect the results because this reaction will produce iodocarbons.
* * *

---

## Referee Comment (RC1) · H. K. Roscoe (Referee) · 5 Jul 2016

This paper makes an important point about atmospheric chemistry. It is scientifically sound and rigorous except for the few items in Minor Comments, and except perhaps for the theoretical calculations in Section 3 on which I am not competent to express an opinion. It is also well written, except for the trivia listed under Editorial Comments below.

Provided it receives a satisfactory review from experts in calculation of reaction rates, I have no hesitation in recommending it for publication in ACP after minor revision.

Minor comments:

1. p14 line22 - according to the caption of Figure 11 (p39) it applies to the region 110 to 106degE and 16 to 23degN. This region just touches the southern tip of Baja California but is centred a long way to its south. It just touches a coastal region of mainland

Mexico, but is never at the "coastal region" even of Mexico let alone the stated Baja California - much of the region is in what might be called the open ocean. Presumably this region is chosen because of the large pollution amounts there that we infer from Figures 7, 8 and 9; but there is no discussion of why they should be so large - is it a concentration of shipping using the Panama Canal that spreads out further north?

2. p6 line21 - given the argument of p6 lines17-18, why does a transition state 110 kN/mole above the reactants allow the reaction to proceed?

3. Why do Figures 4 and 5 have time co-ordinates starting at 48 hours? Is this to allow a steady state to build up? - if so it should be discussed. And what version of time is it - time since midnight or time since noon? - a careful reading of text and figures tells us which, but it should be spelled out in the caption. And why do Figures 10 and 11 have time co-ordinates that start at 0 hours rather than 48? And although we can guess that time in Figures 10 and 11 is since midnight, is it mean solar midnight over the region, or solar midnight at the geographic centre of the region, or midnight in the local time zone at 108degE?

4. We are told in the text (p14 line2) that Figure 8 has "as in the previous figure".. "nighttime averaged differences", yet p13 line17 tells us that the previous figure, Figure 7, uses "midnight averages". Which are used in which figures, and why do the captions not spell out the averaging hours as opposed to having them buried in the text?

Editorial comments:

p3 line3 introduces and defines MBL but it was already used without definition on p2 line15.

p3 line10 - surely, hyphens after "iodine" and "bromine" ?

p4 line2 - delete "and"

p6 line12 - insert "of" after "energies".

p7 line13 - insert "the" after "of".

Fig4 lowest panel - the meanings of the four lines are not in the caption and their panel legends are obscure.

Figs 7, 8 and 9 - the captions do not say the altitude or the vertical extent of the averaging.

Fig10 - the right hand axis legend says "mixing ration".

Fig10 caption - insert "the" after "at".

Fig11 caption - say the altitude.

---

## Referee Comment (RC2) · Anonymous Referee #3 · 13 Jul 2016

The manuscript describes new model calculations on the atmospheric chemistry of reactive iodine species encompassing a hypothetical reaction (R4) NO3 + HOI → IO + HNO3. The possibility of R1 actually occurring is investigated by molecular structure reactions. Moreover some possible discrepancies between observations and model calculations based on "conventional" I-chemistry may be solved by including R1.

The bulk of the manuscript is devoted to a comprehensive study of the consequences of introducing R4 (along with the earlier suggested reaction R1) in two models (1D and 2D). While one may ask whether a study based on a hypothetical reaction is warranted, I feel that the manuscript contains valuable material, which is within the scope of ACP and of interest to the scientific community. However, the manuscript contains a number of errors and unclear points (see list below) which must be corrected before publication. Also, given its speculative nature the manuscript is much too long and should be

shortened considerably. This could be done by for instance removing most of the plots based on the 2D model calculations.

In detail there are the following deficiencies:

1) Page 4, lines 16, 17: Dawn spike of NO3 not seen in measurements: Are these data conclusive? The spike is only short and the quoted measurements had comparatively poor temporal resolution. It should also be noted that atmospheric stability over the ocean is low at night because the atmosphere cools radiatively while the ocean surface temperature stays virtually constant (this is quite opposite to land conditions). Thus the IO precursors might simply be diluted during the night. Since much of the manuscript hinges on the absence of the IO spike these points must be discussed.

2) Page 5, first two paragraphs of section 2: Here the description is not sufficiently clear, the way this reviewer understands it is: R1 is hypothetical, but its consequences were investigated earlier. In this work R2 – R4 are studied by molecular modelling finding that only R4 might play a role. In the rest of the manuscript, therefore the effects of including R1 + R4 are studied in detail.

3) Page 8, lines 9 to 13: How can the rate constant of R4 be only uncertain by a factor of 2 when the overall exothermicity of R4 is 11 KJ/mole (page 7, line 4) while the (one sigma?) uncertainty in the overall energy is 10 KJ/mole (page 6, line 15) and thus may be as low as 1 KJ/mole?

4) Page 11, lines 10-12: The "significant increase of the sea-air flux of HOI and I2" might simply be an effect of the parameterisation of the process: Is the flux of the two species just given by the concentration difference between the two phases or is it (partly) determined by the rate of formation of the species? If the latter is the case then the flux might not change at all (or less than assumed by the model). This point needs discussing.

5) Page 12, lines 14 to 15: See comments about the "dawn spike" above.

6) Page 13, lines 8 to 11 and Fig. 6: It is not clear how the authors come to this conclusion: Fig. 6 is drawn on a semi-log scale and shows that (delta NO3/NO3) / (delta k4/k4) is about 1/300. In other words a factor of 2 change in the rate constant of R4 has negligible effect (less than 1% change) on the NO3 concentration (or mixing ratio).

7) Page 13, lines 15 to 16 and Fig. 7: It is unclear what exactly is plotted in Fig. 7. (a) is it (calculated mixing ratio without R1, R4) minus (calculated mixing ratio including R1, R4). (b) Which mixing ratio is actually shown? Is it the surface value or the vertically averaged (over which altitude range?) mixing ratio? The caption of Fig. 7 uses the term "vertical mixing ratio", which is unknown to this reviewer.

8) Page 14, line 1 and Fig. 8: See comment to Fig. 7. Also, here the text refers to "nighttime averaged differences" as opposed to 0AM to 1AM differences referred to in the explanation to Fig.7. This must be clarified.

9) Page 14, lines 9 to 11: The calculations about changes in NO3 levels are already speculative, to calculate changes in DMS (and other species) appears to be even more speculative (and not unexpected if one believes in the results regarding NO3). Therefore Fig. 9 adds little information and distracts from the main thrust of the manuscript, it should be removed.

Minor points:

Page 3, lines 15-18: This appears to be too many reference for a topic (iodine particle formation) that is not mentioned later in the manuscript.

Page 4, line 5: Organic precursors contribute 1/3?

Page 5, line 2 and following: What about BrO + DMS? The role of this reaction is neither mentioned nor discussed in the manuscript.

Page 8, lines 14 to 17: The NO3 + CH2I2 reaction can not be ruled out on the basis of the rate constant of NO3 + CH2I2 being smaller than that of R4 since the concentration

of CH2I2 may be higher than that of HOI.

Page 9, line 2: Clarify that "this new chemistry" only means the introduction of R4.

Page 11, line 7: How much better is the agreement?

Page 11 line 13: The term "uptake" means aerosol uptake?

Page 11 line 13 to page 12, line 10: The discussion of aerosol uptake appears to be out of place in the results and discussion section.

Page 12, line 11 and Fig. 5: IONO2 appears to be wrongly labelled as NO3.

―――――――――――――――――――

---

## Referee Comment (RC3) · Anonymous Referee #1 · 16 Jul 2016

Saiz-Lopez et al. investigate the nighttime chemistry of iodine. The study is very interesting and I recommend publication in ACP after considering several changes as described below.

- Title:

I find the expression "after dark" quite unusual for a scientific paper. Why not simply call it "nighttime chemistry"?

- Section 4:

Instead of presenting the full chemical mechanism, the authors refer to 6 previous publications. I find it quite tedious that I have to obtain and read 6 additional papers if I want to check the currently used mechansim. I suggest to provide the full mechansim (exactly as it was used in this study) together with this paper, e.g. in the supplement.

- Page 11, line 1:

[Figure]

It is said that "HOI peaks during the daytime". I think a better description would be to say that it peaks just before sunset. What is the reason for the sunset peak?

- Page 11, lines 13-14:

"It should be noted that during nighttime the uptake of emitted species such as I2 and HOI, and the uptake of reservoir species such as IONO2, can play a major role in the cycling of iodine."

What is meant by "uptake"? Uptake on aerosols? On clouds?

- Page 11, line 21:

The outdated JPL recommendation Sander et al. 2006 is cited here for mass accommodation coefficients. Has it been checked if there are any updates in the current recommendation JPL 2015?

- Page 12, lines 16-17:

"The IO dawn spike [...] is due to a buildup of the emitted I2 and HOI [...] over the night".

I cannot see a buildup of HOI in Fig. 4.

- Page 12, line 21:

"Reactions R1 and R4 also reduce the NO3 mixing ratio (Fig. 4, middle panels)."

Should this be Fig. 5?

- Page 13, line 17:

Please define "LT".

- Table 2: The numbers listed here are probably wave numbers, not vibrational frequencies.

- Figs. 4 and 5: A color scale should be shown. Also, it would be easier to compare

the left hand panels with the right hand panels if the same color scale was used.

- Fig. 5: For consistency, the name IONO2 should be used in the plots, not INO3.

- Fig. 7: What is a "vertical mixing ratio"?

- Figs. 7,8,9:

"without reactions (1) and (2)"

Should this be "without reactions (1) and (4)"?

---

## Author Comment (AC1) · 2 Sep 2016

We thank Dr. Louis for the insightful comments. Below we provide a detailed point-by-point answer (AC – Author Comment) to each comment on our manuscript (RC – Referee Comment).

**RC:**

**This paper presents a very interesting study on the iodine atmospheric chemistry using two modelling approaches: one at the molecular level, one at the global level. The potential energy surfaces for three different reactions were explored for the first time by theoretical calculations: HOI + NO$_2$, HOI + HNO$_3$, and HOI + NO$_3$. I have several comments concerning this work. 1) Spin-orbit correction (SOC) is very important for iodine-containing species. The authors stated on page 6 line 12 that "spin-orbit splittings of -17 and -5 kJ mol-1 were applied to energies of I and IO". These values do not correspond to the well-known value for I atom (-30.3 kJ mol-1 from C.E. Moore, Atomic Energy Levels, USGPO, Vols. II and III. NSRDS-NBS 35, Washington, DC, 1971). Over the last years, my group performed theoretical calculations to get the SOC values for numerous iodine-containing species using the CASPT2/RASSI methodology. The corresponding values for I, IO, and HOI are -30.0, -14.4, and -5.9 kJ/mol (Meciarova et al., CPL, 2011, 517, 149; Khanniche et al., JPCA, 2016, 120, 1737; Sulkova et al., JPCA, 2013, 117, 771). These calculations were also validated by comparison to few available data. I recommend the authors to update their energetics according to the correct SOC values. 2) In Table 2, there are several low frequency modes for molecular complexes and transition state. Are they among them one or several which should be treated as hindered rotors? It is also interesting to compare calculated vibrational frequencies for HOI, NO3, IO, HNO3 with their available experimental counterparts. 3) Nitrogen oxides also exhibits often an instability of the wavefunction (internal or RHF -> UHF). Is-it the case here for all stationary points? This instability will affect the energetics. The Gaussian09 software includes an option to check it. 4) What is the level of theory used for the energetics? It is not clear.**

AC:

Response to Comment 1: the sentence on page 6 should have read spin-orbit "corrections*" rather than "splittings". The sentence has now been rephrased to make this clear.

Response to Comment 2: we not treat the low frequency modes in the HOI-NO$_3$ and IO-HNO$_3$ complexes as hindered rotors. In our experience, this level of sophistication in RRKM calculations is only warranted if experimental rate coefficients are available, to which hindered rotor barrier heights can be fitted (ab initio barrier heights are generally not accurate enough).

We have added the following sentence on page 7, line 18 to make these points.

We have previously compared the calculated vibrational frequencies of iodine-containing molecules with available experimental data [Plane et al., 2006], and found this level of theory to be good enough for the purpose of this study.

Response to Comment 3: the nitrogen oxides of relevance to this study are $HNO_3$ (which does not exhibit an RHF $\rightarrow$ UHF instability at the level of theory used), and $NO_2$ which is a radical.

Response to Comment 4: the energetics are determined at the same level of theory as the geometry optimisations and vibrational frequency calculations. This is made clearer on page 6.

**RC:**

**Concerning the atmospheric modelling, the reaction of $NO_3$ with iodocarbons is important as noticed on page 8 line 15, it could be also important to see if adding the reactivity of atmospheric VOC with iodine-containing species will affect the results because this reaction will produce iodocarbons.**

AC:

As stated in the referred paragraph, there are many uncertainties about the products of these reactions, and in the case of the $NO_3+CH_2I_2$ reaction the rate constant is significantly slower than R4. Therefore we have decided not to include these reactions due to the lack of information in the bibliography.

[revised manuscript text omitted]

 H 1.484,-0.657,-0.043
 I 0.009,1.205,0.286
 N -0.456,-2.265,0.030
 O -1.052, -3.321,-0.0473
 O -1.147,-1.195,-0.228
 O 0.742,-2.161,0.333 | 55, 84, 118, 161, 196, 615, 629, 667, 705, 803, 968, 1228, 1273,1491, 3268 | 5.610, 0.916, 0.806 | -24.0 |
| IO-H-NO$_2$ TS | O 0.309,1.515,0.247
 H -0.834,1.314,-0.017
 I 1.280,-0.089,-0.093
 N -2.349,-0.133,0.019
 O -3.518, ,-0.429,-0.035
 O -1.444,-0.962,0.257
 O -2.019,1.117,-0.187 | 1249$i$, 70, 97, 103, 225, 472, 676, 698, 797, 806, 1041, 1147, 1308, 1513, 1626 | 6.300, 0.864, 0.767 | -16.4 |
| IO-HNO$_3$ complex | O 0.571,1.350,0.348
 H -1.111,1.098,-0.020
 I 1.870,0.0645,-0.152
 N -2.503,-0.202,0.0186
 O -3.673,-0.396,-0.170
 O -1.654,-0.986,0.401
 O -2.081,1.090,-0.242 | 35, 43, 76, 126, 198, 623, 677, 703, 772, 798, 939, 1331, 1416, 1713, 3281 | 7.058, 0.605, 0.566 | -34.8 |
| IO + HNO$_3$ | | 648 & 477, 585, 649, 782, 901, 1320, 1345, 1738, 3724 | 9.844 & 13.01, 12.05, 6.258 | -10.6 |

[a] Cartesian co-ordinates in Å.  [b] In cm$^{-1}$.  [c] In GHz.  [d] In kJ mol$^{-1}$, including zero-point energy and spin-orbit coupling of I and IO (see text).

[Figure]

**Figure 1**. New nocturnal iodine chemistry (in white) implemented in the THAMO and CAM-Chem models.

[Figure]

**Figure 2:** (a) Transition state for the reaction between HOI and $NO_2$ to form $HNO_3$ + I; (b) complex formed between HOI and $HNO_3$, which then reacts via transition state (c) to form $IONO2$ + $H_2O$.

[Figure]

**Figure 3**. Potential energy surface for the reaction between HOI and $NO_3$, which contains two
intermediate complexes separated by a submerged barrier.

[Figure]

**Figure 4.** THAMO modeled diurnal variation of HOI, $I_2$ (upper panels) and the HOI/$I_2$ flux from the ocean surface (bottom panel). The right hand panels are from scenario 1, which do not include night time reactions of HOI and $I_2$ with $NO_3$, while the left hand panels include the reactions in scenario 2. In bottom panel red lines represent scenario 1, while black lines correspond to scenario 2.

[Figure]

**Figure 5.** THAMO modeled diurnal variation of IO, NO$_3$ and the IONO$_2$. The right hand panels
are from scenario 1, which do not include night time reactions of HOI and I$_2$ with NO$_3$, while the
left hand panels include the reactions in scenario 2.

[Figure]

**Figure 6**. Sensitivity run showing the effect of the uncertainty in the rate constant estimation on the reduction of $NO_3$ at the surface - the red point is the theoretical estimate.

[Figure]

**Figure 7.** Modelled annual average of HOI (a) and I₂ (b) during night time at the surface level. The panels show the difference in volume mixing ratio between the simulations with and without reactions (1) and (4).

[Figure]

**Figure 8.** Modelled annual average of IONO₂ (a) and NO₃ (b) during night time at the surface level, as the difference in volume mixing ratio between the simulations with and without reactions (1) and (4).

[Figure]

**Figure 9**. Increase in the DMS levels during night time at the surface level due to the inclusion
of the reactions R1 and R4 in CAM-Chem.

[Figure]

**Figure 10**. Hourly averaged concentration of HOI, IONO$_2$ and I$_2$ in the Mediterranean Sea at the
surface level (lon:10º→20ºE, lat:33º→40ºN)

[Figure]

**Figure 11**. Hourly averaged concentration of HOI, IONO$_2$ and I$_2$ (upper panel) and NO$_3$ (bottom
panel) in the Pacific Ocean at the south of Baja California peninsula at the surface level
(lon: -110º→-106ºE, lat:16º→23ºN)

Supplementary information for

**Iodine chemistry after dark**

Alfonso Saiz-Lopez[1], John M.C. Plane[2], Carlos A. Cuevas[1], Anoop S. Mahajan[3], Jean-François Lamarque[4] and Douglas E. Kinnison[4]

[1]Department of Atmospheric Chemistry and Climate, Institute of Physical Chemistry Rocasolano, CSIC, Madrid, Spain

[2]School of Chemistry, University of Leeds, Leeds, UK

[3]Indian Institute of Tropical Meteorology, Pune, India

[4]Atmospheric Chemistry Observations and Modelling, NCAR, Colorado, USA

Correspondence to: A. Saiz-Lopez (a.saiz@csic.es)

Table 1. Iodine chemistry scheme in CAM-Chem: Bimolecular, thermal decomposition and termolecular reactions.

| Reaction | k / cm$^3$ molecule$^{-1}$ s$^{-1}$ | Notes |
|---|---|---|
| $I + O_3 \rightarrow IO + O_2$ | $2.1 \times 10^{-11}$ e$^{(-830/T)}$ | 1 |
| $IO + O_3 \rightarrow OIO + O_2$ | $3.6 \times 10^{-16}$ | 2 |
| $I + HO_2 \rightarrow HI + O_2$ | $1.5 \times 10^{-11}$ e$^{(-1090/T)}$ | 3 |
| $IO + NO \rightarrow I + NO_2$ | $7.15 \times 10^{-12}$ e$^{(300/T)}$ | 1 |
| $IO + HO_2 \rightarrow HOI + O_2$ | $1.4 \times 10^{-11}$ e$^{(540/T)}$ | 1 |
| $IO + IO \rightarrow OIO + I$ | $2.13 \times 10^{-11}$ e$^{(180/T)} \times$ [$1 + $e$^{(-p/191.42)}$] | 1, 4 |
| $IO + IO \rightarrow I_2O_2$ | $3.27 \times 10^{-11}$ e$^{(180/T)} \times$ [$1 - 0.65$ e$^{(-p/191.42)}$] | 1, 4 |
| $IO + OIO \rightarrow I_2O_3$ | $w_1 \cdot$ exp $(w_2 \cdot T)$ [a] | 4, 5, 6 [g] |
| $OIO + OIO \rightarrow I_2O_4$ | $w_1 \cdot$ exp $(w_2 \cdot T)$ [b] | 4, 5, 6 [g] |
| $I_2 + O \rightarrow IO + I$ | $1.25 \times 10^{-10}$ | 1 |
| $IO + O \rightarrow I + O_2$ | $1.4 \times 10^{-10}$ | 1 |
| $IO + OH \rightarrow HO_2 + I$ | $1.0 \times 10^{-10}$ | 7 |
| $I_2O_2 \rightarrow OIO + I$ | $w_1 \cdot$ exp $(w_2/T)$ [c] | 5, 6, 8 [g] |
| $I_2O_2 \rightarrow IO + IO$ | $w_1 \cdot$ exp $(w_2/T)$ [d] | 5, 6, 8 [g] |
| $I_2O_4 \rightarrow 2\ OIO$ | $w_1 \cdot$ exp $(w_2/T)$ [e] | 5, 8 [g] |
| $I_2 + OH \rightarrow HOI + I$ | $1.8 \times 10^{-10}$ | 3 |
| $I_2 + NO_3 \rightarrow I + IONO_2$ | $1.5 \times 10^{-12}$ | 9 |
| $I + NO_3 \rightarrow IO + NO_2$ | $1.0 \times 10^{-10}$ | 1 |
| $OH + HI \rightarrow I + H_2O$ | $1.6 \times 10^{-11}$ e$^{(440/T)}$ | 1 |
| $I + IONO_2 \rightarrow I_2 + NO_3$ | $9.1 \times 10^{-11}$ e$^{(-146/T)}$ | 5 |
| $HOI + OH \rightarrow IO + H_2O$ | $2.0 \times 10^{-13}$ | 10 |
| $IO + DMS \rightarrow DMSO + I$ | $3.2 \times 10^{-13}$ e$^{(-925/T)}$ | 11 |
| $INO_2 \rightarrow I + NO_2$ | $1008 \times 10^{15}$ e$^{(-13670/T)}$ | 12, 13, 14 |
| $IONO_2 \rightarrow IO + NO_2$ | $w_1 \cdot$ exp $(w_2/T)$ [f] | 5, 15 |
| $INO + INO \rightarrow I_2 + 2NO$ | $8.4 \times 10^{-11}$ e$^{(-2620/T)}$ | 3 |
| $INO_2 + INO_2 \rightarrow I_2 + 2NO_2$ | $4.7 \times 10^{-13}$ e$^{(-1670/T)}$ | 1 |
| $OIO + NO \rightarrow IO + NO_2$ | $1.1 \times 10^{-12}$ e$^{(542/T)}$ | 14 |
| $HI + NO_3 \rightarrow I + HNO_3$ | $1.3 \times 10^{-12}$ e$^{(-1830/T)}$ | 16 |
| $IO + BrO \rightarrow Br + I + O_2$ | $0.30 \times 10^{-11}$ e$^{(510/T)}$ | 1 |
| $IO + BrO \rightarrow Br + OIO$ | $1.20 \times 10^{-11}$ e$^{(510/T)}$ | 1 |
| $I + BrO \rightarrow IO + Br$ | $1.44 \times 10^{-11}$ | 17, 18, 19 |

| Reaction | Rate | Ref |
|---|---|---|
| $IO + ClO \rightarrow I + OClO$ | $2.585 \times 10^{-12}\ e^{(280/T)}$ | 1 |
| $IO + ClO \rightarrow I + Cl + O_2$ | $1.175 \times 10^{-12}\ e^{(280/T)}$ | 1 |
| $IO + ClO \rightarrow ICl + O_2$ | $0.940 \times 10^{-12}\ e^{(280/T)}$ | 1 |
| $IO + Br \rightarrow I + BrO$ | $2.49 \times 10^{-11}$ | 18, 19 |
| $IO + NO_3 \rightarrow OIO + NO_2$ | $9.0 \times 10^{-12}$ | 20 |
| $IO + CH_3O_2 \rightarrow CH_2O + I + HO_2$ | $2.0 \times 10^{-12}$ | 2[h] |
| $CH_3I + OH \rightarrow I + H_2O + HO_2$ | $2.90 \times 10^{-12}\ e^{(-1100/T)}$ | 3 |
| $I + NO_2\ (+ M) \rightarrow INO_2\ (+ M)$ | $k_0 = 3 \times 10^{-31} \times (T/300)^{-1}$
 $k_\infty = 6.6 \times 10^{-11}$ | 3[i] |
| $IO + NO_2\ (+ M) \rightarrow IONO_2\ (+ M)$ | $k_0 = 6.5 \times 10^{-31} \times (T/300)^{-3.5}$
 $k_\infty = 7.6 \times 10^{-12} \times (T/300)^{-1.5}$ | 3[i] |
| $I + NO\ (+ M) \rightarrow INO\ (+ M)$ | $k_0 = 1.8 \times 10^{-32} \times (T/300)^{-1}$
 $k_\infty = 1.7 \times 10^{-11}$ | 3[i] |
| $OIO + OH\ (+ M) \rightarrow HOIO_2\ (+ M)$ | $k_0 = 1.5 \times 10^{-27} \times (T/300)^{-3.93}$
 $k_\infty = 7.76 \times 10^{-10} \times (T/300)^{-0.8}$ | 14[j] |
| $HOI + NO_3 \rightarrow IO + HNO_3$ | $2.7 \times 10^{-12}\ (300/T)^{2.66}$ | 21 |

[1] IUPAC-2008 (Atkinson et al., 2007) ; [2](Dillon et al., 2006b); [3] JPL-2010 (Sander et al., 2011); [4](Gómez Martín et al., 2007); [5](Kaltsoyannis and Plane, 2008); [6](Galvez et al., 2013); [7](Bösch et al., 2003); [8] (Gómez Martín and Plane, 2009); [9](Chambers et al., 1992); [10](Chameides and Davis, 1980); [11](Dillon et al., 2006a); [12](McFiggans et al., 2000); [13](Jenkin et al., 1985); [14](Plane et al., 2006); [15](Allan and Plane, 2002); [16] (Lancar et al., 1991); [17](Laszlo et al., 1997); [18](Bedjanian et al., 1997); [19](Gilles et al., 1997); [20](Dillon et al., 2008); [21]This work.

[a]  $w1 = 4.687 \times 10^{-10} - 1.3855 \times 10^{-5} \times e^{(-0.75\,p\,/\,1.62265)} + 5.51868 \times 10^{-10} \times e^{(-0.75\,p\,/\,199.328)}$

  $w2 = -0.00331 - 0.00514 \times e^{(-0.75\,p\,/\,325.68711)} - 0.00444 \times e^{(-0.75\,p\,/\,40.81609)}$

[b]  $w1 = 1.1659 \times 10^{-9} - 7.79644 \times 10^{-10}\ e^{(-0.75\,p/22.09281)} + 1.03779 \times 10^{-9} \times e^{(-0.75\,p\,/\,568.15381)}$

  $w2 = -0.00813 - 0.00382 \times e^{(-0.75\,p\,/\,45.57591)} - 0.00643 \times e^{(-0.75\,p\,/\,417.95061)}$

[c]  $w1 = 3.54288 \times 10^{10} + 1.8523 \times 10^{11} \times 0.75\,p - 1.45435 \times 10^{8} \times (0.75p)^{2} + 60799.4344 \times (0.75p)^{3}$
  $w2 = -9681.65989 + 346.95538 \times e^{(-0.75\,p\,/\,343.25322)} + 251.78032 \times e^{(-0.75\,p\,/\,44.1466)}$

[d]  $w1 = 255335000000 - 4418880000 \times 0.75\,p + 85618600 \times (0.75\,p)^{2} + 14218.81 \times (0.75\,p)^{3}$
  $w2 = -11466.82304 + 597.01334 \times e^{(-0.75\,p\,/\,1382.62325)} - 167.3391 \times e^{(-0.75\,p\,/\,43.75089)}$

[e]  $w1 = -1.92626 \times 10^{14} + 4.67414 \times 10^{13} \times 0.75\,p - 3.68651 \times 10^{8} \times (0.75\,p)^{2} - 3.09109 \times 10^{6} \times (0.75\,p)^{3}$
  $w2 = -12302.15294 + 252.78367 \times e^{(-0.75\,p\,/\,46.12733)} + 437.62868 \times e^{(-0.75\,p\,/\,428.4413)}$

$f$ $\quad$ w1 = -2.63544 x $10^{13}$ + 4.32845 x $10^{12}$ x (0.75 p) + 3.73758 x $10^8$ x (0.75 p) $^2$ - 628468.76313 x (0.75 p) $^3$

$\quad$ w2 = -13847.85015 + 240.34465 x e $^{(-0.75\,p\,/\,49.27141)}$ + 451.35864 x e $^{(-0.75\,p\,/\,436.87605)}$

$g$ $\quad$ The empirical expressions of the form $w_1 \cdot \exp(w_2 \cdot T)$ were obtained by non-linear least squares fitting of *Rice–Ramsperger–Kassel–Marcus* (RRKM) theoretical results for the indicated reaction rate constants and thermal dissociation rates in the (27 – 1013) hPa pressure range. RRKM calculations were carried out using the MESMER algorithm (Glowacki et al., 2012) as indicated in the corresponding references (e.g. (Galvez et al., 2013). Expression $a$ produces negative values outside the range of modelled rate constants (p < 20 hPa), and therefore a fixed rate constant of 3 x $10^{-11}$ $cm^3$ $molecule^{-1}$ $s^{-1}$ was assumed. Expressions $e$ and $f$ generate negligible dissociation rates below ~500 hPa which become negative at ~8 hPa – in this case they are set to zero below that pressure.

$h$ $\quad$ Updated heats of formation for IO, OIO, and $CH_3O_2$ (Dooley et al., 2008; Gómez Martín and Plane, 2009; Knyazev and Slagle, 1998) show that the only accessible exothermic product channel of $CH_3O_2$ + IO (Drougas and Kosmas, 2007) is $CH_2O$ + I + $O_2$ ($\Delta H_r$ = -5 ± 6 kJ $mol^{-1}$), consistent with the high yield of I and low yield of OIO found experimentally (Bale et al., 2005; Enami et al., 2006). Sensitivity studies have been carried out (Saiz-Lopez et al., 2014) using the preferred rate constant for this reaction of 2 × $10^{-12}$ $cm^3$ $molecule^{-1}$ $s^{-1}$ (Dillon et al., 2006b), resulting in an enhancement of the ozone loss of 0.5% in the MBL and of less than 0.1% integrated throughout the troposphere in the $J_{IxOy}$ scenario, and similarly negligible enhancements in the Base scenario. Impacts in the $I_y$ partitioning are also very minor.

$i$ $\quad$ The temperature and pressure dependent rate constant (k) is computed based on the low pressure ($k_0$) and the high-pressure ($k_\infty$) rate coefficients following JPL-2010 (Sander et al., 2011).

$j$ $\quad$ The Fast rate constants and a thermally stable product $HOIO_2$ have been predicted theoretically (Plane et al., 2006), but no experimental studies reporting observation of $HOIO_2$ and its photochemical properties in the gas phase are available. Since the level of uncertainty is even larger than for the $I_xO_y$, it has not been included in the mechanism.

Table 2. Iodine chemistry scheme in CAM-Chem: Photochemical reactions.

| Reaction |
| --- |
| $CH_3I + h\nu \rightarrow CH_3O_2 + I$ |
| $CH_2I_2 + h\nu \rightarrow 2I$ [a] |
| $CH_2IBr + h\nu \rightarrow Br + I$ [a] |
| $CH_2ICl + h\nu \rightarrow Cl + I$ [a] |
| $I_2 + h\nu \rightarrow 2I$ |
| $IO + h\nu \rightarrow I + O$ |
| $OIO + h\nu \rightarrow I + O_2$ |
| $INO + h\nu \rightarrow I + NO$ |
| $INO_2 + h\nu \rightarrow I + NO_2$ [b] |
| $IONO_2 + h\nu \rightarrow I + NO_3$ |
| $HOI + h\nu \rightarrow I + OH$ |
| $IBr + h\nu \rightarrow I + Br$ |
| $ICl + h\nu \rightarrow I + Cl$ |
| $I_2O_2 + h\nu \rightarrow I + OIO$ [c] |
| $I_2O_3 + h\nu \rightarrow IO + OIO$ [c] |
| $I_2O_4 + h\nu \rightarrow OIO + OIO$ [c] |

Photolysis rates are computed online considering the actinic flux calculation in CAM-Chem. The absorption cross-sections and quantum yields for all species besides the $I_xO_y$ have been taken from IUPAC-2008 (Atkinson et al., 2007; Atkinson et al., 2008) and JPL-2010 (Sander et al., 2011).

[a] radical organic products are not considered.
[b] only the reaction channel reported in JPL 06-02 (Sander et al., 2006) is considered.
[c] photolysis reactions only considered in the $J_{IxOy}$ scheme (Saiz-Lopez et al., 2014).

Table 3. Iodine chemistry scheme in CAM-Chem: Heterogeneous reactions.

| Sea-salt aerosol reactions | Reactive uptake |
|---|---|
| $IONO_2 \rightarrow 0.5\ IBr + 0.5\ ICl$ | $\gamma = 0.01$ |
| $INO_2 \rightarrow 0.5\ IBr + 0.5\ ICl$ | $\gamma = 0.02$ |
| $HOI \rightarrow 0.5\ IBr + 0.5\ ICl$ | $\gamma = 0.06$ |
| $I_2O_2 \rightarrow$ | $\gamma = 0.01$[§] |
| $I_2O_3 \rightarrow$ | $\gamma = 0.01$[§] |
| $I_2O_4 \rightarrow$ | $\gamma = 0.01$[§] |

Values based on the THAMO model (Saiz-Lopez et al., 2008) and implemented in CAM-Chem following (Ordóñez et al., 2012).

[§] Deposition of $I_xO_y$ species on sea-salt aerosols has been included following the free regime approximation.

Table 4. Iodine chemistry scheme in CAM-Chem: Henry's Law constants and dry deposition velocities.

| Species | $k_0$ (M atm$^{-1}$) | Deposition velocity[§] (cm s$^{-1}$) | Reference |
|---|---|---|---|
| IBr [ice] | $2.4 \times 10^1$ | – | 1 |
| ICl [ice] | $1.1 \times 10^2$ | – | 1 |
| HI | $7.8 \times 10^{-1}$ | 1.0 | 1 [a] |
| HOI – ($J_{IxOy}$ / Base) | $1.9 \times 10^3$ / $4.5 \times 10^3$ | 0.75 | 1 [b] |
| IONO$_2$ [ice] | $1.0 \times 10^6$ | 0.75 | 2 [c] |
| INO$_2$ [ice] | $3.0 \times 10^{-1}$ | 0.75 | 1 [d] |
| IO | $4.5 \times 10^2$ | – | 2 |
| OIO | $1.0 \times 10^4$ | – | 2 |
| I$_2$O$_2$ | $1.0 \times 10^4$ | 1.0 | 2 |
| I$_2$O$_3$ | $1.0 \times 10^4$ | 1.0 | 2 |
| I$_2$O$_4$ | $1.0 \times 10^4$ | 1.0 | 2 |

[§] Dry deposition velocities are based on the THAMO model (Saiz-Lopez et al., 2008).
[1] Values reported in (Sander, 1999).
[2] Values based on the THAMO model (Saiz-Lopez et al., 2008).
[a] Considering a dissociation constant $K_a = 3.2 \times 10^9$ and a temperature dependent coefficient c = 9800 K
[b] Within the range of values given in the corresponding reference.
[c] Virtually infinite solubility is represented by using a very large arbitrary number.
[d] Value assumed to be equal to those of BrNO$_2$.
[ice] Species for which ice-uptake is considered following (Neu and Prather, 2012).

---

## Author Comment (AC2) · 2 Sep 2016

We would like to thank Howard Roscoe for his comments and support. Below we provide a detailed point-by-point answer (AC – Author Comment) to each comment on our manuscript (RC – Referee Comment).

**RC:**

**This paper makes an important point about atmospheric chemistry. It is scientifically sound and rigorous except for the few items in Minor Comments, and except perhaps for the theoretical calculations in Section 3 on which I am not competent to express an opinion. It is also well written, except for the trivia listed under Editorial Comments below.**

**Provided it receives a satisfactory review from experts in calculation of reaction rates, I have no hesitation in recommending it for publication in ACP after minor revision.**

**Minor comments:**

**RC:**

**1. p14 line22 - according to the caption of Figure 11 (p39) it applies to the region 110 to 106degE and 16 to 23degN. This region just touches the southern tip of Baja California but is centred a long way to its south. It just touches a coastal region of mainland Mexico, but is never at the "coastal region" even of Mexico let alone the stated Baja California - much of the region is in what might be called the open ocean. Presumably this region is chosen because of the large pollution amounts there that we infer from Figures 7, 8 and 9; but there is no discussion of why they should be so large - is it a concentration of shipping using the Panama Canal that spreads out further north?**

AC:

The text and the caption have been changed to refer to an Ocean Pacific Region at the south of Baja California, instead of a "coastal region". As pointed out by the reviewer, that zone was selected due to the high levels of $NO_3$ caused by pollution outflow from the west coasts of Mexico and USA and shipping lanes.

**RC:**

**2. p6 line21 - given the argument of p6 lines17-18, why does a transition state 110 kN/mole above the reactants allow the reaction to proceed?**

AC:

Please note that in that paragraph at the end of page 6 we are not saying that reaction 3 is viable. In fact we conclude at the end of the theoretical calculation section that only reactions 1 and 4 are likely to proceed.

**RC:**

**3. Why do Figures 4 and 5 have time co-ordinates starting at 48 hours? Is this to allow a steady state to build up? - if so it should be discussed. And what version of time is it - time since midnight or time since noon? - a careful reading of text and figures tells us which, but it should be spelled out in the caption. And why do Figures 10 and 11 have time co-ordinates that start at 0 hours rather than 48? And although we can guess that time in Figures 10 and 11 is since midnight, is it mean solar midnight over the region, or solar midnight at the geographic centre of the region, or midnight in the local time zone at 108degE?**

AC

Figure 4 and 5 represent data from the 1-D THAMO model, whereas figures 10 and 11 contain data from the 3-D CAM-Chem model. In a 1D model like THAMO 2 days of simulation are enough to reach steady conditions, so we have plotted the third day of simulation. On the other hand the 3D global model CAM-Chem needs at least 1 year to reach steady conditions throughout the marine troposphere. So we have run CAM-Chem for two years and then used the data from the second year. Figures 10 and 11 contain hourly averaged data during August and January respectively. Therefore we have used a more general local time 0-23 hours scale for Figure 10 and 11.

**RC:**

**4. We are told in the text (p14 line2) that Figure 8 has "as in the previous figure".. "nighttime averaged differences", yet p13 line17 tells us that the previous figure, Figure 7, uses "midnight averages". Which are used in which figures, and why do the captions not spell out the averaging hours as opposed to having them buried in the text?**

AC:

Both Figure 7 and 8 correspond to night time averages from 0LT to 01LT. We have changed the text in p13 line17 to "nighttime averaged". We decided not to include that data in the caption to avoid repetition of information.

**Editorial comments:**

**RC:**

**p3 line3 introduces and defines MBL but it was already used without definition on p2 line15.**

AC:

MBL defined for the first time in P2 line 15.

**RC:**

**p3 line10 - surely, hyphens after "iodine" and "bromine" ?**

AC:

Typo corrected

**RC:**

**p4 line2 - delete "and"**

AC:

Typo corrected.

**RC:**

**p6 line12 - insert "of" after "energies".**

AC:

Typo corrected

**RC:**

**p7 line13 - insert "the" after "of".**

AC:

Typo corrected

**RC:**

**Fig4 lowest panel - the meanings of the four lines are not in the caption and their panel legends are obscure.**

AC:

More information has been included in the caption to make the bottom panel clearer.

**RC:**

**Figs 7, 8 and 9 - the captions do not say the altitude or the vertical extent of the averaging.**

AC:

Included "at the surface level" at the end of the caption.

**RC:**

**Fig10 - the right hand axis legend says "mixing ration".**

AC:

Typo corrected.

**RC:**

**Fig10 caption - insert "the" after "at".**

AC:

Typo Corrected.

**RC:**

**Fig11 caption - say the altitude.**

AC:

Included "at the surface level" at the end of the caption.

[revised manuscript text omitted]

H 1.484,-0.657,-0.043
I 0.009,1.205,0.286
N -0.456,-2.265,0.030
O -1.052, -3.321,-0.0473
O -1.147,-1.195,-0.228
O 0.742,-2.161,0.333 | 55, 84, 118, 161, 196, 615, 629, 667, 705, 803, 968, 1228, 1273,1491, 3268 | 5.610, 0.916, 0.806 | -24.0 |
| IO-H-NO$_2$ TS | O 0.309,1.515,0.247
H -0.834,1.314,-0.017
I 1.280,-0.089,-0.093
N -2.349,-0.133,0.019
O -3.518, ,-0.429,-0.035
O -1.444,-0.962,0.257
O -2.019,1.117,-0.187 | 1249$i$, 70, 97, 103, 225, 472, 676, 698, 797, 806, 1041, 1147, 1308, 1513, 1626 | 6.300, 0.864, 0.767 | -16.4 |
| IO-HNO$_3$ complex | O 0.571,1.350,0.348
H -1.111,1.098,-0.020
I 1.870,0.0645,-0.152
N -2.503,-0.202,0.0186
O -3.673,-0.396,-0.170
O -1.654,-0.986,0.401
O -2.081,1.090,-0.242 | 35, 43, 76, 126, 198, 623, 677, 703, 772, 798, 939, 1331, 1416, 1713, 3281 | 7.058, 0.605, 0.566 | -34.8 |
| IO + HNO$_3$ | | 648 & 477, 585, 649, 782, 901, 1320, 1345, 1738, 3724 | 9.844 & 13.01, 12.05, 6.258 | -10.6 |

[a] Cartesian co-ordinates in Å. [b] In cm$^{-1}$. [c] In GHz. [d] In kJ mol$^{-1}$, including zero-point energy and spin-
orbit coupling of I and IO (see text).

[Figure]

**Figure 1**. New nocturnal iodine chemistry (in white) implemented in the THAMO and CAM-Chem models.

[Figure]

**Figure 2:** (a)  Transition state for the reaction between HOI and $NO_2$ to form $HNO_3$ + I; (b)
complex formed between HOI and $HNO_3$, which then reacts via transition state (c) to form
IONO2 + $H_2O$.

[Figure]

**Figure 3**. Potential energy surface for the reaction between HOI and $NO_3$, which contains two
intermediate complexes separated by a submerged barrier.

[Figure]

**Figure 4.** THAMO modeled diurnal variation of HOI, $I_2$ (upper panels) and the HOI/$I_2$ flux from the ocean surface (bottom panel). The right hand panels are from scenario 1, which do not include night time reactions of HOI and $I_2$ with $NO_3$, while the left hand panels include the reactions in scenario 2. In bottom panel red lines represent scenario 1, while black lines correspond to scenario 2.

[Figure]

**Figure 5.** THAMO modeled diurnal variation of IO, NO$_3$ and the IONO$_2$. The right hand panels
are from scenario 1, which do not include night time reactions of HOI and I$_2$ with NO$_3$, while the
left hand panels include the reactions in scenario 2.

[Figure]

**Figure 6**. Sensitivity run showing the effect of the uncertainty in the rate constant estimation on the reduction of $NO_3$ at the surface - the red point is the theoretical estimate.

[Figure]

**Figure 7.** Modelled annual average of HOI (a) and I$_2$ (b) during night time at the surface level. The panels show the difference in volume mixing ratio between the simulations with and without reactions (1) and (4).

[Figure]

**Figure 8.** Modelled annual average of IONO₂ (a) and NO₃ (b) during night time at the surface level, as the difference in volume mixing ratio between the simulations with and without reactions (1) and (4).

[Figure]

Figure 9. Increase in the DMS levels during night time at the surface level due to the inclusion of the reactions R1 and R4 in CAM-Chem.

[Figure]

**Figure 10**. Hourly averaged concentration of HOI, IONO$_2$ and I$_2$ in the Mediterranean Sea at the
surface level (lon:10º→20ºE, lat:33º→40ºN)

[Figure]

**Figure 11**. Hourly averaged concentration of HOI, IONO$_2$ and I$_2$ (upper panel) and NO$_3$ (bottom
panel) in the Pacific Ocean at the south of Baja California peninsula at the surface level
(lon: -110º→-106ºE, lat:16º→23ºN)

Supplementary information for

**Iodine chemistry after dark**

Alfonso Saiz-Lopez[1], John M.C. Plane[2], Carlos A. Cuevas[1], Anoop S. Mahajan[3], Jean-François Lamarque[4] and Douglas E. Kinnison[4]

[1]Department of Atmospheric Chemistry and Climate, Institute of Physical Chemistry Rocasolano, CSIC, Madrid, Spain

[2]School of Chemistry, University of Leeds, Leeds, UK

[3]Indian Institute of Tropical Meteorology, Pune, India

[4]Atmospheric Chemistry Observations and Modelling, NCAR, Colorado, USA

Correspondence to: A. Saiz-Lopez (a.saiz@csic.es)

Table 1. Iodine chemistry scheme in CAM-Chem: Bimolecular, thermal decomposition and termolecular reactions.

| Reaction | $k$ / cm$^3$ molecule$^{-1}$ s$^{-1}$ | Notes |
|---|---|---|
| $I + O_3 \rightarrow IO + O_2$ | $2.1 \times 10^{-11}\ e^{(-830\,/\,T)}$ | 1 |
| $IO + O_3 \rightarrow OIO + O_2$ | $3.6 \times 10^{-16}$ | 2 |
| $I + HO_2 \rightarrow HI + O_2$ | $1.5 \times 10^{-11}\ e^{(-1090\,/\,T)}$ | 3 |
| $IO + NO \rightarrow I + NO_2$ | $7.15 \times 10^{-12}\ e^{(300\,/\,T)}$ | 1 |
| $IO + HO_2 \rightarrow HOI + O_2$ | $1.4 \times 10^{-11}\ e^{(540\,/\,T)}$ | 1 |
| $IO + IO \rightarrow OIO + I$ | $2.13 \times 10^{-11}\ e^{(180/\,T)} \times [1 + e^{(-p\,/\,191.42)}]$ | 1, 4 |
| $IO + IO \rightarrow I_2O_2$ | $3.27 \times 10^{-11}\ e^{(180/\,T)} \times [1 - 0.65\ e^{(-p\,/\,191.42)}]$ | 1, 4 |
| $IO + OIO \rightarrow I_2O_3$ | $w_1 \cdot \exp(w_2 \cdot T)$ [a] | 4, 5, 6 [g] |
| $OIO + OIO \rightarrow I_2O_4$ | $w_1 \cdot \exp(w_2 \cdot T)$ [b] | 4, 5, 6 [g] |
| $I_2 + O \rightarrow IO + I$ | $1.25 \times 10^{-10}$ | 1 |
| $IO + O \rightarrow I + O_2$ | $1.4 \times 10^{-10}$ | 1 |
| $IO + OH \rightarrow HO_2 + I$ | $1.0 \times 10^{-10}$ | 7 |
| $I_2O_2 \rightarrow OIO + I$ | $w_1 \cdot \exp(w_2\,/\,T)$ [c] | 5, 6, 8 [g] |
| $I_2O_2 \rightarrow IO + IO$ | $w_1 \cdot \exp(w_2\,/\,T)$ [d] | 5, 6, 8 [g] |
| $I_2O_4 \rightarrow 2\ OIO$ | $w_1 \cdot \exp(w_2\,/\,T)$ [e] | 5, 8 [g] |
| $I_2 + OH \rightarrow HOI + I$ | $1.8 \times 10^{-10}$ | 3 |
| $I_2 + NO_3 \rightarrow I + IONO_2$ | $1.5 \times 10^{-12}$ | 9 |
| $I + NO_3 \rightarrow IO + NO_2$ | $1.0 \times 10^{-10}$ | 1 |
| $OH + HI \rightarrow I + H_2O$ | $1.6 \times 10^{-11}\ e^{(440\,/\,T)}$ | 1 |
| $I + IONO_2 \rightarrow I_2 + NO_3$ | $9.1 \times 10^{-11}\ e^{(-146\,/\,T)}$ | 5 |
| $HOI + OH \rightarrow IO + H_2O$ | $2.0 \times 10^{-13}$ | 10 |
| $IO + DMS \rightarrow DMSO + I$ | $3.2 \times 10^{-13}\ e^{(-925\,/\,T)}$ | 11 |
| $INO_2 \rightarrow I + NO_2$ | $1008 \times 10^{15}\ e^{(-13670\,/\,T)}$ | 12, 13, 14 |
| $IONO_2 \rightarrow IO + NO_2$ | $w_1 \cdot \exp(w_2\,/\,T)$ [f] | 5, 15 |
| $INO + INO \rightarrow I_2 + 2NO$ | $8.4 \times 10^{-11}\ e^{(-2620/T)}$ | 3 |
| $INO_2 + INO_2 \rightarrow I_2 + 2NO_2$ | $4.7 \times 10^{-13}\ e^{(-1670\,/\,T)}$ | 1 |
| $OIO + NO \rightarrow IO + NO_2$ | $1.1 \times 10^{-12}\ e^{(542\,/\,T)}$ | 14 |
| $HI + NO_3 \rightarrow I + HNO_3$ | $1.3 \times 10^{-12}\ e^{(-1830\,/\,T)}$ | 16 |
| $IO + BrO \rightarrow Br + I + O_2$ | $0.30 \times 10^{-11}\ e^{(510/T)}$ | 1 |
| $IO + BrO \rightarrow Br + OIO$ | $1.20 \times 10^{-11}\ e^{(510/T)}$ | 1 |
| $I + BrO \rightarrow IO + Br$ | $1.44 \times 10^{-11}$ | 17, 18, 19 |

| | | |
|---|---|---|
| $IO + ClO \rightarrow I + OClO$ | $2.585 \times 10^{-12}\, e^{(280/T)}$ | 1 |
| $IO + ClO \rightarrow I + Cl + O_2$ | $1.175 \times 10^{-12}\, e^{(280/T)}$ | 1 |
| $IO + ClO \rightarrow ICl + O_2$ | $0.940 \times 10^{-12}\, e^{(280/T)}$ | 1 |
| $IO + Br \rightarrow I + BrO$ | $2.49 \times 10^{-11}$ | 18, 19 |
| $IO + NO_3 \rightarrow OIO + NO_2$ | $9.0 \times 10^{-12}$ | 20 |
| $IO + CH_3O_2 \rightarrow CH_2O + I + HO_2$ | $2.0 \times 10^{-12}$ | 2[h] |
| $CH_3I + OH \rightarrow I + H_2O + HO_2$ | $2.90 \times 10^{-12}\, e^{(-1100/T)}$ | 3 |
| $I + NO_2\,(+ M) \rightarrow INO_2\,(+ M)$ | $k_0 = 3 \times 10^{-31} \times (T/300)^{-1}$
 $k_\infty = 6.6 \times 10^{-11}$ | 3[i] |
| $IO + NO_2\,(+ M) \rightarrow IONO_2\,(+ M)$ | $k_0 = 6.5 \times 10^{-31} \times (T/300)^{-3.5}$
 $k_\infty = 7.6 \times 10^{-12} \times (T/300)^{-1.5}$ | 3[i] |
| $I + NO\,(+ M) \rightarrow INO\,(+ M)$ | $k_0 = 1.8 \times 10^{-32} \times (T/300)^{-1}$
 $k_\infty = 1.7 \times 10^{-11}$ | 3[i] |
| $OIO + OH\,(+ M) \rightarrow HOIO_2\,(+ M)$ | $k_0 = 1.5 \times 10^{-27} \times (T/300)^{-3.93}$
 $k_\infty = 7.76 \times 10^{-10} \times (T/300)^{-0.8}$ | 14[j] |
| $HOI + NO_3 \rightarrow IO + HNO_3$ | $2.7 \times 10^{-12}\, (300/T)^{2.66}$ | 21 |

[1] IUPAC-2008 (Atkinson et al., 2007) ; [2](Dillon et al., 2006b); [3] JPL-2010 (Sander et al., 2011); [4](Gómez Martín et al., 2007); [5](Kaltsoyannis and Plane, 2008); [6](Galvez et al., 2013); [7](Bösch et al., 2003); [8] (Gómez Martín and Plane, 2009); [9](Chambers et al., 1992); [10](Chameides and Davis, 1980); [11](Dillon et al., 2006a); [12](McFiggans et al., 2000); [13](Jenkin et al., 1985); [14](Plane et al., 2006); [15](Allan and Plane, 2002); [16] (Lancar et al., 1991); [17](Laszlo et al., 1997); [18](Bedjanian et al., 1997); [19](Gilles et al., 1997); [20](Dillon et al., 2008); [21]This work.

[a] $w1 = 4.687 \times 10^{-10} - 1.3855 \times 10^{-5} \times e^{(-0.75\,p\,/\,1.62265)} + 5.51868 \times 10^{-10} \times e^{(-0.75\,p\,/\,199.328)}$

$w2 = -0.00331 - 0.00514 \times e^{(-0.75\,p\,/\,325.68711)} - 0.00444 \times e^{(-0.75\,p\,/\,40.81609)}$

[b] $w1 = 1.1659 \times 10^{-9} - 7.79644 \times 10^{-10}\, e^{(-0.75\,p/22.09281)} + 1.03779 \times 10^{-9} \times e^{(-0.75\,p\,/\,568.15381)}$

$w2 = -0.00813 - 0.00382 \times e^{(-0.75\,p\,/\,45.57591)} - 0.00643 \times e^{(-0.75\,p\,/\,417.95061)}$

[c] $w1 = 3.54288 \times 10^{10} + 1.8523 \times 10^{11} \times 0.75\,p - 1.45435 \times 10^{8} \times (0.75p)^2 + 60799.4344 \times (0.75p)^3$

$w2 = -9681.65989 + 346.95538 \times e^{(-0.75\,p\,/\,343.25322)} + 251.78032 \times e^{(-0.75\,p\,/\,44.1466)}$

[d] $w1 = 255335000000 - 4418880000 \times 0.75\,p + 85618600 \times (0.75\,p)^2 + 14218.81 \times (0.75\,p)^3$

$w2 = -11466.82304 + 597.01334 \times e^{(-0.75\,p\,/\,1382.62325)} - 167.3391 \times e^{(-0.75\,p\,/\,43.75089)}$

[e] $w1 = -1.92626 \times 10^{14} + 4.67414 \times 10^{13} \times 0.75\,p - 3.68651 \times 10^{8} \times (0.75\,p)^2 - 3.09109 \times 10^{6} \times (0.75\,p)^3$

$w2 = -12302.15294 + 252.78367 \times e^{(-0.75\,p\,/\,46.12733)} + 437.62868 \times e^{(-0.75\,p\,/\,428.4413)}$

[f]     $w1 = -2.63544 \times 10^{13} + 4.32845 \times 10^{12} \times (0.75\ p) + 3.73758 \times 10^{8} \times (0.75\ p)^{2} - 628468.76313 \times (0.75\ p)^{3}$

$w2 = -13847.85015 + 240.34465 \times e^{(-0.75\ p\ /\ 49.27141)} + 451.35864 \times e^{(-0.75\ p\ /\ 436.87605)}$

[g]     The empirical expressions of the form $w_1 \cdot \exp(w_2 \cdot T)$ were obtained by non-linear least squares fitting of *Rice–Ramsperger–Kassel–Marcus* (RRKM) theoretical results for the indicated reaction rate constants and thermal dissociation rates in the (27 – 1013) hPa pressure range. RRKM calculations were carried out using the MESMER algorithm (Glowacki et al., 2012) as indicated in the corresponding references (e.g. (Galvez et al., 2013). Expression [a] produces negative values outside the range of modelled rate constants (p < 20 hPa), and therefore a fixed rate constant of $3 \times 10^{-11}$ cm$^3$ molecule$^{-1}$ s$^{-1}$ was assumed. Expressions [e] and [f] generate negligible dissociation rates below ~500 hPa which become negative at ~8 hPa – in this case they are set to zero below that pressure.

[h]     Updated heats of formation for IO, OIO, and $CH_3O_2$ (Dooley et al., 2008; Gómez Martín and Plane, 2009; Knyazev and Slagle, 1998) show that the only accessible exothermic product channel of $CH_3O_2 + IO$ (Drougas and Kosmas, 2007) is $CH_2O + I + O_2$ ($\Delta H_r = -5 \pm 6$ kJ mol$^{-1}$), consistent with the high yield of I and low yield of OIO found experimentally (Bale et al., 2005; Enami et al., 2006). Sensitivity studies have been carried out (Saiz-Lopez et al., 2014) using the preferred rate constant for this reaction of $2 \times 10^{-12}$ cm$^3$ molecule$^{-1}$ s$^{-1}$ (Dillon et al., 2006b), resulting in an enhancement of the ozone loss of 0.5% in the MBL and of less than 0.1% integrated throughout the troposphere in the $J_{IxOy}$ scenario, and similarly negligible enhancements in the Base scenario. Impacts in the $I_y$ partitioning are also very minor.

[i]     The temperature and pressure dependent rate constant (k) is computed based on the low pressure ($k_0$) and the high-pressure ($k_\infty$) rate coefficients following JPL-2010 (Sander et al., 2011).

[j]     The Fast rate constants and a thermally stable product $HOIO_2$ have been predicted theoretically (Plane et al., 2006), but no experimental studies reporting observation of $HOIO_2$ and its photochemical properties in the gas phase are available. Since the level of uncertainty is even larger than for the $I_xO_y$, it has not been included in the mechanism.

Table 2. Iodine chemistry scheme in CAM-Chem: Photochemical reactions.

| Reaction |
| --- |
| $CH_3I + h\nu \rightarrow CH_3O_2 + I$ |
| $CH_2I_2 + h\nu \rightarrow 2I$ [a] |
| $CH_2IBr + h\nu \rightarrow Br + I$ [a] |
| $CH_2ICl + h\nu \rightarrow Cl + I$ [a] |
| $I_2 + h\nu \rightarrow 2I$ |
| $IO + h\nu \rightarrow I + O$ |
| $OIO + h\nu \rightarrow I + O_2$ |
| $INO + h\nu \rightarrow I + NO$ |
| $INO_2 + h\nu \rightarrow I + NO_2$ [b] |
| $IONO_2 + h\nu \rightarrow I + NO_3$ |
| $HOI + h\nu \rightarrow I + OH$ |
| $IBr + h\nu \rightarrow I + Br$ |
| $ICl + h\nu \rightarrow I + Cl$ |
| $I_2O_2 + h\nu \rightarrow I + OIO$ [c] |
| $I_2O_3 + h\nu \rightarrow IO + OIO$ [c] |
| $I_2O_4 + h\nu \rightarrow OIO + OIO$ [c] |

Photolysis rates are computed online considering the actinic flux calculation in CAM-Chem. The absorption cross-sections and quantum yields for all species besides the $I_xO_y$ have been taken from IUPAC-2008 (Atkinson et al., 2007; Atkinson et al., 2008) and JPL-2010 (Sander et al., 2011).

[a] radical organic products are not considered.
[b] only the reaction channel reported in JPL 06-02 (Sander et al., 2006) is considered.
[c] photolysis reactions only considered in the $J_{IxOy}$ scheme (Saiz-Lopez et al., 2014).

Table 3. Iodine chemistry scheme in CAM-Chem: Heterogeneous reactions.

| Sea-salt aerosol reactions | Reactive uptake |
|---|---|
| $IONO_2 \rightarrow 0.5\ IBr + 0.5\ ICl$ | $\gamma = 0.01$ |
| $INO_2 \rightarrow 0.5\ IBr + 0.5\ ICl$ | $\gamma = 0.02$ |
| $HOI \rightarrow 0.5\ IBr + 0.5\ ICl$ | $\gamma = 0.06$ |
| $I_2O_2 \rightarrow$ | $\gamma = 0.01$[§] |
| $I_2O_3 \rightarrow$ | $\gamma = 0.01$[§] |
| $I_2O_4 \rightarrow$ | $\gamma = 0.01$[§] |

Values based on the THAMO model (Saiz-Lopez et al., 2008) and implemented in CAM-Chem following (Ordóñez et al., 2012).

[§] Deposition of $I_xO_y$ species on sea-salt aerosols has been included following the free regime approximation.

Table 4. Iodine chemistry scheme in CAM-Chem: Henry's Law constants and dry deposition velocities.

| Species | $k_0$ (M atm$^{-1}$) | Deposition velocity$^\S$ (cm s$^{-1}$) | Reference |
|---|---|---|---|
| IBr $^{ice}$ | $2.4 \times 10^1$ | – | 1 |
| ICl $^{ice}$ | $1.1 \times 10^2$ | – | 1 |
| HI | $7.8 \times 10^{-1}$ | 1.0 | 1 [a] |
| HOI – ($J_{IxOy}$ / Base) | $1.9 \times 10^3$ / $4.5 \times 10^3$ | 0.75 | 1 [b] |
| IONO$_2$ $^{ice}$ | $1.0 \times 10^6$ | 0.75 | 2 [c] |
| INO$_2$ $^{ice}$ | $3.0 \times 10^{-1}$ | 0.75 | 1 [d] |
| IO | $4.5 \times 10^2$ | – | 2 |
| OIO | $1.0 \times 10^4$ | – | 2 |
| I$_2$O$_2$ | $1.0 \times 10^4$ | 1.0 | 2 |
| I$_2$O$_3$ | $1.0 \times 10^4$ | 1.0 | 2 |
| I$_2$O$_4$ | $1.0 \times 10^4$ | 1.0 | 2 |

$^\S$ Dry deposition velocities are based on the THAMO model (Saiz-Lopez et al., 2008).
[1] Values reported in (Sander, 1999).
[2] Values based on the THAMO model (Saiz-Lopez et al., 2008).
[a] Considering a dissociation constant $K_a = 3.2 \times 10^9$ and a temperature dependent coefficient c = 9800 K
[b] Within the range of values given in the corresponding reference.
[c] Virtually infinite solubility is represented by using a very large arbitrary number.
[d] Value assumed to be equal to those of BrNO$_2$.
$^{ice}$ Species for which ice-uptake is considered following (Neu and Prather, 2012).

---

## Author Comment (AC3) · 2 Sep 2016

We thank the reviewer his/her time to carefully read through and for the comments. Below we provide a detailed point-by-point answer (AC – Author Comment) to each comment on our manuscript (RC – Referee Comment).

RC:

**Review of the Manuscript entitled: Iodine chemistry after Dark By A. Saiz-Lopez et al.**

**The manuscript describes new model calculations on the atmospheric chemistry of reactive iodine species encompassing a hypothetical reaction (R4) $NO_3$ + HOI $\rightarrow$ IO + $HNO_3$. The possibility of R1 actually occurring is investigated by molecular structure reactions. Moreover some possible discrepancies between observations and model calculations based on "conventional" I-chemistry may be solved by including R1.**

**The bulk of the manuscript is devoted to a comprehensive study of the consequences of introducing R4 (along with the earlier suggested reaction R1) in two models (1D and 2D). While one may ask whether a study based on a hypothetical reaction is warranted, I feel that the manuscript contains valuable material, which is within the scope of ACP and of interest to the scientific community. However, the manuscript contains a number of errors and unclear points (see list below) which must be corrected before publication. Also, given its speculative nature the manuscript is much too long and should be shortened considerably. This could be done by for instance removing most of the plots based on the 2D model calculations.**

AC:

Even though this work is based on modelling studies, we consider that all the information contained in this manuscript is necessary to present the new proposed reaction scheme and assess its effects on nighttime chemistry. Therefore we prefer to keep the manuscript as it is, regarding number of figures. The 3D model plots constitute a helpful way to show the global effect of this nighttime chemistry, so we also prefer to keep them in the manuscript.

**In detail there are the following deficiencies:**

RC:

**1) Page 4, lines 16, 17: Dawn spike of $NO_3$ not seen in measurements: Are these data conclusive? The spike is only short and the quoted measurements had comparatively poor temporal resolution. It should also be noted that atmospheric stability over the ocean is low at night because the atmosphere cools radiatively while the ocean surface temperature stays virtually constant (this is quite opposite to land conditions). Thus the IO precursors might simply be diluted during the**

**night. Since much of the manuscript hinges on the absence of the IO spike these points must be discussed.**

AC:

We assume that the reviewer is referring to the "dawn spike" in IO, rather than $NO_3$ as written in the comment. While the Read et al., 2008 and the Mahajan et al., 2010 papers present the hourly averaged diurnal profiles of IO at CVAO, the observations were made at a higher temporal resolution of 30 s. The observations made at higher temporal resolution also did not detect any IO above the detection limit, which was about 1 pptv for the 30 s data. Hence we feel that this is not a case of dilution, but rather the absence of an IO spike.

**RC:**

**2) Page 5, first two paragraphs of section 2: Here the description is not sufficiently clear, the way this reviewer understands it is: R1 is hypothetical, but its consequences were investigated earlier. In this work R2 – R4 are studied by molecular modelling finding that only R4 might play a role. In the rest of the manuscript, therefore the effects of including R1 + R4 are studied in detail.**

AC:

We are not saying in those paragraphs that R1 is hypothetical. In fact R1 was already included in the Saiz-Lopez et al., 2014 study. What we are doing is studying the feasibility of reactions R2-R4, and after checking that only R4 is viable, we study the effects of the nighttime R1 and R4 reactions using the THAMO and CAM-Chem models.

**RC:**

**3) Page 8, lines 9 to 13: How can the rate constant of R4 be only uncertain by a factor of 2 when the overall exothermicity of R4 is 11 KJ/mole (page 7, line 4) while the (one sigma?) uncertainty in the overall energy is 10 KJ/mole (page 6, line 15) and thus may be as low as 1 KJ/mole?**

AC:

Sentence changed to: "The absence of a barrier above the entrance channel, and the fact that the intermediate complexes and barrier are well below the entrance channel within their uncertainties, means that the uncertainty in $k_4$ principally arises from the estimated capture rate coefficient and so is likely to be no more than a factor of 2."

**RC:**

**4) Page 11, lines 10-12: The "significant increase of the sea-air flux of HOI and I2" might simply be an effect of the parameterisation of the process: Is the flux of the two species just given by the concentration difference between the two phases or is it (partly) determined by the rate of formation of the species? If the latter is the case then the flux might not change at all (or less than assumed by the model). This point needs discussing.**

AC:

The flux is given by the species concentration difference between the two phases.

**RC:**

**5) Page 12, lines 14 to 15: See comments about the "dawn spike" above.**

AC:

Please see answer above.

**RC:**

**6) Page 13, lines 8 to 11 and Fig. 6: It is not clear how the authors come to this conclusion: Fig. 6 is drawn on a semi-log scale and shows that (delta O3/NO3) / (delta k4/k4) is about 1/300. In other words a factor of 2 change in the rate constant of R4 has negligible effect (less than 1% change) on the NO3 concentration (or mixing ratio).**

AC:

This is correct. The sentence is wrong. It should be over two orders of magnitude change in $K_4$ leads to a factor of two change in $NO_3$. This has now been corrected in the manuscript.

**RC:**

**7) Page 13, lines 15 to 16 and Fig. 7: It is unclear what exactly is plotted in Fig. 7. (a) is it (calculated mixing ratio without R1, R4) minus (calculated mixing ratio including R1, R4). (b) Which mixing ratio is actually shown? Is it the surface value or the vertically averaged (over which altitude range?) mixing ratio? The caption of Fig. 7 uses the term "vertical mixing ratio", which is unknown to this reviewer.**

AC:

The plot is showing the difference in volume mixing ratio (for HOI and $I_2$) between the simulation including R1,R4 and the simulation without R1,R4.

We are showing the surface level mixing ratio averaged from 0 to ~150m altitude. The caption of figure 7 has been modified to include "at the surface level", and the typo "vertical mixing ratio" has been corrected to "volume mixing ratio".

**RC:**

**8) Page 14, line 1 and Fig. 8: See comment to Fig. 7. Also, here the text refers to "nighttime averaged differences" as opposed to 0AM to 1AM differences referred to in the explanation to Fig.7. This must be clarified**

AC:

Figure 8 is the same as figure 7 but for $IONO_2$ and $NO_3$. Nevertheless, for consistency we have changed in P13 line 17 "midnight averaged" for "nighttime averaged" in the reference to figure 7.

**RC:**

**9) Page 14, lines 9 to 11: The calculations about changes in NO3 levels are already speculative, to calculate changes in DMS (and other species) appears to be even more speculative (and not unexpected if one believes in the results regarding NO3). Therefore Fig. 9 adds little information and distracts from the main thrust of the manuscript, it should be removed.**

AC:

Even though the effect on DMS is not the main point of this work, we consider interesting to show the possible different ramifications of the new scheme, in this case how continental pollution outflow could affect DMS levels both geographically and quantitatively. The purpose is to show that this nighttime chemistry could potentially have an important impact on the nocturnal oxidizing capacity of the MBL.

**Minor points:**

**RC:**

**Page 3, lines 15-18: This appears to be too many reference for a topic (iodine particle formation) that is not mentioned later in the manuscript.**

AC:

Certainly there are too many references. We have now removed several of them.

**RC:**

**Page 4, line 5: Organic precursors contribute 1/3?**

AC:

Indeed they contribute ¼. Corrected to "only up to a fourth".

**RC:**

**Page 5, line 2 and following: What about BrO + DMS? The role of this reaction is neither mentioned nor discussed in the manuscript.**

AC:

We think this reaction is not within the scope of this work, although is part of the model´s chemical mechanism.

**RC:**

**Page 8, lines 14 to 17: The NO3 + CH2I2 reaction can not be ruled out on the basis of the rate constant of NO3 + CH2I2 being smaller than that of R4 since the concentration of CH2I2 may be higher than that of HOI.**

AC:

Even though $[CH_2I_2]$ could be higher than [HOI], the products of the reaction $NO_3+CH_2I_2$ are uncertain, as reported in the bibliography. So for now we have decided not to include that reaction in the models.

**RC:**

**Page 9, line 2**: Clarify that "this new chemistry" only means the introduction of R4.

AC:

We are using the 1D and 3D models to study the nocturnal effects of R1 and R4, not just R4. Previously we have theoretically studied the feasibility of two other reactions. Nevertheless the impacts of these reactions on the nocturnal oxidizing capacity of the atmosphere in marine regions affected by continental pollution outflow is considerable, affecting the levels of $NO_3$, HOI, $I_2$ and $IONO_2$ (among others). That is why we consider it "new chemistry". We will further clarify this point in the revised manuscript.

**RC:**

**Page 11, line 7: How much better is the agreement?**

AC:

Lawler et al.; 2014 reported $I_2$ values of ~1 pmol mol$^{-1}$ (ranging from 0.02 to 1.67 pmol mol$^{-1}$), so the results in figure 4 for the scenario including R1 and R4 are in better agreement than those for the scenario without R1 and R4. The modelled results without R1 and R4 are three times higher than Lawler et al., 2014 measurements.

**RC:**

**Page 11 line 13: The term "uptake" means aerosol uptake?**

AC:

Certainly, we have included "on aerosols" after "the uptake".

**RC:**

**Page 11 line 13 to page 12, line 10: The discussion of aerosol uptake appears to be out of place in the results and discussion section.**

AC:

We have now moved this discussion to the end of the "Atmospheric modelling" section.

**RC:**

**Page 12, line 11 and Fig. 5**: $IONO_2$ appears to be wrongly labelled as NO3.

AC:

This has been corrected in Fig 5.

[revised manuscript text omitted]

H 1.484,-0.657,-0.043
I  0.009,1.205,0.286
N -0.456,-2.265,0.030
O -1.052, -3.321,-0.0473
O -1.147,-1.195,-0.228
O 0.742,-2.161,0.333 | 55, 84, 118,   161, 196, 615, 629, 667, 705, 803, 968,   1228,   1273,1491, 3268 | 5.610, 0.916, 0.806 | -24.0 |
| IO-H-NO$_2$ TS | O 0.309,1.515,0.247
H -0.834,1.314,-0.017
I  1.280,-0.089,-0.093
N -2.349,-0.133,0.019
O -3.518, ,-0.429,-0.035
O -1.444,-0.962,0.257
O -2.019,1.117,-0.187 | 1249$i$, 70, 97, 103,   225, 472, 676, 698, 797, 806, 1041, 1147, 1308, 1513, 1626 | 6.300, 0.864, 0.767 | -16.4 |
| IO-HNO$_3$ complex | O 0.571,1.350,0.348
H -1.111,1.098,-0.020
I 1.870,0.0645,-0.152
N -2.503,-0.202,0.0186
O -3.673,-0.396,-0.170
O -1.654,-0.986,0.401
O -2.081,1.090,-0.242 | 35, 43, 76, 126, 198, 623, 677, 703, 772, 798, 939, 1331, 1416, 1713, 3281 | 7.058, 0.605, 0.566 | -34.8 |
| IO + HNO$_3$ | | 648 & 477, 585, 649, 782, 901, 1320, 1345, 1738, 3724 | 9.844 & 13.01, 12.05, 6.258 | -10.6 |

[a] Cartesian co-ordinates in Å.  [b] In cm$^{-1}$.  [c] In GHz.  [d] In kJ mol$^{-1}$, including zero-point energy and spin-
orbit coupling of I and IO (see text).

[Figure]

**Figure 1**. New nocturnal iodine chemistry (in white) implemented in the THAMO and CAM-Chem models.

[Figure]

**Figure 2:** (a) Transition state for the reaction between HOI and $NO_2$ to form $HNO_3$ + I; (b)
complex formed between HOI and $HNO_3$, which then reacts via transition state (c) to form
IONO2 + $H_2O$.

[Figure]

**Figure 3**. Potential energy surface for the reaction between HOI and $NO_3$, which contains two
intermediate complexes separated by a submerged barrier.

[Figure]

**Figure 4.** THAMO modeled diurnal variation of HOI, $I_2$ (upper panels) and the HOI/$I_2$ flux from the ocean surface (bottom panel). The right hand panels are from scenario 1, which do not include night time reactions of HOI and $I_2$ with $NO_3$, while the left hand panels include the reactions in scenario 2. In bottom panel red lines represent scenario 1, while black lines correspond to scenario 2.

[Figure]

**Figure 5.** THAMO modeled diurnal variation of IO, $NO_3$ and the $IONO_2$. The right hand panels
are from scenario 1, which do not include night time reactions of HOI and $I_2$ with $NO_3$, while the
left hand panels include the reactions in scenario 2.

[Figure]

**Figure 6**. Sensitivity run showing the effect of the uncertainty in the rate constant estimation on the reduction of $NO_3$ at the surface - the red point is the theoretical estimate.

[Figure]

**Figure 7.** Modelled annual average of HOI (a) and I$_2$ (b) during night time at the surface level. The panels show the difference in volume mixing ratio between the simulations with and without reactions (1) and (4).

[Figure]

**Figure 8.** Modelled annual average of IONO$_2$ (a) and NO$_3$ (b) during night time at the surface level, as the difference in volume mixing ratio between the simulations with and without reactions (1) and (4).

[Figure]

**Figure 9**. Increase in the DMS levels during night time at the surface level due to the inclusion
of the reactions R1 and R4 in CAM-Chem.

[Figure]

**Figure 10**. Hourly averaged concentration of HOI, $IONO_2$ and $I_2$ in the Mediterranean Sea at the
surface level (lon:$10^{\circ} \rightarrow 20^{\circ}$E, lat:$33^{\circ} \rightarrow 40^{\circ}$N)

[Figure]

**Figure 11**. Hourly averaged concentration of HOI, IONO$_2$ and I$_2$ (upper panel) and NO$_3$ (bottom panel) in the Pacific Ocean at the south of Baja California peninsula at the surface level (lon: -110º→-106ºE, lat:16º→23ºN)

Supplementary information for

**Iodine chemistry after dark**

Alfonso Saiz-Lopez[1], John M.C. Plane[2], Carlos A. Cuevas[1], Anoop S. Mahajan[3], Jean-François Lamarque[4] and Douglas E. Kinnison[4]

[1]Department of Atmospheric Chemistry and Climate, Institute of Physical Chemistry Rocasolano, CSIC, Madrid, Spain

[2]School of Chemistry, University of Leeds, Leeds, UK

[3]Indian Institute of Tropical Meteorology, Pune, India

[4]Atmospheric Chemistry Observations and Modelling, NCAR, Colorado, USA

Correspondence to: A. Saiz-Lopez (a.saiz@csic.es)

Table 1. Iodine chemistry scheme in CAM-Chem: Bimolecular, thermal decomposition and termolecular reactions.

| Reaction | $k$ / cm$^3$ molecule$^{-1}$ s$^{-1}$ | Notes |
|---|---|---|
| $I + O_3 \rightarrow IO + O_2$ | $2.1 \times 10^{-11}\ e^{(-830/T)}$ | 1 |
| $IO + O_3 \rightarrow OIO + O_2$ | $3.6 \times 10^{-16}$ | 2 |
| $I + HO_2 \rightarrow HI + O_2$ | $1.5 \times 10^{-11}\ e^{(-1090/T)}$ | 3 |
| $IO + NO \rightarrow I + NO_2$ | $7.15 \times 10^{-12}\ e^{(300/T)}$ | 1 |
| $IO + HO_2 \rightarrow HOI + O_2$ | $1.4 \times 10^{-11}\ e^{(540/T)}$ | 1 |
| $IO + IO \rightarrow OIO + I$ | $2.13 \times 10^{-11}\ e^{(180/T)} \times$ $[1 + e^{(-p/191.42)}]$ | 1, 4 |
| $IO + IO \rightarrow I_2O_2$ | $3.27 \times 10^{-11}\ e^{(180/T)} \times$ $[1 - 0.65\ e^{(-p/191.42)}]$ | 1, 4 |
| $IO + OIO \rightarrow I_2O_3$ | $w_1 \cdot exp\ (w_2 \cdot T)$ [a] | 4, 5, 6 [g] |
| $OIO + OIO \rightarrow I_2O_4$ | $w_1 \cdot exp\ (w_2 \cdot T)$ [b] | 4, 5, 6 [g] |
| $I_2 + O \rightarrow IO + I$ | $1.25 \times 10^{-10}$ | 1 |
| $IO + O \rightarrow I + O_2$ | $1.4 \times 10^{-10}$ | 1 |
| $IO + OH \rightarrow HO_2 + I$ | $1.0 \times 10^{-10}$ | 7 |
| $I_2O_2 \rightarrow OIO + I$ | $w_1 \cdot exp\ (w_2/T)$ [c] | 5, 6, 8 [g] |
| $I_2O_2 \rightarrow IO + IO$ | $w_1 \cdot exp\ (w_2/T)$ [d] | 5, 6, 8 [g] |
| $I_2O_4 \rightarrow 2\ OIO$ | $w_1 \cdot exp\ (w_2/T)$ [e] | 5, 8 [g] |
| $I_2 + OH \rightarrow HOI + I$ | $1.8\ x\ 10^{-10}$ | 3 |
| $I_2 + NO_3 \rightarrow I + IONO_2$ | $1.5 \times 10^{-12}$ | 9 |
| $I + NO_3 \rightarrow IO + NO_2$ | $1.0 \times 10^{-10}$ | 1 |
| $OH + HI \rightarrow I + H_2O$ | $1.6 \times 10^{-11}\ e^{(440/T)}$ | 1 |
| $I + IONO_2 \rightarrow I_2 + NO_3$ | $9.1 \times 10^{-11}\ e^{(-146/T)}$ | 5 |
| $HOI + OH \rightarrow IO + H_2O$ | $2.0 \times 10^{-13}$ | 10 |
| $IO + DMS \rightarrow DMSO + I$ | $3.2 \times 10^{-13}\ e^{(-925/T)}$ | 11 |
| $INO_2 \rightarrow I + NO_2$ | $1008 \times 10^{15}\ e^{(-13670/T)}$ | 12, 13, 14 |
| $IONO_2 \rightarrow IO + NO_2$ | $w_1 \cdot exp\ (w_2/T)$ [f] | 5, 15 |
| $INO + INO \rightarrow I_2 + 2NO$ | $8.4 \times 10^{-11}\ e^{(-2620/T)}$ | 3 |
| $INO_2 + INO_2 \rightarrow I_2 + 2NO_2$ | $4.7 \times 10^{-13}\ e^{(-1670/T)}$ | 1 |
| $OIO + NO \rightarrow IO + NO_2$ | $1.1 \times 10^{-12}\ e^{(542/T)}$ | 14 |
| $HI + NO_3 \rightarrow I + HNO_3$ | $1.3 \times 10^{-12}\ e^{(-1830/T)}$ | 16 |
| $IO + BrO \rightarrow Br + I + O_2$ | $0.30 \times 10^{-11}\ e^{(510/T)}$ | 1 |
| $IO + BrO \rightarrow Br + OIO$ | $1.20 \times 10^{-11}\ e^{(510/T)}$ | 1 |
| $I + BrO \rightarrow IO + Br$ | $1.44 \times 10^{-11}$ | 17, 18, 19 |

| | | |
|---|---|---|
| $IO + ClO \rightarrow I + OClO$ | $2.585 \times 10^{-12}\, e^{(280/T)}$ | 1 |
| $IO + ClO \rightarrow I + Cl + O_2$ | $1.175 \times 10^{-12}\, e^{(280/T)}$ | 1 |
| $IO + ClO \rightarrow ICl + O_2$ | $0.940 \times 10^{-12}\, e^{(280/T)}$ | 1 |
| $IO + Br \rightarrow I + BrO$ | $2.49 \times 10^{-11}$ | 18, 19 |
| $IO + NO_3 \rightarrow OIO + NO_2$ | $9.0 \times 10^{-12}$ | 20 |
| $IO + CH_3O_2 \rightarrow CH_2O + I + HO_2$ | $2.0 \times 10^{-12}$ | 2[h] |
| $CH_3I + OH \rightarrow I + H_2O + HO_2$ | $2.90 \times 10^{-12}\, e^{(-1100/T)}$ | 3 |
| $I + NO_2\, (+ M) \rightarrow INO_2\, (+ M)$ | $k_0 = 3 \times 10^{-31} \times (T/300)^{-1}$
 $k_\infty = 6.6 \times 10^{-11}$ | 3[i] |
| $IO + NO_2\, (+ M) \rightarrow IONO_2\, (+ M)$ | $k_0 = 6.5 \times 10^{-31} \times (T/300)^{-3.5}$
 $k_\infty = 7.6 \times 10^{-12} \times (T/300)^{-1.5}$ | 3[i] |
| $I + NO\, (+ M) \rightarrow INO\, (+ M)$ | $k_0 = 1.8 \times 10^{-32} \times (T/300)^{-1}$
 $k_\infty = 1.7 \times 10^{-11}$ | 3[i] |
| $OIO + OH\, (+ M) \rightarrow HOIO_2\, (+ M)$ | $k_0 = 1.5 \times 10^{-27} \times (T/300)^{-3.93}$
 $k_\infty = 7.76 \times 10^{-10} \times (T/300)^{-0.8}$ | 14[j] |
| $HOI + NO_3 \rightarrow IO + HNO_3$ | $2.7 \times 10^{-12}\, (300/T)^{2.66}$ | 21 |

[1] IUPAC-2008 (Atkinson et al., 2007) ; [2](Dillon et al., 2006b); [3] JPL-2010 (Sander et al., 2011); [4](Gómez Martín et al., 2007); [5](Kaltsoyannis and Plane, 2008); [6](Galvez et al., 2013); [7](Bösch et al., 2003); [8] (Gómez Martín and Plane, 2009); [9](Chambers et al., 1992); [10](Chameides and Davis, 1980); [11](Dillon et al., 2006a); [12](McFiggans et al., 2000); [13](Jenkin et al., 1985); [14](Plane et al., 2006); [15](Allan and Plane, 2002); [16] (Lancar et al., 1991); [17](Laszlo et al., 1997); [18](Bedjanian et al., 1997); [19](Gilles et al., 1997); [20](Dillon et al., 2008); [21]This work.

[a] $w1 = 4.687 \times 10^{-10} - 1.3855 \times 10^{-5} \times e^{(-0.75\, p\, /\, 1.62265)} + 5.51868 \times 10^{-10} \times e^{(-0.75\, p\, /\, 199.328)}$

$w2 = -0.00331 - 0.00514 \times e^{(-0.75\, p\, /\, 325.68711)} - 0.00444 \times e^{(-0.75\, p\, /\, 40.81609)}$

[b] $w1 = 1.1659 \times 10^{-9} - 7.79644 \times 10^{-10}\, e^{(-0.75\, p\, /\, 22.09281)} + 1.03779 \times 10^{-9} \times e^{(-0.75\, p\, /\, 568.15381)}$

$w2 = -0.00813 - 0.00382 \times e^{(-0.75\, p\, /\, 45.57591)} - 0.00643 \times e^{(-0.75\, p\, /\, 417.95061)}$

[c] $w1 = 3.54288 \times 10^{10} + 1.8523 \times 10^{11} \times 0.75\, p - 1.45435 \times 10^{8} \times (0.75p)^2 + 60799.4344 \times (0.75p)^3$

$w2 = -9681.65989 + 346.95538 \times e^{(-0.75\, p\, /\, 343.25322)} + 251.78032 \times e^{(-0.75\, p\, /\, 44.1466)}$

[d] $w1 = 255335000000 - 4418880000 \times 0.75\, p + 85618600 \times (0.75\, p)^2 + 14218.81 \times (0.75\, p)^3$

$w2 = -11466.82304 + 597.01334 \times e^{(-0.75\, p\, /\, 1382.62325)} - 167.3391 \times e^{(-0.75\, p\, /\, 43.75089)}$

[e] $w1 = -1.92626 \times 10^{14} + 4.67414 \times 10^{13} \times 0.75\, p - 3.68651 \times 10^{8} \times (0.75\, p)^2 - 3.09109 \times 10^{6} \times (0.75\, p)^3$

$w2 = -12302.15294 + 252.78367 \times e^{(-0.75\, p\, /\, 46.12733)} + 437.62868 \times e^{(-0.75\, p\, /\, 428.4413)}$

$f$      w1 = -2.63544 x $10^{13}$ + 4.32845 x $10^{12}$ x (0.75 p) + 3.73758 x $10^8$ x (0.75 p) $^2$ - 628468.76313 x (0.75 p) $^3$

   w2 = -13847.85015 + 240.34465 x e $^{(-0.75\,p\,/\,49.27141)}$ + 451.35864 x e $^{(-0.75\,p\,/}$ $_{436.87605)}$

$g$      The empirical expressions of the form $w_1 \cdot \exp(w_2 \cdot T)$ were obtained by non-linear least squares fitting of *Rice–Ramsperger–Kassel–Marcus* (RRKM) theoretical results for the indicated reaction rate constants and thermal dissociation rates in the (27 – 1013) hPa pressure range. RRKM calculations were carried out using the MESMER algorithm (Glowacki et al., 2012) as indicated in the corresponding references (e.g. (Galvez et al., 2013). Expression $^a$ produces negative values outside the range of modelled rate constants (p < 20 hPa), and therefore a fixed rate constant of 3 x $10^{-11}$ $cm^3$ molecule$^{-1}$ s$^{-1}$ was assumed. Expressions $^e$ and $^f$ generate negligible dissociation rates below ~500 hPa which become negative at ~8 hPa – in this case they are set to zero below that pressure.

$h$      Updated heats of formation for IO, OIO, and $CH_3O_2$ (Dooley et al., 2008; Gómez Martín and Plane, 2009; Knyazev and Slagle, 1998) show that the only accessible exothermic product channel of $CH_3O_2$ + IO (Drougas and Kosmas, 2007) is $CH_2O$ + I + $O_2$ ($\Delta H_r$ = -5 ± 6 kJ mol$^{-1}$), consistent with the high yield of I and low yield of OIO found experimentally (Bale et al., 2005; Enami et al., 2006). Sensitivity studies have been carried out (Saiz-Lopez et al., 2014) using the preferred rate constant for this reaction of $2 \times 10^{-12}$ $cm^3$ molecule$^{-1}$ s$^{-1}$ (Dillon et al., 2006b), resulting in an enhancement of the ozone loss of 0.5% in the MBL and of less than 0.1% integrated throughout the troposphere in the $J_{IxOy}$ scenario, and similarly negligible enhancements in the Base scenario. Impacts in the $I_y$ partitioning are also very minor.

$i$      The temperature and pressure dependent rate constant (k) is computed based on the low pressure ($k_0$) and the high-pressure ($k_\infty$) rate coefficients following JPL-2010 (Sander et al., 2011).

$j$      The Fast rate constants and a thermally stable product $HOIO_2$ have been predicted theoretically (Plane et al., 2006), but no experimental studies reporting observation of $HOIO_2$ and its photochemical properties in the gas phase are available. Since the level of uncertainty is even larger than for the $I_xO_y$, it has not been included in the mechanism.

Table 2. Iodine chemistry scheme in CAM-Chem: Photochemical reactions.

| Reaction |
| --- |
| $CH_3I + h\nu \rightarrow CH_3O_2 + I$ |
| $CH_2I_2 + h\nu \rightarrow 2I$ [a] |
| $CH_2IBr + h\nu \rightarrow Br + I$ [a] |
| $CH_2ICl + h\nu \rightarrow Cl + I$ [a] |
| $I_2 + h\nu \rightarrow 2I$ |
| $IO + h\nu \rightarrow I + O$ |
| $OIO + h\nu \rightarrow I + O_2$ |
| $INO + h\nu \rightarrow I + NO$ |
| $INO_2 + h\nu \rightarrow I + NO_2$ [b] |
| $IONO_2 + h\nu \rightarrow I + NO_3$ |
| $HOI + h\nu \rightarrow I + OH$ |
| $IBr + h\nu \rightarrow I + Br$ |
| $ICl + h\nu \rightarrow I + Cl$ |
| $I_2O_2 + h\nu \rightarrow I + OIO$ [c] |
| $I_2O_3 + h\nu \rightarrow IO + OIO$ [c] |
| $I_2O_4 + h\nu \rightarrow OIO + OIO$ [c] |

Photolysis rates are computed online considering the actinic flux calculation in CAM-Chem. The absorption cross-sections and quantum yields for all species besides the $I_xO_y$ have been taken from IUPAC-2008 (Atkinson et al., 2007; Atkinson et al., 2008) and JPL-2010 (Sander et al., 2011).

[a] radical organic products are not considered.
[b] only the reaction channel reported in JPL 06-02 (Sander et al., 2006) is considered.
[c] photolysis reactions only considered in the $J_{IxOy}$ scheme (Saiz-Lopez et al., 2014).

Table 3. Iodine chemistry scheme in CAM-Chem: Heterogeneous reactions.

| Sea-salt aerosol reactions | Reactive uptake |
|---|---|
| $IONO_2 \rightarrow 0.5\ IBr + 0.5\ ICl$ | $\gamma = 0.01$ |
| $INO_2 \rightarrow 0.5\ IBr + 0.5\ ICl$ | $\gamma = 0.02$ |
| $HOI \rightarrow 0.5\ IBr + 0.5\ ICl$ | $\gamma = 0.06$ |
| $I_2O_2 \rightarrow$ | $\gamma = 0.01$[§] |
| $I_2O_3 \rightarrow$ | $\gamma = 0.01$[§] |
| $I_2O_4 \rightarrow$ | $\gamma = 0.01$[§] |

Values based on the THAMO model (Saiz-Lopez et al., 2008) and implemented in CAM-Chem following (Ordóñez et al., 2012).

[§] Deposition of $I_xO_y$ species on sea-salt aerosols has been included following the free regime approximation.

Table 4. Iodine chemistry scheme in CAM-Chem: Henry's Law constants and dry deposition velocities.

| Species | $k_0$ (M atm$^{-1}$) | Deposition velocity$^§$ (cm s$^{-1}$) | Reference |
|---------|----------------------|----------------------------------------|-----------|
| IBr $^{ice}$ | $2.4 \times 10^1$ | – | 1 |
| ICl $^{ice}$ | $1.1 \times 10^2$ | – | 1 |
| HI | $7.8 \times 10^{-1}$ | 1.0 | 1 [a] |
| HOI – ($J_{IxOy}$ / Base) | $1.9 \times 10^3$ / $4.5 \times 10^3$ | 0.75 | 1 [b] |
| IONO$_2$ $^{ice}$ | $1.0 \times 10^6$ | 0.75 | 2 [c] |
| INO$_2$ $^{ice}$ | $3.0 \times 10^{-1}$ | 0.75 | 1 [d] |
| IO | $4.5 \times 10^2$ | – | 2 |
| OIO | $1.0 \times 10^4$ | – | 2 |
| I$_2$O$_2$ | $1.0 \times 10^4$ | 1.0 | 2 |
| I$_2$O$_3$ | $1.0 \times 10^4$ | 1.0 | 2 |
| I$_2$O$_4$ | $1.0 \times 10^4$ | 1.0 | 2 |

$^§$ Dry deposition velocities are based on the THAMO model (Saiz-Lopez et al., 2008).
[1] Values reported in (Sander, 1999).
[2] Values based on the THAMO model (Saiz-Lopez et al., 2008).
[a] Considering a dissociation constant $K_a = 3.2 \times 10^9$ and a temperature dependent coefficient c = 9800 K
[b] Within the range of values given in the corresponding reference.
[c] Virtually infinite solubility is represented by using a very large arbitrary number.
[d] Value assumed to be equal to those of BrNO$_2$.
$^{ice}$ Species for which ice-uptake is considered following (Neu and Prather, 2012).

---

## Author Comment (AC4) · 2 Sep 2016

We thank the Referee for his/her comments. Below we provide a detailed point-by-point answer (AC – Author Comment) to each comment on our manuscript (RC – Referee Comment).

**RC:**

**Saiz-Lopez et al. investigate the nighttime chemistry of iodine. The study is very interesting and I recommend publication in ACP after considering several changes as described below.**

**- Title:**

**I find the expression "after dark" quite unusual for a scientific paper. Why not simply call it "nighttime chemistry"?**

AC:

Thank you for the suggestion about the title change, however we prefer to keep the title as is.

**RC**

**- Section 4:**

**Instead of presenting the full chemical mechanism, the authors refer to 6 previous publications. I find it quite tedious that I have to obtain and read 6 additional papers if I want to check the currently used mechansim. I suggest to provide the full mechanism (exactly as it was used in this study) together with this paper, e.g. in the supplement.**

AC:

As the reviewer suggests, we have included the full iodine chemistry scheme in a supplementary information document. The scheme is referenced in page 22 line 12 in the main text.

**RC:**

**- Page 11, line 1:**

**It is said that "HOI peaks during the daytime". I think a better description would be to say that it peaks just before sunset. What is the reason for the sunset peak?**

AC:

We have modified that phrase to "HOI is present during daytime, due to its production through the reaction of IO with $HO_2$, and peaks just before sunset" as suggested by the reviewer.

The main loss channel for HOI is the photolysis to atomic iodine. Therefore before sunset the solar radiation intensity decreases and this channel is less effective. Another important loss channel for HOI is the heterogeneous reaction with halide anions on aerosols. After sunset the precursor IO is not formed, and consequently the HOI levels drop.

**RC:**

**- Page 11, lines 13-14:**

**"It should be noted that during nighttime the uptake of emitted species such as I2 and HOI, and the uptake of reservoir species such as IONO2, can play a major role in the cycling of iodine."**

**What is meant by "uptake"? Uptake on aerosols? On clouds?**

AC:

In this paragraph we are referring to uptake on aerosols. We have now mentioned in the paragraph that we are referring to aerosols.

**RC:**

**- Page 11, line 21:**

**The outdated JPL recommendation Sander et al. 2006 is cited here for mass accommodation coefficients. Has it been checked if there are any updates in the current recommendation JPL 2015?**

AC:

Yes, we have checked the mass accommodation coefficients but they haven't been updated.

**RC:**

**- Page 12, lines 16-17:**

**"The IO dawn spike [...] is due to a buildup of the emitted I2 and HOI [...] over the night".**

**I cannot see a buildup of HOI in Fig. 4.**

AC:

Perhaps it is difficult to appreciate, but the levels of HOI in figure 4 for scenario 1 never drop to zero during the night. Note that at midnight the levels are between 4 and 5 pmol mol$^{-1}$, and from 0:00 to 6:00 hours the concentrations only drop to 3.5 pmol mol$^{-1}$ in the lowest levels of the model. This situation is different from scenario 2, in which [HOI] at midnight is around 1 pmol mol$^{-1}$, and drop to values close to zero at 6:00 hours. So we believe that there is a substantial difference between the two scenarios when it comes to [HOI], with a considerable buildup of HOI during night for scenario 1.

**RC:**

**- Page 12, line 21:**

**"Reactions R1 and R4 also reduce the NO3 mixing ratio (Fig. 4, middle panels)."**

**Should this be Fig. 5?**

AC:

Certainly, typo corrected.

**RC:**

**- Page 13, line 17:**

**Please define "LT".**

AC:

LT refers to Local Time.

**RC:**

**- Table 2: The numbers listed here are probably wave numbers, not vibrational frequencies.**

AC:

It is customary to quote vibrational frequencies as wave numbers. There is a mistake in the footnotes, which should be:

[a] Cartesian co-ordinates in Å.  [b] In cm$^{-1}$.  [c] In GHz.  [d] In kJ mol$^{-1}$, including zero-point energy and spin-orbit coupling of I and IO (see text).

**RC:**

**- Figs. 4 and 5: A color scale should be shown. Also, it would be easier to compare the left hand panels with the right hand panels if the same color scale was used.**

AC:

Thanks, however we respectfully think that the figure is clear enough as is, the numbers are shown in the plots.

**RC:**

**- Fig. 5: For consistency, the name IONO2 should be used in the plots, not INO3.**

AC:

We have corrected Fig 5.

**RC:**

**- Fig. 7: What is a "vertical mixing ratio"?**

AC:

Thanks, it is a typo. Corrected in the caption of figure 7 to "volume mixing ratio".

**RC:**

**- Figs. 7,8,9:**

**"without reactions (1) and (2)"**

**Should this be "without reactions (1) and (4)"?**

AC:

Indeed, we are referring in the caption to reaction 1 and 4. The typo has now been corrected.

[revised manuscript text omitted]

H 1.484,-0.657,-0.043
I  0.009,1.205,0.286
N -0.456,-2.265,0.030
O -1.052, -3.321,-0.0473
O -1.147,-1.195,-0.228
O 0.742,-2.161,0.333 | 55, 84, 118,   161, 196, 615, 629, 667, 705, 803, 968,   1228,   1273,1491, 3268 | 5.610, 0.916, 0.806 | -24.0 |
| IO-H-NO$_2$ TS | O 0.309,1.515,0.247
H -0.834,1.314,-0.017
I  1.280,-0.089,-0.093
N -2.349,-0.133,0.019
O -3.518, ,-0.429,-0.035
O -1.444,-0.962,0.257
O -2.019,1.117,-0.187 | 1249*i*, 70, 97, 103,   225, 472, 676, 698, 797, 806, 1041, 1147, 1308, 1513, 1626 | 6.300, 0.864, 0.767 | -16.4 |
| IO-HNO$_3$ complex | O 0.571,1.350,0.348
H -1.111,1.098,-0.020
I 1.870,0.0645,-0.152
N -2.503,-0.202,0.0186
O -3.673,-0.396,-0.170
O -1.654,-0.986,0.401
O -2.081,1.090,-0.242 | 35, 43, 76, 126, 198, 623, 677, 703, 772, 798, 939, 1331, 1416, 1713, 3281 | 7.058, 0.605, 0.566 | -34.8 |
| IO + HNO$_3$ | | 648 & 477, 585, 649, 782, 901, 1320, 1345, 1738, 3724 | 9.844 & 13.01, 12.05, 6.258 | -10.6 |

[a] Cartesian co-ordinates in Å.  [b] In cm$^{-1}$.  [c] In GHz.  [d] In kJ mol$^{-1}$, including zero-point energy and spin-orbit coupling of I and IO (see text).

[Figure]

**Figure 1**. New nocturnal iodine chemistry (in white) implemented in the THAMO and CAM-Chem models.

[Figure]

**Figure 2:** (a)  Transition state for the reaction between HOI and $NO_2$ to form $HNO_3$ + I; (b)
complex formed between HOI and $HNO_3$, which then reacts via transition state (c) to form
IONO2 + $H_2O$.

[Figure]

**Figure 3**. Potential energy surface for the reaction between HOI and $NO_3$, which contains two
intermediate complexes separated by a submerged barrier.

[Figure]

**Figure 4.** THAMO modeled diurnal variation of HOI, $I_2$ (upper panels) and the HOI/$I_2$ flux from the ocean surface (bottom panel). The right hand panels are from scenario 1, which do not include night time reactions of HOI and $I_2$ with $NO_3$, while the left hand panels include the reactions in scenario 2. In bottom panel red lines represent scenario 1, while black lines correspond to scenario 2.

[Figure]

**Figure 5.** THAMO modeled diurnal variation of IO, NO$_3$ and the IONO$_2$. The right hand panels
are from scenario 1, which do not include night time reactions of HOI and I$_2$ with NO$_3$, while the
left hand panels include the reactions in scenario 2.

[Figure]

**Figure 6**. Sensitivity run showing the effect of the uncertainty in the rate constant estimation on the reduction of $NO_3$ at the surface - the red point is the theoretical estimate.

[Figure]

**Figure 7.** Modelled annual average of HOI (a) and I$_2$ (b) during night time at the surface level. The panels show the difference in volume mixing ratio between the simulations with and without reactions (1) and (4).

[Figure]

**Figure 8.** Modelled annual average of IONO₂ (a) and NO₃ (b) during night time at the surface
level, as the difference in volume mixing ratio between the simulations with and without
reactions (1) and (4).

[Figure]

**Figure 9**. Increase in the DMS levels during night time at the surface level due to the inclusion
of the reactions R1 and R4 in CAM-Chem.

[Figure]

**Figure 10**. Hourly averaged concentration of HOI, $IONO_2$ and $I_2$ in the Mediterranean Sea at the
surface level (lon:10º→20ºE, lat:33º→40ºN)

[Figure]

**Figure 11**. Hourly averaged concentration of HOI, IONO$_2$ and I$_2$ (upper panel) and NO$_3$ (bottom panel) in the Pacific Ocean at the south of Baja California peninsula at the surface level (lon: -110º→-106ºE, lat:16º→23ºN)

Supplementary information for

**Iodine chemistry after dark**

Alfonso Saiz-Lopez[1], John M.C. Plane[2], Carlos A. Cuevas[1], Anoop S. Mahajan[3], Jean-François Lamarque[4] and Douglas E. Kinnison[4]

[1]Department of Atmospheric Chemistry and Climate, Institute of Physical Chemistry Rocasolano, CSIC, Madrid, Spain

[2]School of Chemistry, University of Leeds, Leeds, UK

[3]Indian Institute of Tropical Meteorology, Pune, India

[4]Atmospheric Chemistry Observations and Modelling, NCAR, Colorado, USA

Correspondence to: A. Saiz-Lopez (a.saiz@csic.es)

Table 1. Iodine chemistry scheme in CAM-Chem: Bimolecular, thermal decomposition and termolecular reactions.

| Reaction | $k$ / cm$^3$ molecule$^{-1}$ s$^{-1}$ | Notes |
|---|---|---|
| $I + O_3 \rightarrow IO + O_2$ | $2.1 \times 10^{-11}$ e$^{(-830/T)}$ | 1 |
| $IO + O_3 \rightarrow OIO + O_2$ | $3.6 \times 10^{-16}$ | 2 |
| $I + HO_2 \rightarrow HI + O_2$ | $1.5 \times 10^{-11}$ e$^{(-1090/T)}$ | 3 |
| $IO + NO \rightarrow I + NO_2$ | $7.15 \times 10^{-12}$ e$^{(300/T)}$ | 1 |
| $IO + HO_2 \rightarrow HOI + O_2$ | $1.4 \times 10^{-11}$ e$^{(540/T)}$ | 1 |
| $IO + IO \rightarrow OIO + I$ | $2.13 \times 10^{-11}$ e$^{(180/T)} \times$ $[1 + $e$^{(-p/191.42)}]$ | 1, 4 |
| $IO + IO \rightarrow I_2O_2$ | $3.27 \times 10^{-11}$ e$^{(180/T)} \times$ $[1 - 0.65 $e$^{(-p/191.42)}]$ | 1, 4 |
| $IO + OIO \rightarrow I_2O_3$ | $w_1 \cdot \exp(w_2 \cdot T)$ $^a$ | 4, 5, 6 $^g$ |
| $OIO + OIO \rightarrow I_2O_4$ | $w_1 \cdot \exp(w_2 \cdot T)$ $^b$ | 4, 5, 6 $^g$ |
| $I_2 + O \rightarrow IO + I$ | $1.25 \times 10^{-10}$ | 1 |
| $IO + O \rightarrow I + O_2$ | $1.4 \times 10^{-10}$ | 1 |
| $IO + OH \rightarrow HO_2 + I$ | $1.0 \times 10^{-10}$ | 7 |
| $I_2O_2 \rightarrow OIO + I$ | $w_1 \cdot \exp(w_2/T)$ $^c$ | 5, 6, 8 $^g$ |
| $I_2O_2 \rightarrow IO + IO$ | $w_1 \cdot \exp(w_2/T)$ $^d$ | 5, 6, 8 $^g$ |
| $I_2O_4 \rightarrow 2\ OIO$ | $w_1 \cdot \exp(w_2/T)$ $^e$ | 5, 8 $^g$ |
| $I_2 + OH \rightarrow HOI + I$ | $1.8 \times 10^{-10}$ | 3 |
| $I_2 + NO_3 \rightarrow I + IONO_2$ | $1.5 \times 10^{-12}$ | 9 |
| $I + NO_3 \rightarrow IO + NO_2$ | $1.0 \times 10^{-10}$ | 1 |
| $OH + HI \rightarrow I + H_2O$ | $1.6 \times 10^{-11}$ e$^{(440/T)}$ | 1 |
| $I + IONO_2 \rightarrow I_2 + NO_3$ | $9.1 \times 10^{-11}$ e$^{(-146/T)}$ | 5 |
| $HOI + OH \rightarrow IO + H_2O$ | $2.0 \times 10^{-13}$ | 10 |
| $IO + DMS \rightarrow DMSO + I$ | $3.2 \times 10^{-13}$ e$^{(-925/T)}$ | 11 |
| $INO_2 \rightarrow I + NO_2$ | $1008 \times 10^{15}$ e$^{(-13670/T)}$ | 12, 13, 14 |
| $IONO_2 \rightarrow IO + NO_2$ | $w_1 \cdot \exp(w_2/T)$ $^f$ | 5, 15 |
| $INO + INO \rightarrow I_2 + 2NO$ | $8.4 \times 10^{-11}$ e$^{(-2620/T)}$ | 3 |
| $INO_2 + INO_2 \rightarrow I_2 + 2NO_2$ | $4.7 \times 10^{-13}$ e$^{(-1670/T)}$ | 1 |
| $OIO + NO \rightarrow IO + NO_2$ | $1.1 \times 10^{-12}$ e$^{(542/T)}$ | 14 |
| $HI + NO_3 \rightarrow I + HNO_3$ | $1.3 \times 10^{-12}$ e$^{(-1830/T)}$ | 16 |
| $IO + BrO \rightarrow Br + I + O_2$ | $0.30 \times 10^{-11}$ e$^{(510/T)}$ | 1 |
| $IO + BrO \rightarrow Br + OIO$ | $1.20 \times 10^{-11}$ e$^{(510/T)}$ | 1 |
| $I + BrO \rightarrow IO + Br$ | $1.44 \times 10^{-11}$ | 17, 18, 19 |

| | | |
|---|---|---|
| $IO + ClO \rightarrow I + OClO$ | $2.585 \times 10^{-12} e^{(280/T)}$ | 1 |
| $IO + ClO \rightarrow I + Cl + O_2$ | $1.175 \times 10^{-12} e^{(280/T)}$ | 1 |
| $IO + ClO \rightarrow ICl + O_2$ | $0.940 \times 10^{-12} e^{(280/T)}$ | 1 |
| $IO + Br \rightarrow I + BrO$ | $2.49 \times 10^{-11}$ | 18, 19 |
| $IO + NO_3 \rightarrow OIO + NO_2$ | $9.0 \times 10^{-12}$ | 20 |
| $IO + CH_3O_2 \rightarrow CH_2O + I + HO_2$ | $2.0 \times 10^{-12}$ | 2[h] |
| $CH_3I + OH \rightarrow I + H_2O + HO_2$ | $2.90 \times 10^{-12} e^{(-1100/T)}$ | 3 |
| $I + NO_2 (+ M) \rightarrow INO_2 (+ M)$ | $k_0 = 3 \times 10^{-31} \times (T / 300)^{-1}$
 $k_\infty = 6.6 \times 10^{-11}$ | 3[i] |
| $IO + NO_2 (+ M) \rightarrow IONO_2 (+ M)$ | $k_0 = 6.5 \times 10^{-31} \times (T / 300)^{-3.5}$
 $k_\infty = 7.6 \times 10^{-12} \times (T / 300)^{-1.5}$ | 3[i] |
| $I + NO (+ M) \rightarrow INO (+ M)$ | $k_0 = 1.8 \times 10^{-32} \times (T / 300)^{-1}$
 $k_\infty = 1.7 \times 10^{-11}$ | 3[i] |
| $OIO + OH (+ M) \rightarrow HOIO_2 (+ M)$ | $k_0 = 1.5 \times 10^{-27} \times (T / 300)^{-3.93}$
 $k_\infty = 7.76 \times 10^{-10} \times (T / 300)^{-0.8}$ | 14[j] |
| $HOI + NO_3 \rightarrow IO + HNO_3$ | $2.7 \times 10^{-12} (300/T)^{2.66}$ | 21 |

[1] IUPAC-2008 (Atkinson et al., 2007) ; [2](Dillon et al., 2006b); [3] JPL-2010 (Sander et al., 2011); [4](Gómez Martín et al., 2007); [5](Kaltsoyannis and Plane, 2008); [6](Galvez et al., 2013); [7](Bösch et al., 2003); [8] (Gómez Martín and Plane, 2009); [9](Chambers et al., 1992); [10](Chameides and Davis, 1980); [11](Dillon et al., 2006a); [12](McFiggans et al., 2000); [13](Jenkin et al., 1985); [14](Plane et al., 2006); [15](Allan and Plane, 2002); [16] (Lancar et al., 1991); [17](Laszlo et al., 1997); [18](Bedjanian et al., 1997); [19](Gilles et al., 1997); [20](Dillon et al., 2008); [21]This work.

[a]    w1= $4.687 \times 10^{-10} - 1.3855 \times 10^{-5} \times e^{(- 0.75\, p / 1.62265)} + 5.51868 \times 10^{-10} \times e^{(-0.75\, p / 199.328)}$

w2 $= -0.00331 - 0.00514 \times e^{(- 0.75\, p / 325.68711)} - 0.00444 \times e^{(- 0.75\, p / 40.81609)}$

[b]    w1 $= 1.1659 \times 10^{-9} - 7.79644 \times 10^{-10} e^{(-0.75\, p/22.09281)} + 1.03779 \times 10^{-9} \times e^{(-0.75\, p / 568.15381)}$

w2 $= -0.00813 - 0.00382 \times e^{(- 0.75\, p / 45.57591)} - 0.00643 \times e^{(-0.75\, p / 417.95061)}$

[c]    w1 $= 3.54288 \times 10^{10} + 1.8523 \times 10^{11} \times 0.75\, p - 1.45435 \times 10^8 \times (0.75p)^2 + 60799.4344 \times (0.75p)^3$
w2 $= -9681.65989 + 346.95538 \times e^{(-0.75\, p / 343.25322)} + 251.78032 \times e^{(-0.75\, p / 44.1466)}$

[d]    w1 $= 255335000000 - 4418880000 \times 0.75\, p + 85618600 \times (0.75\, p)^2 + 14218.81 \times (0.75\, p)^3$
w2 $= -11466.82304 + 597.01334 \times e^{(-0.75\, p / 1382.62325)} - 167.3391 \times e^{(-0.75\, p / 43.75089)}$

[e]    w1 $= -1.92626 \times 10^{14} + 4.67414 \times 10^{13} \times 0.75\, p - 3.68651 \times 10^8 \times (0.75\, p)^2 - 3.09109 \times 10^6 \times (0.75\, p)^3$
w2 $= -12302.15294 + 252.78367 \times e^{(-0.75\, p / 46.12733)} + 437.62868 \times e^{(-0.75\, p / 428.4413)}$

*f*      w1 = -2.63544 x $10^{13}$ + 4.32845 x $10^{12}$ x (0.75 p) + 3.73758 x $10^8$ x (0.75 p) $^2$ - 628468.76313 x (0.75 p) $^3$

      w2 = -13847.85015 + 240.34465 x e $^{(-0.75\ p\ /\ 49.27141)}$ + 451.35864 x e $^{(-0.75\ p\ /\ 436.87605)}$

*g*      The empirical expressions of the form $w_1$ · exp ( $w_2$ · T) were obtained by non-linear least squares fitting of *Rice–Ramsperger–Kassel–Marcus* (RRKM) theoretical results for the indicated reaction rate constants and thermal dissociation rates in the (27 – 1013) hPa pressure range. RRKM calculations were carried out using the MESMER algorithm (Glowacki et al., 2012) as indicated in the corresponding references (e.g. (Galvez et al., 2013). Expression *a* produces negative values outside the range of modelled rate constants (p < 20 hPa), and therefore a fixed rate constant of 3 x $10^{-11}$ $cm^3$ molecule $^{-1}$ s $^{-1}$ was assumed. Expressions *e* and *f* generate negligible dissociation rates below ~500 hPa which become negative at ~8 hPa – in this case they are set to zero below that pressure.

*h*      Updated heats of formation for IO, OIO, and $CH_3O_2$ (Dooley et al., 2008; Gómez Martín and Plane, 2009; Knyazev and Slagle, 1998) show that the only accessible exothermic product channel of $CH_3O_2$ + IO (Drougas and Kosmas, 2007) is $CH_2O$ + I + $O_2$ ($\Delta H_r$ = -5 ± 6 kJ $mol^{-1}$), consistent with the high yield of I and low yield of OIO found experimentally (Bale et al., 2005; Enami et al., 2006). Sensitivity studies have been carried out (Saiz-Lopez et al., 2014) using the preferred rate constant for this reaction of 2 × $10^{-12}$ $cm^3$ molecule $^{-1}$ s $^{-1}$ (Dillon et al., 2006b), resulting in an enhancement of the ozone loss of 0.5% in the MBL and of less than 0.1% integrated throughout the troposphere in the $J_{IxOy}$ scenario, and similarly negligible enhancements in the Base scenario. Impacts in the $I_y$ partitioning are also very minor.

*i*      The temperature and pressure dependent rate constant (k) is computed based on the low pressure ($k_0$) and the high-pressure ($k_\infty$) rate coefficients following JPL-2010 (Sander et al., 2011).

*j*      The Fast rate constants and a thermally stable product $HOIO_2$ have been predicted theoretically (Plane et al., 2006), but no experimental studies reporting observation of $HOIO_2$ and its photochemical properties in the gas phase are available. Since the level of uncertainty is even larger than for the $I_xO_y$, it has not been included in the mechanism.

Table 2. Iodine chemistry scheme in CAM-Chem: Photochemical reactions.

| Reaction |
| --- |
| $CH_3I + h\nu \rightarrow CH_3O_2 + I$ |
| $CH_2I_2 + h\nu \rightarrow 2I$ [a] |
| $CH_2IBr + h\nu \rightarrow Br + I$ [a] |
| $CH_2ICl + h\nu \rightarrow Cl + I$ [a] |
| $I_2 + h\nu \rightarrow 2I$ |
| $IO + h\nu \rightarrow I + O$ |
| $OIO + h\nu \rightarrow I + O_2$ |
| $INO + h\nu \rightarrow I + NO$ |
| $INO_2 + h\nu \rightarrow I + NO_2$ [b] |
| $IONO_2 + h\nu \rightarrow I + NO_3$ |
| $HOI + h\nu \rightarrow I + OH$ |
| $IBr + h\nu \rightarrow I + Br$ |
| $ICl + h\nu \rightarrow I + Cl$ |
| $I_2O_2 + h\nu \rightarrow I + OIO$ [c] |
| $I_2O_3 + h\nu \rightarrow IO + OIO$ [c] |
| $I_2O_4 + h\nu \rightarrow OIO + OIO$ [c] |

Photolysis rates are computed online considering the actinic flux calculation in CAM-Chem. The absorption cross-sections and quantum yields for all species besides the $I_xO_y$ have been taken from IUPAC-2008 (Atkinson et al., 2007; Atkinson et al., 2008) and JPL-2010 (Sander et al., 2011).

[a] radical organic products are not considered.
[b] only the reaction channel reported in JPL 06-02 (Sander et al., 2006) is considered.
[c] photolysis reactions only considered in the $J_{IxOy}$ scheme (Saiz-Lopez et al., 2014).

Table 3. Iodine chemistry scheme in CAM-Chem: Heterogeneous reactions.

| Sea-salt aerosol reactions | Reactive uptake |
|---|---|
| $IONO_2 \rightarrow 0.5\ IBr + 0.5\ ICl$ | $\gamma = 0.01$ |
| $INO_2 \rightarrow 0.5\ IBr + 0.5\ ICl$ | $\gamma = 0.02$ |
| $HOI \rightarrow 0.5\ IBr + 0.5\ ICl$ | $\gamma = 0.06$ |
| $I_2O_2 \rightarrow$ | $\gamma = 0.01$ [§] |
| $I_2O_3 \rightarrow$ | $\gamma = 0.01$ [§] |
| $I_2O_4 \rightarrow$ | $\gamma = 0.01$ [§] |

Values based on the THAMO model (Saiz-Lopez et al., 2008) and implemented in CAM-Chem following (Ordóñez et al., 2012).

[§] Deposition of $I_xO_y$ species on sea-salt aerosols has been included following the free regime approximation.

Table 4. Iodine chemistry scheme in CAM-Chem: Henry's Law constants and dry deposition velocities.

| Species | $k_0$ (M atm$^{-1}$) | Deposition velocity[§] (cm s$^{-1}$) | Reference |
|---|---|---|---|
| IBr [ice] | $2.4 \times 10^1$ | – | 1 |
| ICl [ice] | $1.1 \times 10^2$ | – | 1 |
| HI | $7.8 \times 10^{-1}$ | 1.0 | 1 [a] |
| HOI – ($J_{IxOy}$ / $Base$) | $1.9 \times 10^3$ / $4.5 \times 10^3$ | 0.75 | 1 [b] |
| IONO$_2$ [ice] | $1.0 \times 10^6$ | 0.75 | 2 [c] |
| INO$_2$ [ice] | $3.0 \times 10^{-1}$ | 0.75 | 1 [d] |
| IO | $4.5 \times 10^2$ | – | 2 |
| OIO | $1.0 \times 10^4$ | – | 2 |
| I$_2$O$_2$ | $1.0 \times 10^4$ | 1.0 | 2 |
| I$_2$O$_3$ | $1.0 \times 10^4$ | 1.0 | 2 |
| I$_2$O$_4$ | $1.0 \times 10^4$ | 1.0 | 2 |

[§] Dry deposition velocities are based on the THAMO model (Saiz-Lopez et al., 2008).
[1] Values reported in (Sander, 1999).
[2] Values based on the THAMO model (Saiz-Lopez et al., 2008).
[a] Considering a dissociation constant $K_a = 3.2 \times 10^9$ and a temperature dependent coefficient c = 9800 K
[b] Within the range of values given in the corresponding reference.
[c] Virtually infinite solubility is represented by using a very large arbitrary number.
[d] Value assumed to be equal to those of BrNO$_2$.
[ice] Species for which ice-uptake is considered following (Neu and Prather, 2012).

---

## Author Response (AR2)

**Co-Editor Decision: Reconsider after major revisions (23 Sep 2016) by Dr. Jens-Uwe Grooß**

We thank Dr. Grooß for the insightful comments. Below we provide a detailed point-by-point answer (AC – Author Comment) to each comment on our manuscript (CEC – Co-Editor Comment).

**CEC:**

**Comments to the Author:**

**Dear authors,**

**In my impression, you should have considered more of the reviewers comments. So please consider the remaining points expressed by the two reviewers as well as the following technical points:**

**Title: "after dark" means to my knowledge something like "after it did get dark". It could be mis-understood as "after the dark period", which would be at morning. You also mention the peak at sunrise, so I also would prefer to be the title as suggested by reviewer #1 ("Nighttime Iodine chemistry" or so.)**

AC:

      We have changed the title to "Nighttime atmospheric chemistry of iodine".

**CEC:**

**"buildup of HOI"**

**In your figure 4 (HOI scenario 1) HOI is clearly decreasing during night and it is not building up, although the shown HOI emissions are increasing. You could likely visualize the nighttime values better, if the used colour label differences would not be linear, but e.g. logarithmic. But in the text should be clarified, that part of the emitted HOI has a faster or slower sink depending on scenario.**

AC:

We have corrected that statement in the manuscript (page 13, lines from 6 to 9)

**CEC:**
**"which is converted into I2/IBr/ICl through heterogeneous recycling" (2 times) What do you mean by that? According to your reaction list in the attachment there the HOI would react to IBr/ICl on the aerosols but not to I2.**

AC:

Thanks!. You are right; we have corrected it and replaced "I$_2$/IBr/ICl" by "IBr/ICl".

**CEC:**

**"Figures 4 and 5" please use "local time" a x-coordinate as suggested by the reviewer and mention the two-day spin-up elsewhere.**

AC:

We have replaced "Time" by "Local time" in x-axis labels, and mentioned "after two days of simulation time" in the text (P13, L2 and P13, L18).

**CEC:**
**"Figure 4 middle right" please do not truncate the colour labels of the contour plot.**
AC:

Labels have been replaced by a color scale.

**CEC:**

**"Figure 5": please use also IONO2 instead of INO3 in the bottom panels to be consistent with the text.**

AC:

corrected

**CEC:**
**"Figure 6": please explain in more detail which NO3 value is plotted (certain local time, average nighttime or what).**

AC:

We now mention in the text (P14, L16) and in the caption of figure 6 that it refers to NO$_3$ peak nighttime concentration.

**CEC:**
**"figure 7-9, colour bar label" use (greek) Delta HOI and Delta I2 as you show a difference.**
AC:

Thank you for the suggestion. We have now included the Delta in the label of the three figures.

**Non-public comments to the Author:**

**Dear authors,**

**thank you for submitting the revised version. Since I am not an expert in modelling at the molecular level, I asked Florent Louis and one other reviewer to look, whether their comments have been answered sufficiently. As you see, they still have some points, which I would ask you to consider.**

**regards, Jens-Uwe Grooß**

**Anonymous Referee #1**

We thank the reviewer for his/her insightful comments. Below we provide a detailed point-by-point answer (AC – Author Comment) to each comment on our manuscript (RC – Reviewer Comment).

**Suggestions for revision or reasons for rejection (will be published if the paper is accepted for final publication)**

**In my opinion, the scientific content of the revised version of the manuscript is sufficient to justify publication in ACP. Unfortunately, however, the authors ignored several suggestions from the reviewers to improve the presentation quality which is also important for a high-quality article. In particular, I would like to reiterate the following points:**

**RC:**

**Thank you for showing the iodine reaction mechanism in the supplement. You mention that the scheme is referenced in page 22 line 12 in the main text but I cannot find it there. Although the supplement now shows all iodine reactions, the non-iodine reactions in the model are still not listed. For example, IO + BrO is listed but BrO + BrO is not listed, even though this reaction will also have an indirect effect on reactive iodine concentrations.**

AC:

You are correct, supplementary information is referenced in page 5 line 10. Now, we have also cited, in the same page and line, the work of Ordoñez et al., 2012, which includes all iodine and non-iodine reactions included in the model. We think that it would be redundant to include the same supplement that has already been included in Ordoñez et al., 2012.

**RC:**

**I know that LT refers to local time. However, this may not be clear to all readers. I think that all acronyms should be explained in the text when they are used for the first time.**

AC:

Acronym LT (Local Time) has now been defined in Page 15, line 1.

**RC:**

**It may be customary to quote vibrational frequencies as wave numbers but it is incorrect. It is like saying "the frequency of green light is 500 nm".**

AC:

Although it may be incorrect, this is the conventional units for vibrational frequency which is accepted by the community. So we respectfully prefer to keep table 2 in its current form.

**RC:**

**Provided that enough significant digits are shown, it is fine to show the numbers in the plots of Figs. 4 and 5 instead of using a color scale. Nevertheless, I still find it confusing if two plots in the same row use slightly different color schemes. It makes a comparison unnecessarily complicated.**

AC:

Thanks for the suggestion. We have now replaced figures 4 and 5 by new ones including the same colour scale.

**RC:**

**The information that "nighttime" refers to the average between midnight and 01:00 local time is important and should be shown in the figure caption. If you want to avoid repetition, it could be removed from the text.**

AC:

(from 0:00 to 1:00 LT) has been included in the figure captions.

**RC:**

**The effect of the new chemistry on DMS is quite indirect. You mention that DMS changes because of NO3. In addition, the new iodine chemistry will most probably also affect bromine chemistry, and different BrO will also affect DMS concentrations. I think that either this should also be discussed, or figure 9 should be removed, as another reviewer had suggested.**

AC:

The effect of this iodine chemistry on bromine is negligible, so we would expect a direct effect on DMS through $NO_3$. Therefore we respectfully prefer to keep figure 9 in the manuscript.

**Referee #4: LOUIS, Florent**
We thank Dr. Louis for the insightful comments. Below we provide a detailed point-by-point answer (AC – Author Comment) to each comment on our manuscript (RC – Referee Comment).

**RC:**

**I posted an interactive comment on "Iodine chemistry after dark" by Saiz-Lopez et al. on July 4th 2016. This paper represents a very important contribution dealing with atmospheric iodine chemistry. I made several comments. There is still one comment for which the authors do not take into account my recommendation.**

**It concerns the spin-orbit correction (SOC) for iodine-containing species involved in the studied reactions. I previously reported "The authors stated on page 6 line 12 that "spin-orbit splittings of -17 and -5 kJ mol-1 were applied to energies of I and IO". These values do not correspond to the well known value for I atom (-30.3 kJ mol-1 from C.E. Moore, Atomic Energy Levels, USGPO, Vols. II and III. NSRDS-NBS 35, Washington, DC, 1971). Over the last years, my group performed theoretical calculations to get the SOC values for numerous iodine-containing species using the CASPT2/RASSI methodology. The corresponding values for I, IO, and HOI are -30.0, -14.4, and -5.9 kJ/mol (Meciarova et al., CPL, 2011, 517, 149; Khanniche et al., JPCA, 2016, 120, 1737; Sulkova et al., JPCA, 2013, 117, 771). These calculations were also validated by comparison to few available data. I recommend the authors to update their energetics according to the correct SOC values".**

**There are large differences between SOC values for I and IO (-17 and -5 kJ mol-1) in the manuscript and the literature data (-30 and -14 kJ mol-1). I would expect the authors will revise their energetics according to the most reliable data. Kinetic parameters should be also re-evaluated.**

AC:

We have followed the reviewer's suggestion and included these recent spin-orbit corrections in the calculated reaction energetics of the three reactions. For reaction 4, the estimated rate coefficient remains unchanged, although $k_4$ is possibly a lower limit because of the increasing spin-orbit correction across the potential surface from HOI to IO. This is now discussed in the following changes in Section 3 of the paper (page 6, line 11 to page 7, line 11):

"Spin-orbit corrections of -30.0 (Mečiarová et al., 2011), -14.4 (Khanniche et al., 2016), -5.9 (Šulková et al., 2013) and -4.8 (Kaltsoyannis and Plane, 2008) kJ mol$^{-1}$ were applied to the energies of I , IO, HOI and IONO$_2$, respectively.

Reaction 2 is endothermic by 2.6 kJ mol$^{-1}$ and so, within the expected error of 10 kJ mol$^{-1}$ at this level of theory, might be reasonably fast. However, the transition state of the reaction, which is illustrated in Figure 2(a), is 73 kJ mol$^{-1}$ above the reactants and so this reaction will not occur at tropospheric temperatures. Reaction 3 is exothermic by 19.8 kJ mol$^{-1}$. An HOI--HNO$_3$ complex first forms (Figure 2(b)), which is 21 kJ mol$^{-1}$ below the reactants. However, this complex re-arranges to the IONO$_2$ + H$_2$O products via the cyclic transition state shown in Figure 2(c), which is 110 kJ mol$^{-1}$ above the reactants.

The stationary points on the potential energy surface (PES) for reaction 4 are illustrated in Figure 3. HOI and NO$_3$ associate to form a complex which is 24 kJ mol$^{-1}$ below the reactant entrance channel. H-atom transfer involves a submerged transition state to form an IO--HNO$_3$ complex, which can then dissociate to the products IO + HNO$_3$. The vibrational frequencies, rotational energies and geometries (in Cartesian co-ordinates) of these intermediates are listed in Table 2. Overall, the reaction is exothermic by 14 kJ mol$^{-1}$. The energies of the HOI--NO$_3$ complex and the transition state are assigned the same spin-orbit correction as HOI (-5.9 kJ mol$^{-1}$ (Šulková et al., 2013)), whereas the IO--HNO$_3$ complex is assigned the spin-orbit correction of IO (-14.4 kJ mol$^{-1}$ (Khanniche et al., 2016)). This reflects the H-OI bond only increasing from 0.97 Å in HOI to 1.1 Å in the transition state, compared with 1.7 Å in the IO—HNO$_3$ complex. The spin-orbit correction for the transition state is therefore likely to be closer to that of HOI. Assigning the HOI spin-orbit correction therefore means that the barrier is highest with respect to the reactants, so that the estimated rate coefficient (see below) may be a lower limit."

page 8, line 23 to page 9, line 10:

"The uncertainty in $k_4$ arises principally from the estimated capture rate coefficient (see above), and the height of the barrier below the entrance channel. As discussed above, the spin-orbit correction of the transition state is likely to be larger than the value of -5.9 kJ mol$^{-1}$ corresponding to HOI, so $k_4$ is possibly a lower limit. For instance, if the barrier height is decreased by 3 kJ mol$^{-1}$, $k_4$ increases by a factor of 1.9. If the barrier is lower by 8.5 kJ mol$^{-1}$ (corresponding to the transition state having the same spin-orbit correction as IO), then $k_4$ would increase by a factor of 5.1. Nevertheless, noting that the capture rate coefficient could be lower – perhaps by a factor of 2 - than the estimate used here, we prefer to use the value for $k_4$ calculated with the potential surface in Figure 3. Of course, if $k_4$ is larger, then the atmospheric impacts of reaction 4 discussed in Section 4 will be even more pronounced."

**RC:**
**For the second comment dealing with the treatment of low frequency modes in the intermediate species, I would suggest the authors to mention only in the**

**manuscript : "The rigid-rotor harmonic oscillator approximation has been used for all species". Hindered rotor treatment for low frequency modes could influence the entropy contribution. Of course, it will depend on the rotational barrier height, its periodicity and the model used. It is often used without any available experimental                                                                  data.**
AC:

We have adopted this suggestion (page 7, line 23).

**RC:**
*Other comments have been properly answered by the authors.*

**Nighttime atmospheric chemistry of iodine** <s>after dark</s>

[revised manuscript text omitted]

 H 1.484,-0.657,-0.043
 I  0.009,1.205,0.286
 N -0.456,-2.265,0.030
 O -1.052, -3.321,-0.0473
 O -1.147,-1.195,-0.228
 O 0.742,-2.161,0.333 | 55, 84, 118,   161, 196, 615, 629, 667, 705, 803, 968,   1228,   1273,1491, 3268 | 5.610, 0.916, 0.806 | -24.0 |
| IO-H-NO$_2$ TS | O 0.309,1.515,0.247
 H -0.834,1.314,-0.017
 I  1.280,-0.089,-0.093
 N -2.349,-0.133,0.019
 O -3.518, ,-0.429,-0.035
 O -1.444,-0.962,0.257
 O -2.019,1.117,-0.187 | 1249$i$, 70, 97, 103,   225, 472, 676, 698, 797, 806, 1041, 1147, 1308, 1513, 1626 | 6.300, 0.864, 0.767 | -16.4 |
| IO-HNO$_3$ complex | O 0.571,1.350,0.348
 H -1.111,1.098,-0.020
 I 1.870,0.0645,-0.152
 N -2.503,-0.202,0.0186
 O -3.673,-0.396,-0.170
 O -1.654,-0.986,0.401
 O -2.081,1.090,-0.242 | 35, 43, 76, 126, 198, 623, 677, 703, 772, 798, 939, 1331, 1416, 1713, 3281 | 7.058, 0.605, 0.566 | -34.8 |
| IO + HNO$_3$ | | 648 & 477, 585, 649, 782, 901, 1320, 1345, 1738, 3724 | 9.844 & 13.01, 12.05, 6.258 | -10.6 |

[a] Cartesian co-ordinates in Å.  [b] In cm$^{-1}$.  [c] In GHz.  [d] In kJ mol$^{-1}$, including zero-point energy and spin-
orbit coupling of I and IO (see text).

[Figure]

**Figure 1**. New nocturnal iodine chemistry (in white) implemented in the THAMO and CAM-Chem models.

[Figure]

**Figure 2:** (a) Transition state for the reaction between HOI and $NO_2$ to form $HNO_3$ + I; (b) complex formed between HOI and $HNO_3$, which then reacts via transition state (c) to form $IONO2$ + $H_2O$.

[Figure]

**Figure 3**. Potential energy surface for the reaction between HOI and $NO_3$, which contains two
intermediate complexes separated by a submerged barrier.

[Figure]

**Figure 4.** THAMO modeled diurnal variation of HOI, $I_2$ (upper panels) and the HOI/$I_2$ flux from the ocean surface (bottom panel). The right hand panels are from scenario 1, which do not include night time reactions of HOI and $I_2$ with $NO_3$, while the left hand panels include the reactions in scenario 2. In bottom panel red lines represent scenario 1, while black lines correspond to scenario 2.

[Figure]

**Figure 5.** THAMO modeled diurnal variation of IO, NO₃ and the IONO₂. The right hand panels
are from scenario 1, which do not include night time reactions of HOI and I₂ with NO₃, while the
left hand panels include the reactions in scenario 2.

[Figure]

**Figure 6**. Sensitivity run showing the effect of the uncertainty in the rate constant estimation on the reduction of $NO_3$ peak nighttime concentration at the surface - the red point is the theoretical estimate.

[Figure]

[Figure]

**Figure 7.** Modelled annual average of HOI (a) and I₂ (b) during night time (from 0:00 to 1:00 LT) at the surface level. The panels show the difference in volume mixing ratio between the simulations with and without reactions (1) and (4).

[Figure]

[Figure]

**Figure 8.** Modelled annual average of IONO₂ (a) and NO₃ (b) during night time (from 0:00 to 1:00 LT) at the surface level, as the difference in volume mixing ratio between the simulations with and without reactions (1) and (4).

[Figure]

**Figure 9**. Increase in the DMS levels during night time (from 0:00 to 1:00 LT) at the surface
level due to the inclusion of the reactions R1 and R4 in CAM-Chem.

[Figure]

**Figure 10**. Hourly averaged concentration of HOI, IONO$_2$ and I$_2$ in the Mediterranean Sea at the
surface level (lon:10º→20ºE, lat:33º→40ºN)

[Figure]

Figure 11. Hourly averaged concentration of HOI, IONO$_2$ and I$_2$ (upper panel) and NO$_3$ (bottom panel) in the Pacific Ocean at the south of Baja California peninsula at the surface level (lon: -110º→-106ºE, lat:16º→23ºN)

Supplementary information for

**Nighttime atmospheric chemistry of iodine**

Alfonso Saiz-Lopez[1], John M.C. Plane[2], Carlos A. Cuevas[1], Anoop S. Mahajan[3], Jean-François Lamarque[4] and Douglas E. Kinnison[4]

[1]Department of Atmospheric Chemistry and Climate, Institute of Physical Chemistry Rocasolano, CSIC, Madrid, Spain

[2]School of Chemistry, University of Leeds, Leeds, UK

[3]Indian Institute of Tropical Meteorology, Pune, India

[4]Atmospheric Chemistry Observations and Modelling, NCAR, Colorado, USA

Correspondence to: A. Saiz-Lopez (a.saiz@csic.es)

Table 1. Iodine chemistry scheme in CAM-Chem: Bimolecular, thermal decomposition and termolecular reactions.

| Reaction | $k$ / $cm^3$ $molecule^{-1}$ $s^{-1}$ | Notes |
|---|---|---|
| $I + O_3 \rightarrow IO + O_2$ | $2.1 \times 10^{-11}$ $e^{(-830/T)}$ | 1 |
| $IO + O_3 \rightarrow OIO + O_2$ | $3.6 \times 10^{-16}$ | 2 |
| $I + HO_2 \rightarrow HI + O_2$ | $1.5 \times 10^{-11}$ $e^{(-1090/T)}$ | 3 |
| $IO + NO \rightarrow I + NO_2$ | $7.15 \times 10^{-12}$ $e^{(300/T)}$ | 1 |
| $IO + HO_2 \rightarrow HOI + O_2$ | $1.4 \times 10^{-11}$ $e^{(540/T)}$ | 1 |
| $IO + IO \rightarrow OIO + I$ | $2.13 \times 10^{-11}$ $e^{(180/T)} \times [1 + e^{(-p/191.42)}]$ | 1, 4 |
| $IO + IO \rightarrow I_2O_2$ | $3.27 \times 10^{-11}$ $e^{(180/T)} \times [1 - 0.65 e^{(-p/191.42)}]$ | 1, 4 |
| $IO + OIO \rightarrow I_2O_3$ | $w_1 \cdot \exp(w_2 \cdot T)$ [a] | 4, 5, 6 [g] |
| $OIO + OIO \rightarrow I_2O_4$ | $w_1 \cdot \exp(w_2 \cdot T)$ [b] | 4, 5, 6 [g] |
| $I_2 + O \rightarrow IO + I$ | $1.25 \times 10^{-10}$ | 1 |
| $IO + O \rightarrow I + O_2$ | $1.4 \times 10^{-10}$ | 1 |
| $IO + OH \rightarrow HO_2 + I$ | $1.0 \times 10^{-10}$ | 7 |
| $I_2O_2 \rightarrow OIO + I$ | $w_1 \cdot \exp(w_2/T)$ [c] | 5, 6, 8 [g] |
| $I_2O_2 \rightarrow IO + IO$ | $w_1 \cdot \exp(w_2/T)$ [d] | 5, 6, 8 [g] |
| $I_2O_4 \rightarrow 2\ OIO$ | $w_1 \cdot \exp(w_2/T)$ [e] | 5, 8 [g] |
| $I_2 + OH \rightarrow HOI + I$ | $1.8 \times 10^{-10}$ | 3 |
| $I_2 + NO_3 \rightarrow I + IONO_2$ | $1.5 \times 10^{-12}$ | 9 |
| $I + NO_3 \rightarrow IO + NO_2$ | $1.0 \times 10^{-10}$ | 1 |
| $OH + HI \rightarrow I + H_2O$ | $1.6 \times 10^{-11}$ $e^{(440/T)}$ | 1 |
| $I + IONO_2 \rightarrow I_2 + NO_3$ | $9.1 \times 10^{-11}$ $e^{(-146/T)}$ | 5 |
| $HOI + OH \rightarrow IO + H_2O$ | $2.0 \times 10^{-13}$ | 10 |
| $IO + DMS \rightarrow DMSO + I$ | $3.2 \times 10^{-13}$ $e^{(-925/T)}$ | 11 |
| $INO_2 \rightarrow I + NO_2$ | $1008 \times 10^{15}$ $e^{(-13670/T)}$ | 12, 13, 14 |
| $IONO_2 \rightarrow IO + NO_2$ | $w_1 \cdot \exp(w_2/T)$ [f] | 5, 15 |
| $INO + INO \rightarrow I_2 + 2NO$ | $8.4 \times 10^{-11}$ $e^{(-2620/T)}$ | 3 |
| $INO_2 + INO_2 \rightarrow I_2 + 2NO_2$ | $4.7 \times 10^{-13}$ $e^{(-1670/T)}$ | 1 |
| $OIO + NO \rightarrow IO + NO_2$ | $1.1 \times 10^{-12}$ $e^{(542/T)}$ | 14 |
| $HI + NO_3 \rightarrow I + HNO_3$ | $1.3 \times 10^{-12}$ $e^{(-1830/T)}$ | 16 |
| $IO + BrO \rightarrow Br + I + O_2$ | $0.30 \times 10^{-11}$ $e^{(510/T)}$ | 1 |
| $IO + BrO \rightarrow Br + OIO$ | $1.20 \times 10^{-11}$ $e^{(510/T)}$ | 1 |
| $I + BrO \rightarrow IO + Br$ | $1.44 \times 10^{-11}$ | 17, 18, 19 |

| | | |
|---|---|---|
| $IO + ClO \rightarrow I + OClO$ | $2.585 \times 10^{-12}\ e^{(280/T)}$ | 1 |
| $IO + ClO \rightarrow I + Cl + O_2$ | $1.175 \times 10^{-12}\ e^{(280/T)}$ | 1 |
| $IO + ClO \rightarrow ICl + O_2$ | $0.940 \times 10^{-12}\ e^{(280/T)}$ | 1 |
| $IO + Br \rightarrow I + BrO$ | $2.49 \times 10^{-11}$ | 18, 19 |
| $IO + NO_3 \rightarrow OIO + NO_2$ | $9.0 \times 10^{-12}$ | 20 |
| $IO + CH_3O_2 \rightarrow CH_2O + I + HO_2$ | $2.0 \times 10^{-12}$ | 2[h] |
| $CH_3I + OH \rightarrow I + H_2O + HO_2$ | $2.90 \times 10^{-12}\ e^{(-1100/T)}$ | 3 |
| $I + NO_2\ (+ M) \rightarrow INO_2\ (+ M)$ | $k_0 = 3 \times 10^{-31} \times (T / 300)^{-1}$
 $k_\infty = 6.6 \times 10^{-11}$ | 3[i] |
| $IO + NO_2\ (+ M) \rightarrow IONO_2\ (+ M)$ | $k_0 = 6.5 \times 10^{-31} \times (T / 300)^{-3.5}$
 $k_\infty = 7.6 \times 10^{-12} \times (T / 300)^{-1.5}$ | 3[i] |
| $I + NO\ (+ M) \rightarrow INO\ (+ M)$ | $k_0 = 1.8 \times 10^{-32} \times (T / 300)^{-1}$
 $k_\infty = 1.7 \times 10^{-11}$ | 3[i] |
| $OIO + OH\ (+ M) \rightarrow HOIO_2\ (+ M)$ | $k_0 = 1.5 \times 10^{-27} \times (T / 300)^{-3.93}$
 $k_\infty = 7.76 \times 10^{-10} \times (T / 300)^{-0.8}$ | 14[j] |
| $HOI + NO_3 \rightarrow IO + HNO_3$ | $2.7 \times 10^{-12}\ (300/T)^{2.66}$ | 21 |

[1] IUPAC-2008 (Atkinson et al., 2007) ; [2](Dillon et al., 2006b); [3] JPL-2010 (Sander et al., 2011); [4](Gómez Martín et al., 2007); [5](Kaltsoyannis and Plane, 2008); [6](Galvez et al., 2013); [7](Bösch et al., 2003); [8] (Gómez Martín and Plane, 2009); [9](Chambers et al., 1992); [10](Chameides and Davis, 1980); [11](Dillon et al., 2006a); [12](McFiggans et al., 2000); [13](Jenkin et al., 1985); [14](Plane et al., 2006); [15](Allan and Plane, 2002); [16] (Lancar et al., 1991); [17](Laszlo et al., 1997); [18](Bedjanian et al., 1997); [19](Gilles et al., 1997); [20](Dillon et al., 2008); [21]This work.

[a]  $w1= 4.687\ x\ 10^{-10} - 1.3855\ x\ 10^{-5}\ x\ e^{(-0.75\ p\ /\ 1.62265)} + 5.51868\ x\ 10^{-10}\ x\ e^{(-0.75\ p\ /\ 199.328)}$

  $w2 = -0.00331 - 0.00514\ x\ e^{(-0.75\ p\ /\ 325.68711)} - 0.00444\ x\ e^{(-0.75\ p\ /\ 40.81609)}$

[b]  $w1 = 1.1659\ x\ 10^{-9} - 7.79644\ x\ 10^{-10}\ e^{(-0.75\ p/22.09281)} + 1.03779\ x\ 10^{-9}\ x\ e^{(-0.75\ p\ /\ 568.15381)}$

  $w2 = -0.00813 - 0.00382\ x\ e^{(-0.75\ p\ /\ 45.57591)} - 0.00643\ x\ e^{(-0.75\ p\ /\ 417.95061)}$

[c]  $w1 = 3.54288\ x\ 10^{10} + 1.8523\ x\ 10^{11}\ x\ 0.75\ p\ -1.45435\ x\ 10^{8}\ x\ (0.75p)^{2} + 60799.4344\ x\ (0.75p)^{3}$
  $w2 = -9681.65989 + 346.95538\ x\ e^{(-0.75\ p\ /\ 343.25322)} + 251.78032\ x\ e^{(-0.75\ p\ /\ 44.1466)}$

[d]  $w1 = 255335000000 - 4418880000\ x\ 0.75\ p + 85618600\ x\ (0.75\ p)^{2} + 14218.81\ x\ (0.75\ p)^{3}$
  $w2 = -11466.82304 + 597.01334\ x\ e^{(-0.75\ p\ /\ 1382.62325)} - 167.3391\ x\ e^{(-0.75\ p\ /\ 43.75089)}$

[e]  $w1= -1.92626\ x\ 10^{14} + 4.67414\ x\ 10^{13}\ x\ 0.75\ p - 3.68651\ x\ 10^{8}\ x\ (0.75\ p)^{2} - 3.09109\ x\ 10^{6}\ x\ (0.75\ p)^{3}$
  $w2 = -12302.15294 + 252.78367\ x\ e^{(-0.75\ p\ /\ 46.12733)} + 437.62868\ x\ e^{(-0.75\ p\ /\ 428.4413)}$

$f$     $w1 = -2.63544 \times 10^{13} + 4.32845 \times 10^{12} \times (0.75\ p) + 3.73758 \times 10^8 \times (0.75\ p)^2 - 628468.76313 \times (0.75\ p)^3$

$w2 = -13847.85015 + 240.34465 \times e^{(-0.75\ p\ /\ 49.27141)} + 451.35864 \times e^{(-0.75\ p\ /\ 436.87605)}$

$g$     The empirical expressions of the form $w_1 \cdot \exp(\ w_2 \cdot T)$ were obtained by non-linear least squares fitting of *Rice–Ramsperger–Kassel–Marcus* (RRKM) theoretical results for the indicated reaction rate constants and thermal dissociation rates in the (27 – 1013) hPa pressure range. RRKM calculations were carried out using the MESMER algorithm (Glowacki et al., 2012) as indicated in the corresponding references (e.g. (Galvez et al., 2013). Expression $a$ produces negative values outside the range of modelled rate constants (p < 20 hPa), and therefore a fixed rate constant of $3 \times 10^{-11}$ cm$^3$ molecule$^{-1}$ s$^{-1}$ was assumed. Expressions $e$ and $f$ generate negligible dissociation rates below ~500 hPa which become negative at ~8 hPa – in this case they are set to zero below that pressure.

$h$     Updated heats of formation for IO, OIO, and $CH_3O_2$ (Dooley et al., 2008; Gómez Martín and Plane, 2009; Knyazev and Slagle, 1998) show that the only accessible exothermic product channel of $CH_3O_2 + IO$ (Drougas and Kosmas, 2007) is $CH_2O + I + O_2$ ($\Delta H_r = -5 \pm 6$ kJ mol$^{-1}$), consistent with the high yield of I and low yield of OIO found experimentally (Bale et al., 2005; Enami et al., 2006). Sensitivity studies have been carried out (Saiz-Lopez et al., 2014) using the preferred rate constant for this reaction of $2 \times 10^{-12}$ cm$^3$ molecule$^{-1}$ s$^{-1}$ (Dillon et al., 2006b), resulting in an enhancement of the ozone loss of 0.5% in the MBL and of less than 0.1% integrated throughout the troposphere in the $J_{IxOy}$ scenario, and similarly negligible enhancements in the Base scenario. Impacts in the $I_y$ partitioning are also very minor.

$i$     The temperature and pressure dependent rate constant (k) is computed based on the low pressure ($k_0$) and the high-pressure ($k_\infty$) rate coefficients following JPL-2010 (Sander et al., 2011).

$j$     The Fast rate constants and a thermally stable product $HOIO_2$ have been predicted theoretically (Plane et al., 2006), but no experimental studies reporting observation of $HOIO_2$ and its photochemical properties in the gas phase are available. Since the level of uncertainty is even larger than for the $I_xO_y$, it has not been included in the mechanism.

Table 2. Iodine chemistry scheme in CAM-Chem: Photochemical reactions.

| Reaction |
| --- |
| $CH_3I + h\nu \rightarrow CH_3O_2 + I$ |
| $CH_2I_2 + h\nu \rightarrow 2I$ [a] |
| $CH_2IBr + h\nu \rightarrow Br + I$ [a] |
| $CH_2ICl + h\nu \rightarrow Cl + I$ [a] |
| $I_2 + h\nu \rightarrow 2I$ |
| $IO + h\nu \rightarrow I + O$ |
| $OIO + h\nu \rightarrow I + O_2$ |
| $INO + h\nu \rightarrow I + NO$ |
| $INO_2 + h\nu \rightarrow I + NO_2$ [b] |
| $IONO_2 + h\nu \rightarrow I + NO_3$ |
| $HOI + h\nu \rightarrow I + OH$ |
| $IBr + h\nu \rightarrow I + Br$ |
| $ICl + h\nu \rightarrow I + Cl$ |
| $I_2O_2 + h\nu \rightarrow I + OIO$ [c] |
| $I_2O_3 + h\nu \rightarrow IO + OIO$ [c] |
| $I_2O_4 + h\nu \rightarrow OIO + OIO$ [c] |

Photolysis rates are computed online considering the actinic flux calculation in CAM-Chem. The absorption cross-sections and quantum yields for all species besides the $I_xO_y$ have been taken from IUPAC-2008 (Atkinson et al., 2007; Atkinson et al., 2008) and JPL-2010 (Sander et al., 2011).

[a] radical organic products are not considered.
[b] only the reaction channel reported in JPL 06-02 (Sander et al., 2006) is considered.
[c] photolysis reactions only considered in the $J_{IxOy}$ scheme (Saiz-Lopez et al., 2014).

Table 3. Iodine chemistry scheme in CAM-Chem: Heterogeneous reactions.

| Sea-salt aerosol reactions | Reactive uptake |
|---|---|
| $IONO_2 \rightarrow 0.5\ IBr + 0.5\ ICl$ | $\gamma = 0.01$ |
| $INO_2 \rightarrow 0.5\ IBr + 0.5\ ICl$ | $\gamma = 0.02$ |
| $HOI \rightarrow 0.5\ IBr + 0.5\ ICl$ | $\gamma = 0.06$ |
| $I_2O_2 \rightarrow$ | $\gamma = 0.01$[§] |
| $I_2O_3 \rightarrow$ | $\gamma = 0.01$[§] |
| $I_2O_4 \rightarrow$ | $\gamma = 0.01$[§] |

Values based on the THAMO model (Saiz-Lopez et al., 2008) and implemented in CAM-Chem following (Ordóñez et al., 2012).

[§] Deposition of $I_xO_y$ species on sea-salt aerosols has been included following the free regime approximation.

Table 4. Iodine chemistry scheme in CAM-Chem: Henry's Law constants and dry deposition velocities.

| Species | $k_0$ (M atm$^{-1}$) | Deposition velocity$^§$ (cm s$^{-1}$) | Reference |
|---|---|---|---|
| IBr [ice] | $2.4 \times 10^1$ | – | 1 |
| ICl [ice] | $1.1 \times 10^2$ | – | 1 |
| HI | $7.8 \times 10^{-1}$ | 1.0 | 1 [a] |
| HOI – ($J_{IxOy}$ / Base) | $1.9 \times 10^3$ / $4.5 \times 10^3$ | 0.75 | 1 [b] |
| IONO$_2$ [ice] | $1.0 \times 10^6$ | 0.75 | 2 [c] |
| INO$_2$ [ice] | $3.0 \times 10^{-1}$ | 0.75 | 1 [d] |
| IO | $4.5 \times 10^2$ | – | 2 |
| OIO | $1.0 \times 10^4$ | – | 2 |
| I$_2$O$_2$ | $1.0 \times 10^4$ | 1.0 | 2 |
| I$_2$O$_3$ | $1.0 \times 10^4$ | 1.0 | 2 |
| I$_2$O$_4$ | $1.0 \times 10^4$ | 1.0 | 2 |

$^§$ Dry deposition velocities are based on the THAMO model (Saiz-Lopez et al., 2008).
[1] Values reported in (Sander, 1999).
[2] Values based on the THAMO model (Saiz-Lopez et al., 2008).
[a] Considering a dissociation constant $K_a = 3.2 \times 10^9$ and a temperature dependent coefficient c = 9800 K
[b] Within the range of values given in the corresponding reference.
[c] Virtually infinite solubility is represented by using a very large arbitrary number.
[d] Value assumed to be equal to those of BrNO$_2$.
[ice] Species for which ice-uptake is considered following (Neu and Prather, 2012).